# Morphological pseudotime ordering and fate mapping reveal diversification of cerebellar inhibitory interneurons

Wendy Xueyi Wang [1,2] & Julie L. Lefebvre [1,2✉]

Understanding how diverse neurons are assembled into circuits requires a framework for describing cell types and their developmental trajectories. Here we combine genetic fate-mapping, pseudotemporal profiling of morphogenesis, and dual morphology and RNA labeling to resolve the diversification of mouse cerebellar inhibitory interneurons. Molecular layer interneurons (MLIs) derive from a common progenitor population but comprise diverse dendritic-, somatic-, and axon initial segment-targeting interneurons. Using quantitative morphology from 79 mature MLIs, we identify two discrete morphological types and presence of extensive within-class heterogeneity. Pseudotime trajectory inference using 732 developmental morphologies indicate the emergence of distinct MLI types during migration, before reaching their final positions. By comparing MLI identities from morphological and transcriptomic signatures, we demonstrate the dissociation between these modalities and that subtype divergence can be resolved from axonal morphogenesis prior to marker gene expression. Our study illustrates the utility of applying single-cell methods to quantify morphology for defining neuronal diversification.

[1] Program for Neuroscience and Mental Health, Hospital for Sick Children, Toronto, Canada. [2] Department of Molecular Genetics, University of Toronto, Toronto, Canada. ✉email: julie.lefebvre@sickkids.ca

Neuronal diversification is essential for the assembly and function of complex nervous systems[1–3]. Neuronal subtypes diverge from shared lineages and develop distinct morphologies and functions[4]. Subtype diversity arises from the interplay of intrinsic genetic programs and extrinsic cues but capturing the underlying dynamics remains challenging. Transcriptome studies provide insights into differentiating neural tissues, including the emergence of cell-type identities, lineage relationships, and timelines of maturation[5,6]. By profiling neurons across development with single-cell transcriptomics, one can trace cell fate and differentiation trajectories[5,7–10]. However, charting neuronal diversification requires additional modalities, such as morphology and developmental context, to determine when phenotypic variations arise and to identify factors that guide differentiation and circuit assembly. Multimodal assessments or cross-modal validations are also important given multiple recent reports of discordance between gene expression signatures with morphology and electrophysiology[11–14].

Morphology is the traditional albeit low throughput modality for cell typing, as it provides intuitive information about how neurons differ based on spatial properties, ontogeny, and connectivity[13,15–18]. New technologies for quantitative morphology are advancing the throughput and resolution for mapping cell types[17,19–23]. Paired with advancements in statistical and computational methods, neuronal subtypes can be effectively parsed using single-neuron anatomy, as illustrated for sensory afferents in the skin[24], olfactory bulb neurons[25], as well as pyramidal and GABAergic interneurons in the cortex[15,16,26]. Drawing fine divisions between subtypes can be challenging however, with increasing examples of continuous variation observed within neuronal types in single-cell transcriptomics studies[13,15,27–29]. A major question driving studies of neural diversity is whether all cells sort into discrete subtypes, given sufficient sampling and granularity, or if continuous and local variation is a biological feature essential for neural processing. A comprehensive framework of neuronal cell types is also crucial for investigating the developmental processes that generate neuronal diversity.

Here, we map the diversification of cerebellar GABAergic interneurons using quantitative morphology, lineage tracing, and in situ analyses of transcriptomic markers. The cerebellar molecular layer interneurons (MLIs) derive from a common progenitor pool and form compact morphologies[30,31]. MLIs encompass an anatomically and functionally diverse population that provides the complement of dendritic-, somatic-, and axon initial segment-targeting inhibition onto principal Purkinje cells. MLIs are classically divided based on morphological features into the basket cells and stellate cells[30]. Basket cells (BCs) are born earlier, populate the lower third of the molecular layer (ML) and form a series of perisomatic basket terminals that enwrap the cell bodies of Purkinje cells. Some BC terminals further specialize into 'pinceaux' formations that align the Purkinje cell axon initial segment[31,32]. By contrast, later-born stellate cells (SCs) integrate into the upper molecular layer where their axons innervate Purkinje cell dendrites. The basket-stellate cell division has long been debated due to the morphological variation that suggests MLIs form one continuously varying population[18,31,33–36]. How MLI diversity arises during development is not known but the postmitotic precursors can remain plastic until they reach target locations and adopt host-specific traits, in contrast to the early subtype commitment that underlies cortical and spinal interneuron diversification[8,37–41]. A barrier to resolving MLI subtypes has been the lack of molecular markers to distinguish BCs and SCs and to track MLI differentiation[31,42,43]. Recent single-cell RNA sequencing studies (scRNA-Seq) of the whole brain or cerebella suggest the existence of MLI transcriptional subtypes, but these molecular signatures have not been systematically compared to BC and SC phenotypes[44,45].

To define MLI subtypes, we applied a cross-modal analysis that combines morphological reconstructions, mapping of novel markers by single-molecule fluorescence in situ hybridization (smFISH), and trajectory inference methods to track phenotypic divergence. We devised a single MLI anatomy platform that: (1) employs genetic tools to sparsely label MLIs across the repertoire of phenotypes, birthdates, and laminar positions throughout development; (2) reconstructs dendritic and axonal morphologies; (3) quantifies phenotypes using morphometric parameters that describe dendritic, axonal, and somatic attributes; and (4) applies unsupervised clustering and pseudotime inference to parse morphological trends among a seemingly heterogeneous neuronal population. We compiled datasets of 79 complete reconstructions of mature MLIs and 732 axonal reconstructions that capture development. Our analyses reveal that mature MLIs sort into two discrete subtypes along morphological (m-) and transcriptional (t-) axes, but that the MLI m-types do not align with recently identified t-types[45]. Morphologically, MLIs separate into canonical basket cells and a continuously heterogeneous population of stellate cells with short- to long-range axons. Transcriptionally, markers belonging to one t-type show continuous variation across basket cells and stellate cells with long axons, while markers of the second t-type are enriched in stellate cells with shorter axons. By adapting pseudotime algorithms recently developed for scRNA-Seq to morphogenesis, we demonstrate that BC and SC identities diverge during neuronal migration and prior to expression of t-type markers. The early subtype divergence of MLIs contrasts with prevailing views that BC and SC phenotypes arise as a function of target laminar position. Our study also presents a novel framework for defining the trajectory of neuronal development using morphological information.

## Results

**Large-scale labeling of MLIs reveals diverse morphologies.** To systematically define MLI morphological diversity, we sought to acquire a high-quality inventory of reconstructions that captures the complete repertoire of mature MLIs (postnatal day 75) across molecular layer positions. Sparse labeling of MLIs was achieved by injection of *Gad2-ires-Cre* mice with AAV vectors encoding multicolor fluorophores at later postnatal stages (Fig. 1a)[46,47]. Although AAV delivery at P0 resulted in the nearly exclusive labeling of Purkinje cells, injections at P5-7 or P10-14 led to enriched labeling of MLIs residing in the deep or superficial ML, respectively (Fig. 1b–e). Images of single MLIs with complete axonal and dendritic arbors were acquired by confocal microscopy and were reconstructed for feature extraction and analyses. We selected neurons in the cerebellar vermis for capturing these planar cells within the sagittal orientation and for consistent molecular layer thickness. We compiled 79 reconstructions of mature MLIs with somas located across laminar positions (Fig. 1f) and extracted 27 features that quantify their dendrites, axon, soma, and location (Supplementary Table 1). Our dataset contains high-resolution anatomical information of MLIs sampled across molecular layer locations.

We began with a manual expert classification of the reconstructions to determine the proportion of MLIs that exhibit canonical basket cell (BC) or stellate cell (SC) characteristics[36,48]. Canonical BCs were identified by soma location in the lower third of the molecular layer, a fan-shaped dendritic arbor that reaches the superficial ML, and a long axonal projection that forms multiple basket terminals onto Purkinje cell somas (16 of 79 cells; Fig. 1g, h). Cells with stereotypical SC features were identified by

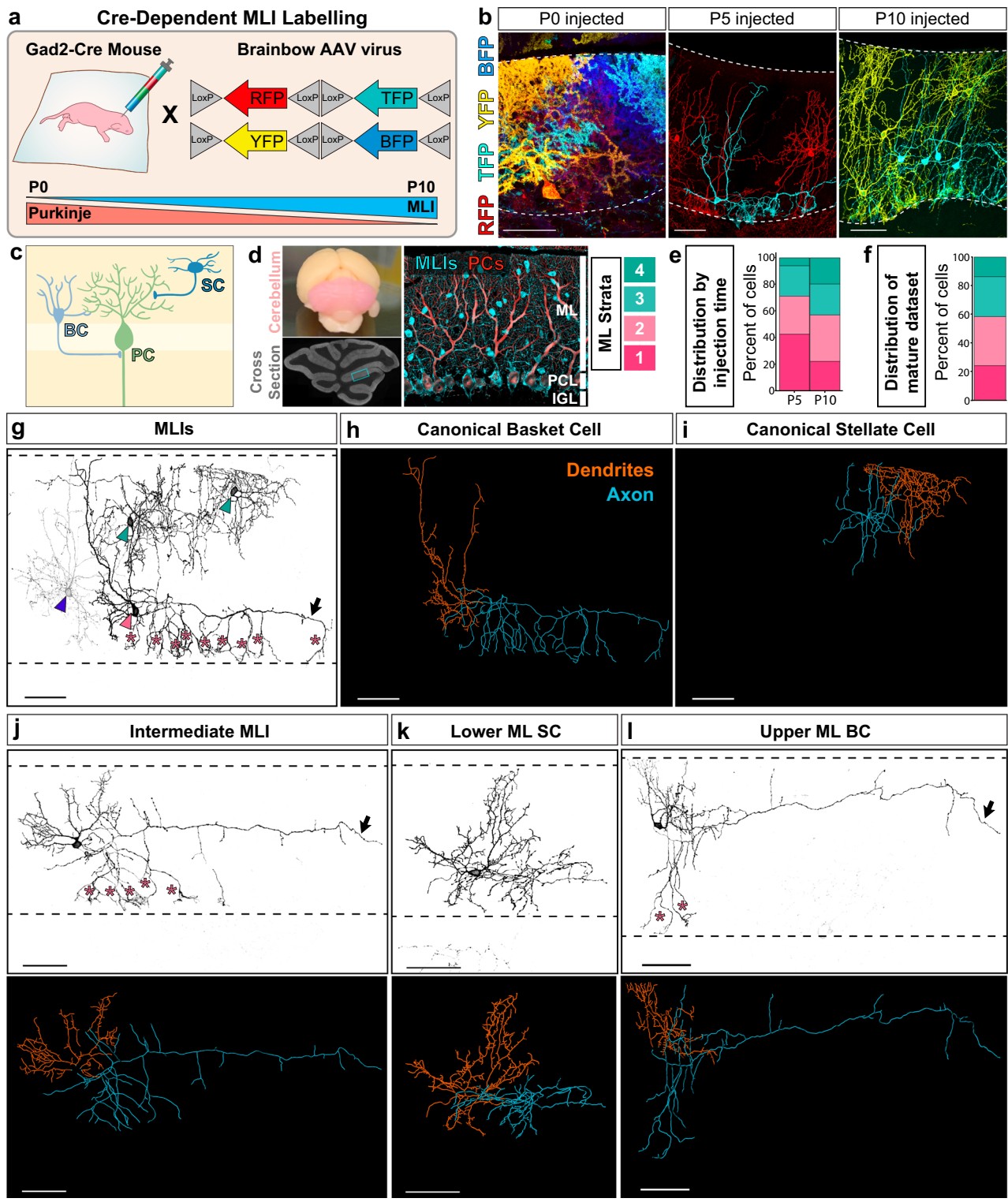

their location in the superficial molecular layer and highly branched, radial dendritic arbors (25 of 79 cells; Fig. 1g, i). Nearly half of the MLIs in our dataset do not fit into either category (non-canonical MLIs, 38 of 79 cells). They include MLIs located in the middle of the molecular layer displaying mixtures of BC and SC morphologies that vary with laminar position (Fig. 1j), as described previously[35,36]. We also observed MLIs with morphologies that do not correlate with molecular layer depth. For example, MLIs lacking basket formations and resembling SCs were observed in the deeper third of the molecular layer (Fig. 1k),

and MLIs extending axon terminals that reached the base of PC somas were detected in the superficial layer (Fig. 1l). Thus, qualitative assessments reveal extensive heterogeneity in axonal and dendritic morphologies among MLIs located across laminar positions.

**MLIs form discrete morphological cell types, with continuous variation within stellate cells**. To test if MLIs can be distinguished into morphological subtypes (m-types), we applied unsupervised clustering methods to analyze the dataset of single

**Fig. 1 A platform for labeling MLI morphologies in the cerebellum. a** Single MLI labeling with Cre-dependent Brainbow AAVs encoding multiple fluorescent proteins injected into Gad2-Cre mice between P0 to P10. **b** Cross-sections of mature cerebellar cortex show enriched labeling of Gad2-Cre positive subpopulations. AAV delivery at P0 predominantly labels Purkinje cells (PC); P5, lower MLIs; P10, upper MLIs. Dashed lines denote molecular layer (ML) boundaries. **c** Schematic of canonical innervation patterns of basket (BC, light blue) and stellate (SC, dark blue) cells residing in the lower and upper ML, respectively (adapted with permission from ref. [110]). **d** Mouse cerebellum (pink), and its cross-sectional morphology. ML outlined in cyan. Immunostaining of MLIs (anti-Parvalbumin (PV), cyan) shows distribution throughout the ML. PCs are co-labeled with PV (cyan) and calbindin (red). ML was divided into four strata to record the laminar position of labeled MLI soma. **e** Percent laminar distribution of AAV-labeled single MLIs. Representative images (**b**) and quantifications (**e**) are from at least three animals per injection time point. **f** Percent laminar distribution of the 79 mature single MLI reconstructions. **g–i** Canonical MLI morphologies. **g** Inverted fluorescent image of four labeled MLIs. The lower MLI (pink arrowhead) displays canonical BC morphology, including axonal basket terminals targeting the PC somas and axon initial segments (AIS; asterisks). The upper MLIs (teal arrowhead) have canonical SC morphologies. The faintly labeled MLI (blue arrowhead) has SC morphological characteristics but resides within the lower ML. **h** Reconstruction of canonical BC and, **i** SC from panel **g** with dendritic (orange) and axonal arbor traces (blue). **j–l** Representative images (top) and reconstructions (bottom) of MLIs showing mixtures of BC and SC characteristics. **j** MLI located in the middle ML with SC dendritic features and a long axon (arrow) with collaterals that reach the PC soma base or AIS (asterisks). **k** MLI in the lower ML with SC-like arbors. **l** MLI in the upper ML with a long axon (arrow) and two collaterals enveloping PC somas (asterisks). Images (**g–l**) and quantifications (**f**) are from 79 cells, $N = 9$ animals. Scale bars are 50 μm.

morphometric properties. Uniform Manifold Approximation and Projection (UMAP)[49], hierarchical clustering, and partition-based graph abstraction (PAGA)[50] all produced two major classes of MLI morphologies (Fig. 2a–c). The first group contained basket cells displaying canonical features and corresponding to the manually curated BCs (hereafter BCs; $n = 19$ cells; Fig. 2d). By contrast, the second group was larger and heterogeneous, comprising canonical SCs and the remaining non-canonical MLIs (hereafter SCs; $n = 60$ cells; Fig. 2e–i). The separation of BC and SC m-types was robust, as clustering of 20 randomly selected cells was sufficient to recapitulate the division (Fig. 2j; 96 ± 1.26%).

It has been proposed that MLIs model a single population that varies continuously with laminar positioning within the molecular layer[35,36]. Given the robust division of BC and SCs, we next examined whether variations within the SC class alone account for the reported continuous heterogeneity. Cells within the SC cluster displayed continuous morphological variation ranging from cells with short- to long-reach axons (Fig. 2b, c, e–i). A subset of long-range SCs extended axon collaterals resembling basket formations that partially enveloped PC somas and reached the soma base or axon initial segment (Fig. 2h; see also Fig. 1j, l) but were otherwise indistinguishable from non-soma-targeting SCs (Fig. 2g). We next evaluated the SC variation by PAGA, which is commonly used to assess continuity of single cells datasets[12,45,50]. PAGA detected connectivity between two nodes within the SC m-type, which upon inspection, correspond to the shorter-range and longer-range SCs, respectively (Fig. 2c). We also confirmed that individual morphological features, such as axon span, presented continuous transitions among SCs when ranked by hierarchical clustering order (Fig. 2k), and that no individual feature exhibited bimodality in distribution (Supplementary Fig. S1a). Finally, iterative hierarchical clustering of SCs did not reliably reproduce SC subgroups (classification accuracy = 89 ± 2.3% for 70 cells) (Fig. 2j). Together, these analyses support a discrete division between BCs and SCs, which is consistent with the classical two population model, while the extensive heterogeneity within the SC group explains the continuous variation that was the initial basis for the one MLI population model.

**Basket and stellate cells are distinguished by axonal signatures.** To define the features most informative for MLI morphological type classification, we repeated the clustering analyses using recursive feature elimination, in which single or groups of morphological features were removed. The BC and SC division remained following the elimination of all dendritic features and, interestingly, elimination of soma position within the molecular layer (Fig. 3a, b). By contrast, BC and SC clustering was lost upon

removal of the six axonal features (Fig. 3c). Restricting UMAP to the axonal parameters recapitulated the BC and SC clusters with all but one cell correctly sorted (Fig. 3d). Therefore, despite the anatomical complexity and heterogeneity, axonal information is necessary (Fig. 3c) and sufficient (Fig. 3d) for the classification of MLIs into BC or SC types. Additionally, BC-SC clustering remained upon removal of the feature describing axon terminals that enwrap PC somas (Fig. 3e; Supplementary Fig. S1b; see Weighted Basket scale in Methods). Thus, BC versus SC classification cannot be defined by the presence of PC soma-targeting terminals alone. Finally, the axonal parameters co-vary only weakly, if at all, with soma position in the molecular layer (e.g. PC soma-targeting terminals, $r^2 = 0.30$; axon span $r^2 = 0.13$; Supplementary Fig. S2), further indicating that BC and SC identities are divided on the basis of axonal morphology rather than laminar position.

**MLI transcriptomic and morphological subtypes do not align.** To determine if basket and stellate morphological (m-) types can be distinguished by gene expression signatures, we analyzed a recently reported single nucleus RNA sequencing (snRNA-Seq) dataset of the adult mouse cerebellum[45,51]. This study identified two discrete subpopulations, MLI1 and MLI2, but surprisingly these transcriptionally-defined subtypes did not obviously correlate with BC or SC features nor with a laminar organization within the molecular layer. To further investigate the molecular profiles, we extracted from this dataset 43,479 cells with a putative MLI identity by sub-setting the two clusters with positive expression for *Gad2*, *Pvalb*, *Prkcd* RNA and negative expression for other cerebellar cell-type markers. Consistent with Kozareva et al, MLIs separated into two transcriptional (t-) types by UMAP visualization, and one t-type exhibited within-class heterogeneity (Fig. 4a). Differential gene expression analyses confirmed that MLI t-types were distinguished by *Sorcs3* or *Nxph1* expression[51]. We also identified other transcripts expressed in enriched or overlapping patterns among the two clusters (Fig. 4b, c). The *Sorcs3*+ group was divided into two subclusters by elevated expression of *Grm8* or *Cacna1e*. Previously identified BC-enriched immunohistological markers NefH and Ret[52,53] showed enrichment within the *Sorcs3*+/*Grm8*+ subcluster but they were not among the top differentially expressed transcripts (Fig. 4b, c).

We next mapped the expression of the transcripts and the spatial organization of MLI t-types within the cerebellar cortex by performing single-molecule fluorescence in situ hybridization (smFISH)[54,55]. We selected two genes that identify all MLIs (*Gad2* and *Parvalbumin*), four genes differentially expressed between the two MLI t-types (*Sorcs3*, *Nxph1*, *Cdh22*, and *Gjd2*) as

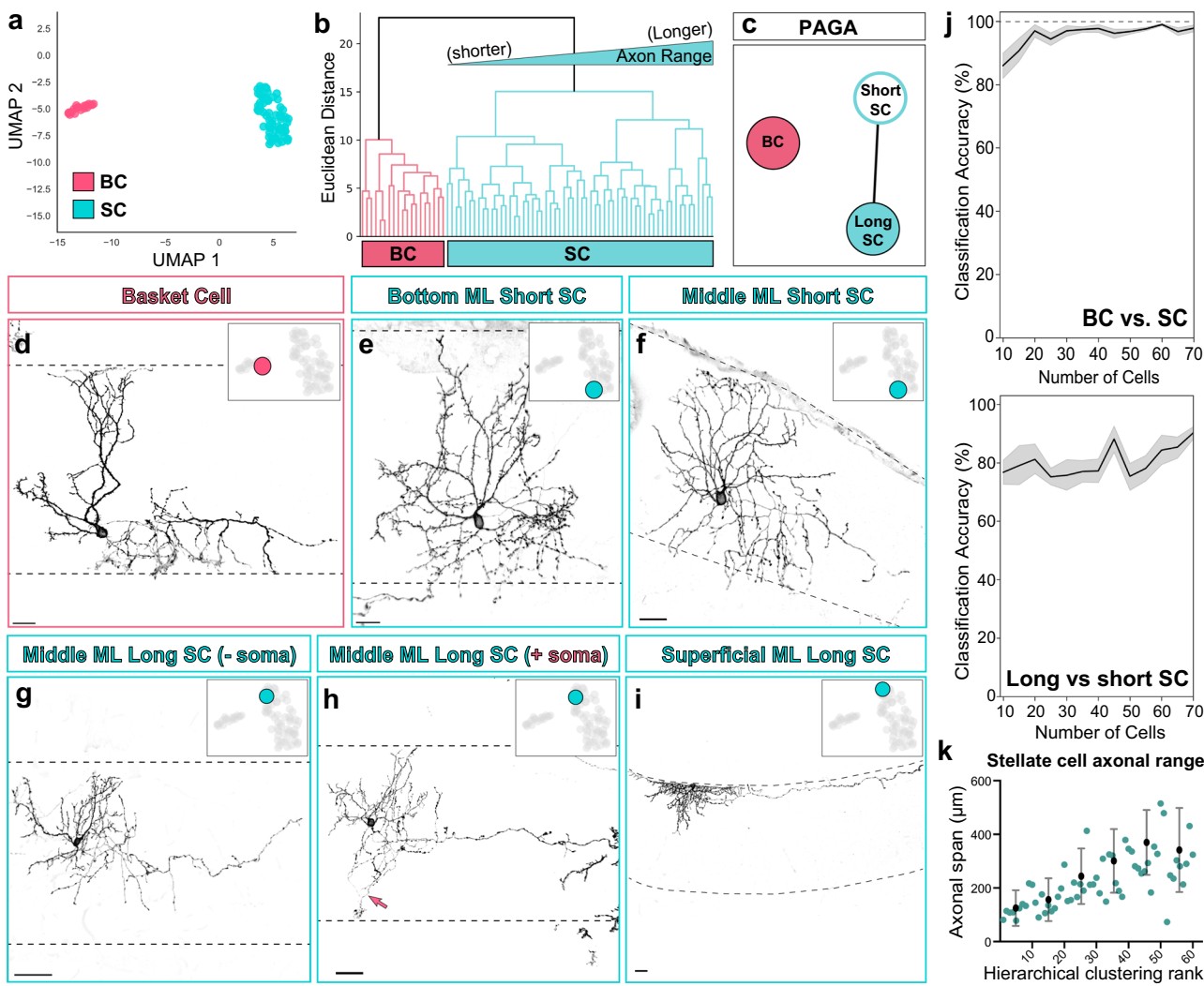

**Fig. 2 Clustering of mature MLI morphologies reveals discrete and continuous heterogeneity. a** UMAP plot separates 79 mature MLI morphological reconstructions into discrete basket (BC, pink) and stellate cell (SC, teal) clusters. Each data point represents a single MLI based on morphometric measurements. **b** Hierarchical clustering separates MLIs into two BC (pink) and SC (teal) clades. Within the broad SC cluster, the dendrogram ordered MLIs from shorter to longer axonal range cells. **c** PAGA separates MLIs into discrete BC and SC clusters. SCs are subdivided into two nodes connected through an edge, denoting continuous variation in morphology. **d–i** Examples of MLIs and their locations within the UMAP manifold. ML boundaries are outlined by dashed lines. **d** MLI from the BC cluster displays canonical BC morphology. **e** An MLI at one pole of the SC cluster displays short-range axonal morphology and its soma resides within the lower ML. **f** An MLI plotted within similar coordinates of the UMAP manifold as **e**, displays short-range axonal morphology but resides within the middle ML. **g** An MLI at the opposite pole of the SC cluster displays long-range axonal morphology. **h** An MLI which resides within similar coordinates of the UMAP as **g**, extends descending PC soma-targeting axon collaterals (pink arrow at PC soma; axon terminal reaches the PC soma base). **i** An MLI located in the very superficial ML extends long axonal range collaterals. **j** Iterative clustering using random subsamples shows that MLI classification into BC/SC clades is robust, as cross-validation using as few as 20 subsampled cells replicates the BC/SC division. By similar measures, classification of SCs into long- and short-range subclades is not robust. Lines show means drawn from 20 trials for each sample size, with SEM represented by shaded regions. **k** Axonal span of SCs vary continuously from shorter-range to longer-range cells, ranked by hierarchical clustering rank. Error bars represent mean and SD of the data, *n* = 61 SCs in UMAP, 60 in hierarchical clustering; 18 BCs in UMAP, 19 in hierarchical clustering. Scale bars are 20 μm in **d–f**, 40 μm in **g–i**.

reported in ref. [51], two genes differentially expressed between the two *Sorcs3*+ MLI subtypes (*Grm8* and *Cacna1e*), two established BC-enriched genes (*NefH* and *Ret*), as well as two genes which showed overlapping and opposing gradient expression between *Nxph1*+, *Sorcs3*+/*Cacna1e*+, and *Sorcs3*+/*Grm8*+ MLI subtypes (*Lrp1b* and *Ptprt*; Fig. 4d). All *Pvalb*+ MLIs within the molecular layer expressed high levels of either *Sorcs3* or *Nxph1* in non-overlapping cells (Fig. 4e, f). Interestingly, a subset of markers showed a laminar pattern with enriched expression within the bottom (*Grm8, Ret, NefH,* and *Lrp1b*) or top (*Cacna1e* and *Nxph1)* molecular layer (Fig. 4d). Furthermore, *Nxph1* expression

is highly enriched in the top 80% of the molecular layer, where SCs reside (Fig. 4e, f). Thus, smFISH demonstrates that all MLIs can be classified into either *Sorcs3* or *Nxph1* expressing cells, and that a subset of t-type markers exhibits a laminar organization.

To identify relationships between MLI m-types and the t-type markers, we performed smFISH on cerebellar tissue labeled for morphology (morFISH). We annotated single MLIs into BC or SC morphological types, and quantified the number of *Sorcs3*, *Nxph1*, *Grm8*, or *Cacna1e* RNA puncta within each cell (Fig. 5a–d). Within the SC m-type, we further focused on cells at extreme ranges of short- and long-axon phenotypes. All BCs

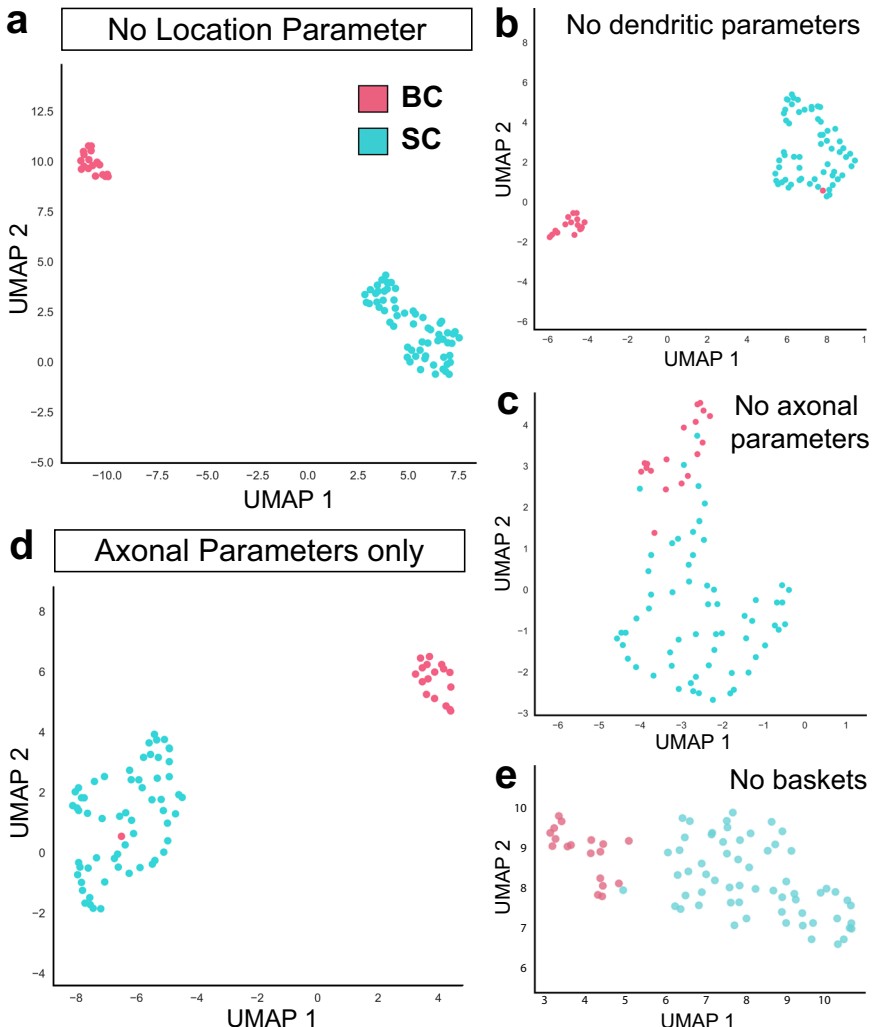

**Fig. 3 Axonal information is necessary and sufficient for MLI morphological subtyping. a** UMAP plots of BC (pink) and SC (cyan) clusters following recursive elimination of morphometric parameters describing MLI location. **b** UMAP following elimination of dendritic parameters. **c** UMAP following elimination of axonal parameters. **d** UMAP with only six axonal parameters is sufficient for BC/SC clustering. **e** UMAP following elimination of PC soma terminal/weighted basket feature.

analyzed expressed *Sorcs3* and high levels of *Grm8* RNA, and little *Cacna1e* (Fig. 5a, d). BCs do not express *Nxph1*. By contrast, *Sorcs3*+ cells in the middle and upper molecular layer are long-range SCs that express increasing levels of *Cacna1e* and decreasing *Grm8* (Fig. 5b, d). Lastly, morFISH identified *Nxph1*+ cells as short-range, non-soma-targeting SCs in the low-mid molecular layer (Fig. 5c, d). However, *Nxph1* may not be limited to short axon range morphologies, as *Nxph1*+ SCs with longer arbors were observed in the superficial molecular layer, intermixed with *Sorcs3*+ SCs (Fig. 5e). Together, morFISH confirms that the two t-types do not align with the BC/SC m-types. However, MLI m-types can be distinguished by expression of t-type marker combinations. The *Sorcs3*+ MLI t-type displays continuous molecular variation through *Grm8* and *Cacna1e*. The BC m-type and longer axon SCs can be distinguished as *Sorcs3*+; $Grm8^{HIGH}$, and *Sorcs3*+; $Grm8^{LOW}$; $Cacna1e^{HIGH}$, respectively (Fig. 5f, g). The molecularly discrete *Nxph1*+ MLI subtype corresponds largely to SCs with short-axon arbors, and to smaller numbers of SC morphologies along the continuum in the superficial layer.

Although the discrete MLI t-types are mosaically arranged within the molecular layer[51] (Fig. 4e), we wondered whether the continuous molecular variation of *Sorcs3*+ cells observed in the

transcriptomic data suggests a spatially graded molecular organization of MLIs. To this end, we quantified the spatial expression of the four MLI markers (*Sorcs3*, *Nxph1*, *Grm8*, and *Cacna1e*) simultaneously by smFISH within single cells. For each *Sorcs3*+ or *Nxph1*+ DAPI nuclei, we created a somatic mask and quantified the number of smFISH puncta for each target gene at the single-cell level (Fig. 6a). The somatic masks were used to spatially register each cell within the molecular layer. We performed UMAP and louvain graph-based clustering using the smFISH puncta quantifications, and projected single-cell spatial information onto the resulting manifold. Analysis of the four markers at P55 reproduced the clustering of MLI t-types (Fig. 6b), including an opposing gradient of *Grm8* and *Cacna1e* across molecular layer depth (Fig. 6c, d, h). *Grm8* expression was restricted to *Sorcs3*+ cells, while *Cacna1e* overlapped with *Sorcs3*+ and *Nxph1*+ populations, consistent with snRNA-Seq data (Figs. 6d and 4c). We extended similar analyses to P17 to determine if these laminar patterns are established by the third postnatal week when MLIs are maturing and reach peak density[56]. Although the global structure of the UMAP manifold was preserved for P17 smFISH, the division of *Grm8*+ and *Cacna1e*+ expressing cells was less defined (Fig. 6e–i). To test if individual subtype markers are initially expressed more

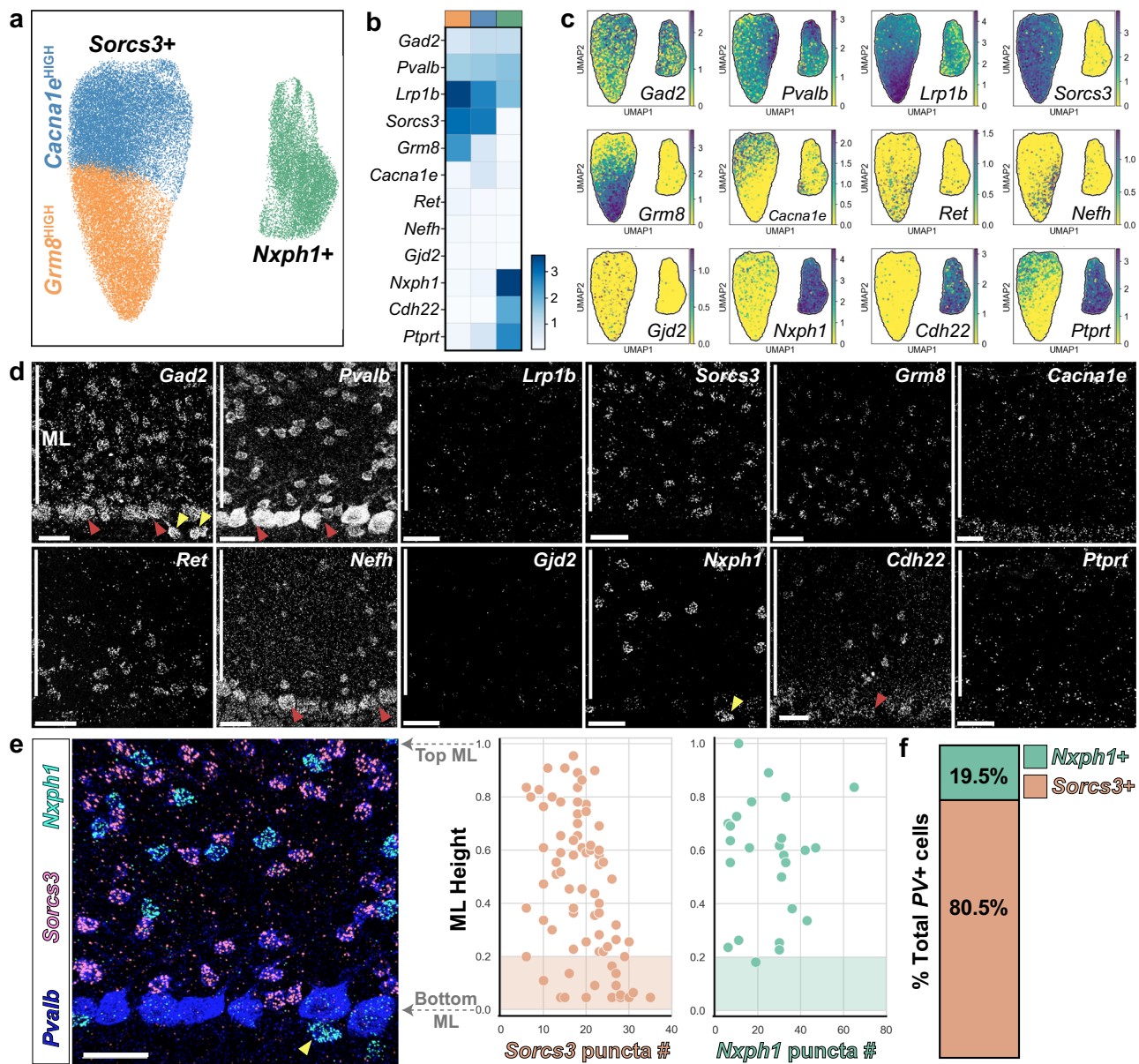

**Fig. 4 Clustering and in situ analyses of mature MLI transcriptional signatures reveal discrete and laminar organization of MLI t-types. a** UMAP showing 43,479 MLIs separating into two clusters (*Sorcs3*+ and *Nxph1*+) based on public single nuclei-extracted RNA sequencing data[51]. The *Sorcs3*+ cluster subdivides into *Grm8*^HIGH and *Cacna1e*^HIGH subclusters. **b** MatrixPlot showing the scaled expression level for transcripts of interest among transcriptional subtypes, *Sorcs3*+; *Grm8*^HIGH (orange column), *Sorcs3*+; *Cacna1e*^HIGH (blue column), and *Nxph1*+ (green column). **c** Transcript levels projected onto UMAP. **d** smFISH images of MLI transcripts within the cerebellar cortex (ML marked by white bars). Red and yellow arrowheads mark Purkinje cells and Purkinje layer interneurons, respectively. N = 2–6 animals. **e** Left, three-color smFISH for *Pvalb* (blue), *Sorcs3* (magenta), and *Nxph1* (cyan). *Sorcs3* and *Nxph1* are expressed in non-overlapping cells within the ML. *Nxph1*+ cells observed below the Purkinje cell layer are likely Purkinje layer interneurons (yellow arrowheads). Right, MLI smFISH quantification and mapping show that *Sorcs3*+ cells are distributed throughout ML while *Nxph1*+ cells are enriched in the upper 80% of the ML (n = 95 total MLIs from two animals). **f** *Pvalb*+ cells are either *Nxph1*+ (green) or *Sorcs3*+ (orange). Scale bars are 30 μm.

broadly across cells, we compared RNA puncta levels for each marker within the *Nxph1*+, *Sorcs3*+/*Grm8*^HIGH, and *Sorcs3*+/*Cacna1e*^HIGH t-type subclusters (Fig. 6j). *Nxph1* and *Sorcs3* expression segregated with their non-overlapping subtypes at P17, as at P55. By contrast, *Grm8* and *Cacna1e* levels were elevated across clusters at P17 but became increasingly restricted within their respective subclusters by P55 (Fig. 6j). To confirm that the overlapping expression patterns were not due to caveats of the clustering protocol, we quantified co-expression of *Cacna1e* and *Grm8* within single cells. Interestingly, 39.3% of *Sorcs3*+ cells and 27% of *Nxph1*+ cells co-expressed both *Cacna1e* and

*Grm8* at P17. This contrasts with P55, when only 6.7% of *Sorcs3*+ cells and 0% of *Nxph1*+ cells co-expressed *Cacna1e* and *Grm8*.

Lastly, we examined the onset of marker expression during MLI development. During the first two postnatal weeks, Pax2-expressing MLI precursors enter the molecular layer and migrate to their final laminar locations, after which they downregulate *Pax2* and upregulate *Pvalb* expression[40,57–61]. At P7, *Sorcs3* and *Nxph1* are expressed in non-overlapping cells within the molecular layer, while *Grm8* RNA is detected at low levels in subsets of *Sorcs3*+ cells (Supplementary Fig. S3). Co-staining with *Pax2* and *Pvalb* revealed that *Sorcs3* and *Nxph1* are expressed

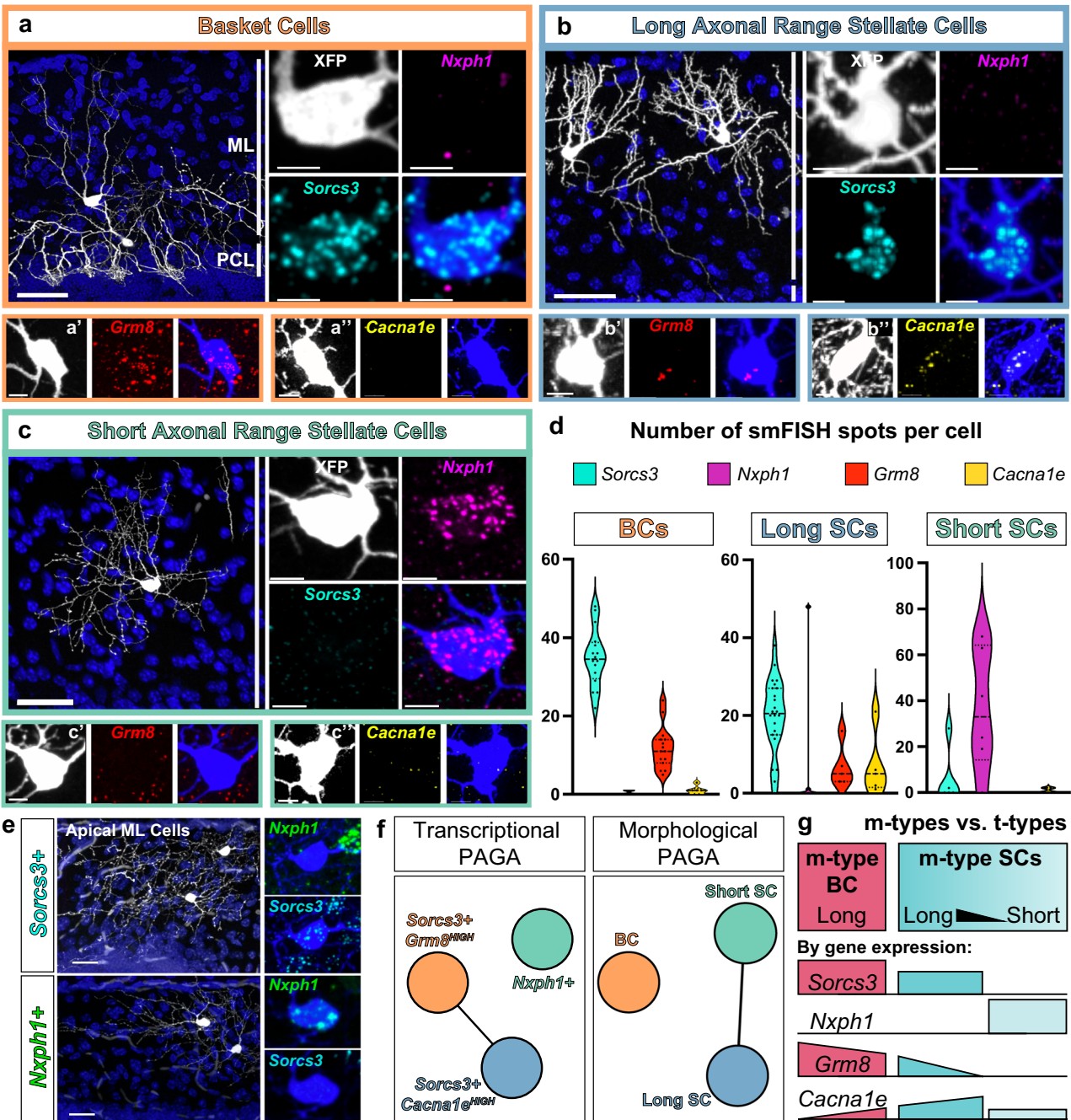

**Fig. 5 Morphologically defined MLI types do not align with transcriptionally defined signatures but can be described by marker combinations.**
**a–c** morFISH co-staining of single fluorescently-labeled MLIs with smFISH for t-type markers *Sorcs3, Nxph1, Grm8,* or *Cacna1e*. Each panel (i.e. **a, a′, a″**) shows three MLI soma from separate morFISH co-stainings. **a** m-type basket cell expresses *Sorcs3* but not *Nxph1*, and expresses high *Grm8* and low *Cacna1e* (**a′, a″**). **b** m-type stellate cells with long-range axons express *Sorcs3* but not *Nxph1*, and express *Cacna1e* and lower *Grm8* (**b′, b″**). **c** m-type stellate cells with short-range axons express *Nxph1* and lower *Sorcs3*; *Grm8* is absent but low levels of *Cacna1e* are detected (**c′, c″**). **d** Violin plot summarizing smFISH puncta quantifications for markers within morphologically defined BCs (left, n = 16 cells), long-range SCs (middle, n = 24 cells), and short-range SCs (right, n = 5 cells). Note the *Sorcs3+; Grm8*$^{HIGH}$; *Cacna1e*$^{LOW}$ pattern among the BCs, the *Sorcs3+; Grm8*$^{LOW}$; *Cacna1e*$^{HIGH}$ among m-type long SCs, and the exclusion of *Grm8* in short-range SCs (**c′**). **e** SCs with long-range axons in the upper ML are either *Sorcs3+* or *Nxph1+*. Images and quantifications from **a–e** are from four animals. **f** Comparison of MLI transcriptional and morphological identities, as assessed through PAGA. By transcriptional signatures, continuous heterogeneity is present between long-range BCs (orange) and SCs (blue), while short-range SCs (teal) are discretely separated from both transcriptional subtypes. By morphological signatures, there is a discrete division between BCs and SCs, with continuous heterogeneity spanning long-range and short-range SCs. **g** Schematic summarizes the expression patterns of t-type transcriptional markers among m-type MLIs. Scale bars are 35 μm in **a–d**; 20 μm in **e**; 5 μm in inset, **a′, a″–c′, c″**.

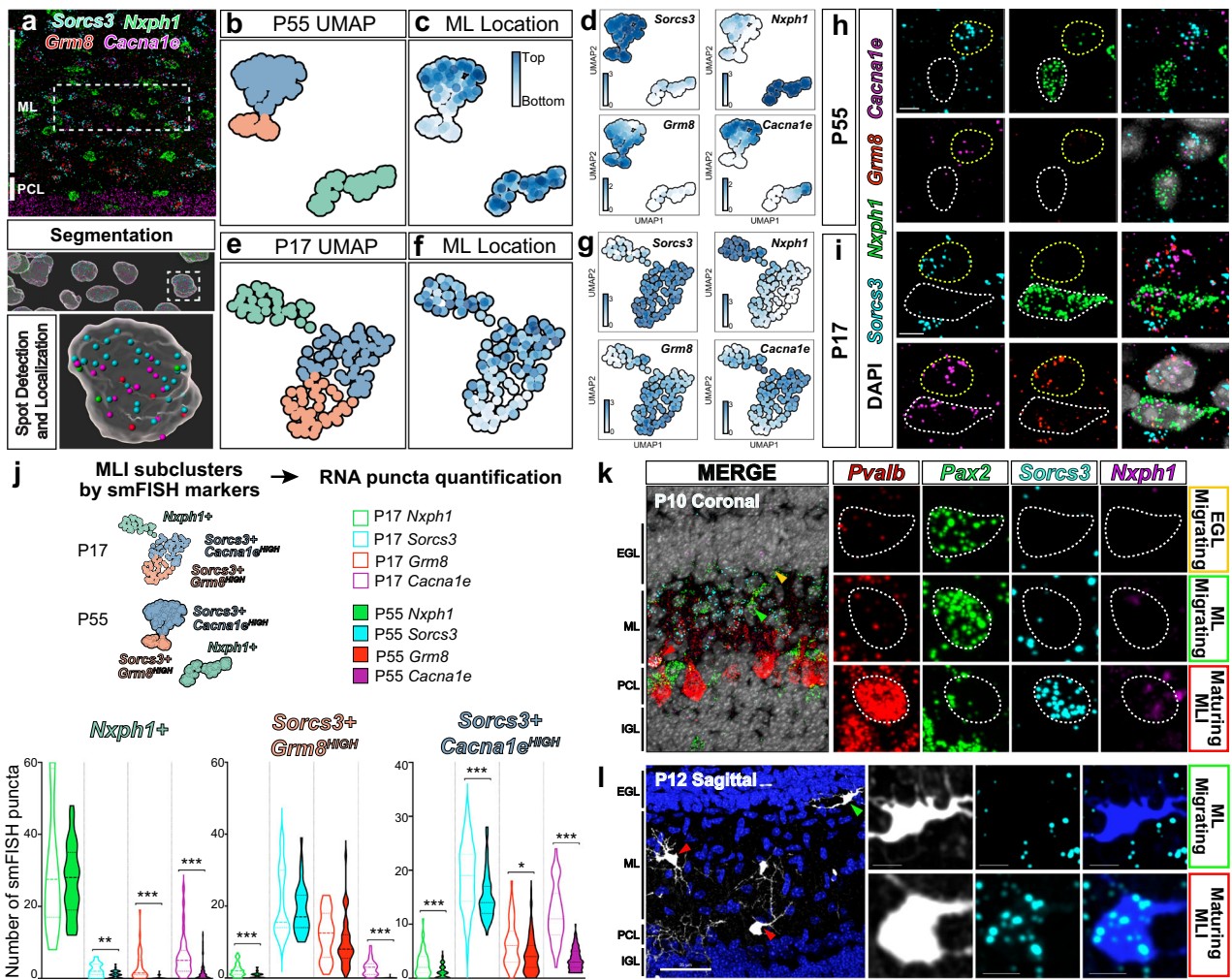

**Fig. 6 Spatial-temporal analysis of markers show that MLI t-type identities appear late in development. a** Top, four-color smFISH image for MLI t-type markers *Sorcs3, Nxph1, Grm8,* and *Cacna1e,* in cerebellar cortex. Bottom, single-cell smFISH spot quantification for each transcript within segmented cell boundaries. **b** UMAP clustering using smFISH RNA data for 232 P55 MLIs. **c** Projection of ML locations, and **d** transcript levels for each cell onto the P55 UMAP plot. The *Sorcs3+* cluster contains two domains: cells within the orange subcluster are located in the lower ML and express high *Grm8;* cells in the blue subcluster are enriched within the mid-upper ML and show graded *Grm8* and *Cacna1e.* Cells within the green cluster are located lower-mid to high ML and express *Nxph1.* **e** UMAP clustering using smFISH RNA data for 126 P17 MLIs. **f** ML locations, and **g** transcript levels for each cell projected onto the P17 UMAP manifold. **h** Representative smFISH image for P55 MLIs showing *Sorcs3* (cyan) and *Cacna1e* (right, magenta), and another with *Nxph1* (green). **i** smFISH image for P17 MLIs showing co-expression of *Cacna1e* and *Grm8* (top cell, red), and co-expression of *Nxph1, Grm8,* and *Cacna1e* (bottom cell). **j** Violin plot summarizing the transcript levels of individual markers within each smFISH-marker-defined subcluster for P17 (unfilled bars) and P55 data (filled bars). Images and quantifications for P17 data are: $n = 126$ cells, two animals; P55 data: 232 cells, two animals. *Nxph1+* t-type: $p = 0.0004$ (P17 vs. P55 *Sorcs3* expression); $p < 0.00001$ (*Grm8*); $p < 0.00001$ (*Cacna1e*). *Sorcs3+; Grm8*[HIGH] t-type: $p = 0.000022$ (*Nxph1*); $p < 0.00001$ (*Cacna1e*). *Sorcs3+; Cacna1e*[HIGH] t-type: $p < 0.00001$ (*Nxph1*); 0.000045 (*Sorcs3*); 0.0012 (*Grm8*); <0.00001 (*Cacna1e*). Means of two groups were compared using the two-tailed Mann-Whitney nonparametric test. **k** smFISH image of P10 coronal cerebellum for both MLI t-type markers (*Sorcs3* in cyan, *Nxph1* in magenta) and maturity markers (*Pvalb* in green, *Pax2* in red). *Sorcs3* and *Nxph1* RNA are absent within migratory *Pax2+* MLIs, but are expressed within post-migratory *Pvalb+* cells. **l** morFISH confirms that *Sorcs3* is not observed within cells with migratory morphology ($N = 2$ animals). Scale bars are 5 μm in **h** and **i**, 35 μm in **k** and **l**.

within maturing *Pvalb*[HIGH]/*Pax2*[LOW] cells but not within *Pax2*[HIGH] MLI precursors (Fig. 6k). By morFISH, we confirmed that the t-type markers are detectable in cells with axonal arbors in the lower layer but absent in cells with immature migratory morphologies in the superficial layer[62,63] (Fig. 6l). Taken together, the onset of t-type marker expression occurs within maturing *Pvalb+* MLIs, once settled in their final position, and the laminar and t-type specific patterns become more refined at maturity.

Altogether, smFISH analyses comparing morphological and molecular identities led to the following findings. First, the two discrete MLI1/MLI2 t-types do not align with the BC/SC m-types.

Second, in spite of the discordance, some of the transcripts within the t-types are expressed in a graded manner and mark the m-types: BCs express *Sorcs3; Grm8*[HIGH]; *Cacna1e*[LOW]; longer axon SCs in the upper molecular layer express *Sorcs3+; Grm8*[LOW]; *Cacna1e*[HIGH]; and short-axon SCs express *Nxph1.* Third, expression of the molecular markers occurs at late stages of MLI development, as MLIs settle at final locations and express *Pvalb.* BC and SC phenotypes are influenced by laminar locations[31,37-40,58], but whether MLI subtype identities diverge at this stage or earlier has not been examined. The late onset of marker expression is not informative for tracking MLI divergence. Since MLI m-types are robustly divided by axonal

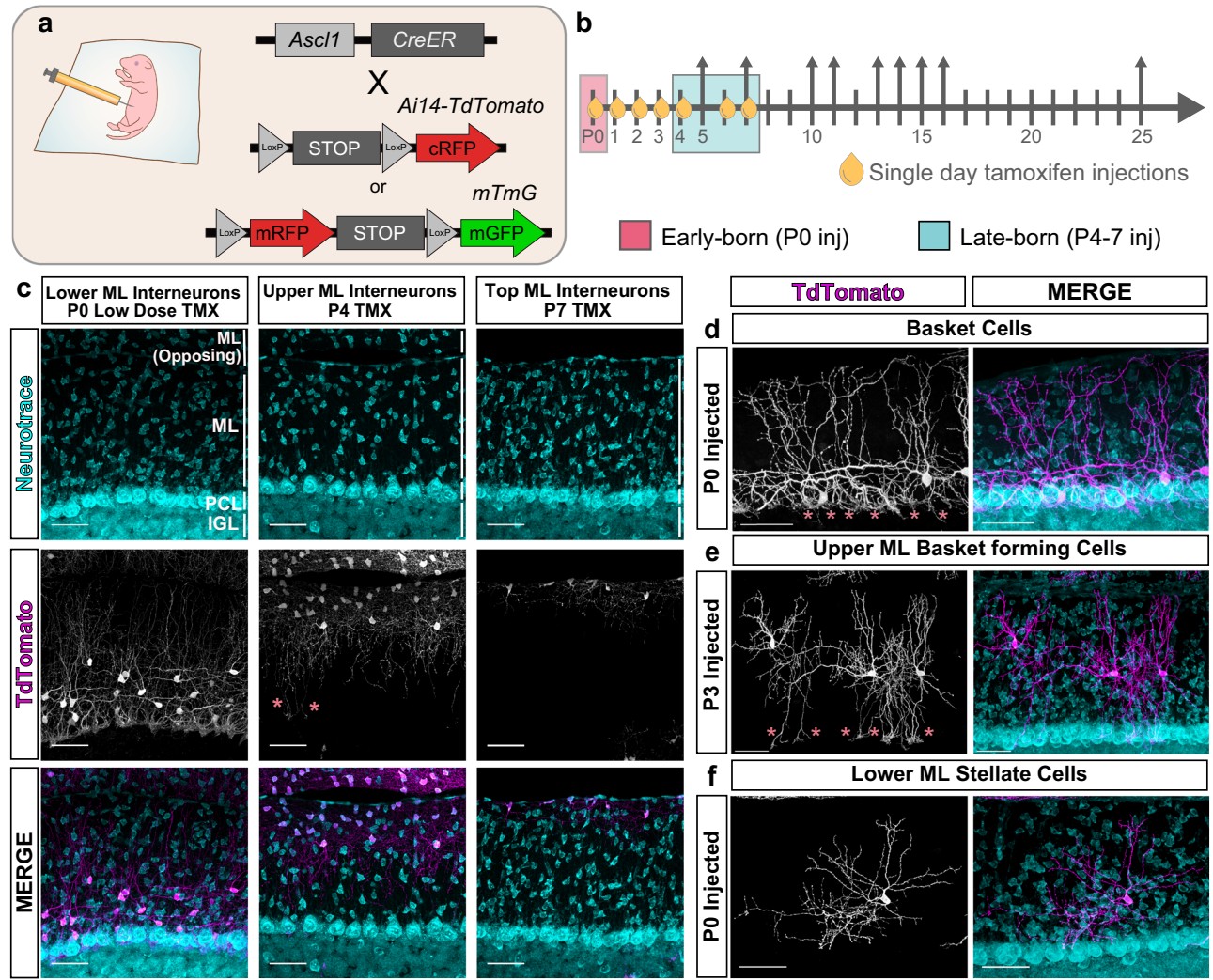

**Fig. 7 Birthdate-dependent targeting of morphological MLI subtypes by Ascl1-CreER. a** MLI labeling strategy by tamoxifen (TMX) inducible Ascl1-CreERT2 and fluorescent Cre reporters, Ai14-TdTomato (cytosolic) or mTmG (membrane-targeted). **b** Schematic for TMX-induced subtype-enriched MLI labeling, with postnatal time points of single dose TMX injections (oil droplets) and collection (arrows). Injection at P0 predominantly labels BC populations (early-born, pink), while P4-7 injections label SCs in the upper ML (late-born, teal). **c** Confocal images of cerebellar sections from P25 Ascl1-CreER; Ai14-TdTomato mice with TMX induction at P0, P4, or P7. **d** P0-induced BCs in the lower ML. **e** P3-induced SCs in the mid-upper ML with descending basket terminals. Pink asterisks denote basket formations enveloping PC soma in **c-e**. **f** P0-induced non-basket forming SC in the lower ML. Representative images were taken at P25. N = 3–5 animals per injection time point (**c-f**). Scale bars are 50 μm.

information, we next investigated whether MLI divergence could be traced by quantifying differences in axon morphogenesis.

**Fate mapping and pseudotime modeling of MLI m-type development.** Birthdating studies have shown that early-born MLIs that first settle in the lower molecular layer adopt BC morphologies, while later-born MLIs that fill the upper layer become SCs[37–40]. To label developing BCs and SCs born on different days, we employed a genetic fate mapping strategy using the tamoxifen-inducible *Ascl1-CreERT2* mouse line (Fig. 7a, b). Similar to previous studies[39,64], intraperitoneal tamoxifen (TMX) injection of *Ascl1-CreER; Rosa^{mTmG}* or *Rosa^{Ai14:STOP-TdTomato}* animals at P0 predominantly labeled BCs that innervate the lower molecular layer (Fig. 7c, d). TMX delivery at P7 marked SC-like cells of the superficial molecular layer (Fig. 7c). MLIs labeled at intermediate stages (i.e. P2-5) exhibited continuous SC phenotypes with fewer PC soma-targeting axon collaterals and increasingly superficial laminar positions (Fig. 7c, e). These findings support the inside-out layering of MLIs within the

molecular layer that is related to birth order[38]. Intriguingly, the short axon SCs that reside within the lower molecular layer were marked at P0, and to a smaller extent at P1 (Fig. 7f). This labeling suggests the existence of an early-born SC subpopulation that shares a birthdate and laminar fate with BCs but adopts a distinct non-basket forming phenotype.

We hypothesized that if MLIs diverge into BC and SC phenotypes early during differentiation, the two m-types should progress through distinct developmental trajectories. Alternatively, if MLI type differentiation is dependent on late positional cues, then migratory MLI precursors will exhibit similar developmental progressions until they settle and elaborate type-specific axonal arbors. To distinguish between these possibilities, we compared the morphologies of genetically fate-labeled BCs and SCs during development. We generated a large dataset of MLI traces collected over a series of time points spanning MLI morphogenesis (P5-25). In total, axonal arbors of 732 MLIs were reconstructed and quantified for 28 morphometric parameters describing their axon, soma, and location (Supplementary Table 2). Each reconstruction was annotated for the cell's inferred

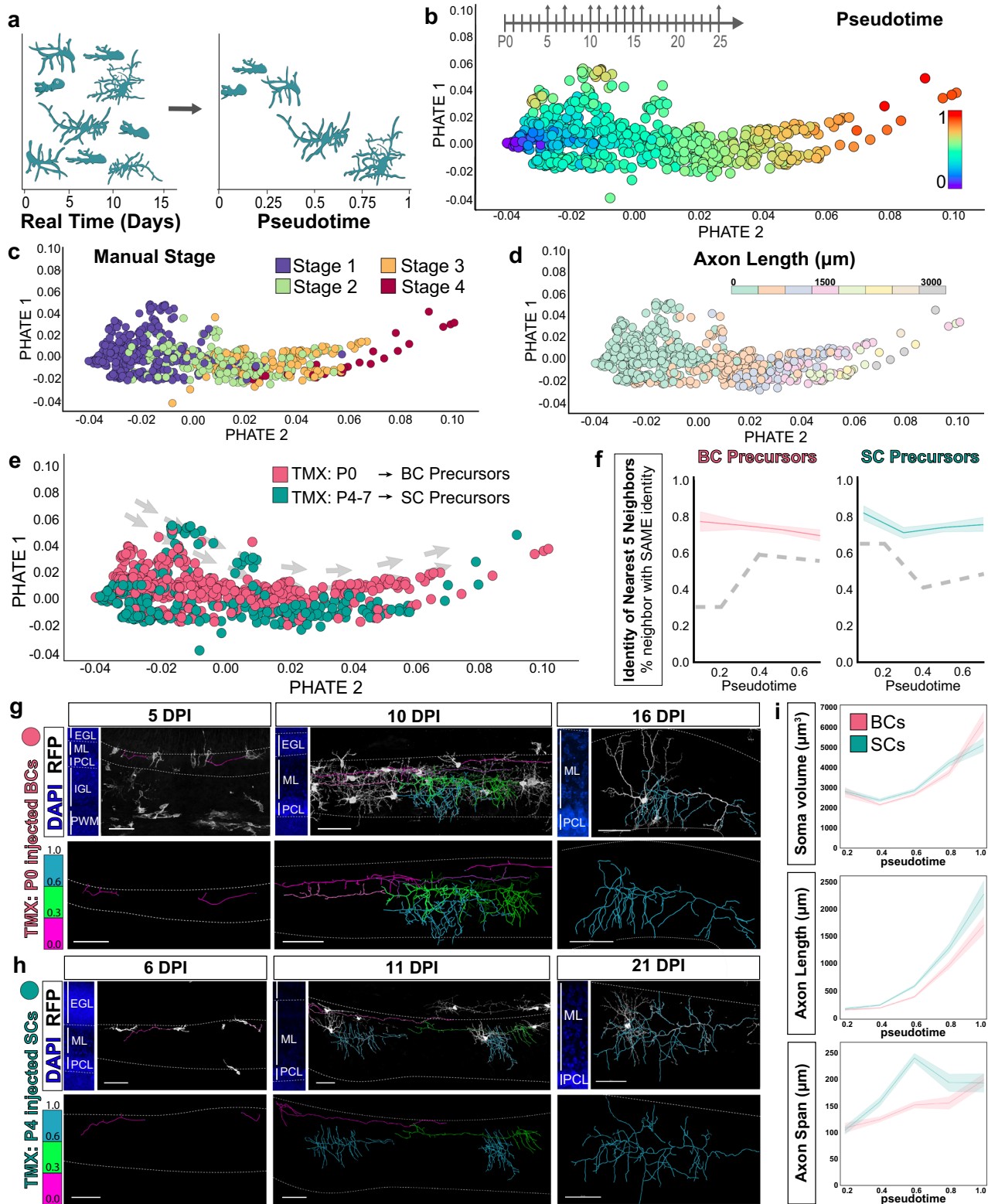

terminal fate (early-born BC or SCs: P0 TMX injected, 423 cells; late-born SCs: P4-7 TMX injected, 309 cells) and age (days post TMX injection, hereafter DPI; Fig. 7b).

One challenge to quantifying developmental changes at a large scale is the variability in the progression of cells present in the tissue at any given time point. To address this confounding feature of development, we adapted a pseudo-temporal ordering approach to align snapshots of single-neuron morphologies over

the course of maturation (Fig. 8a). Pseudotime trajectory inference algorithms are routinely applied to study cell lineages and differentiation from single-cell transcriptomics data, where cells dissociated from developing tissues also contain a spectrum of cellular states[5,6,65,66]. By computationally aligning single-cell data along a continuous trajectory of maturation, pseudotime identifies the progression and bifurcation of cellular states[5]. We reasoned that pseudotime modeling could be applied to high

**Fig. 8 PHATE trajectory inference reveals divergent m-type MLI identities during axonogenesis. a** Schematic depicting pseudotime trajectory inference. Ordering cells and their maturation by animal age (left) is complicated by the developmental variability within any given time point. Pseudotime orders snapshots of developing neurons by morphological maturity (right). **b** PHATE-generated pseudotime ordering of 732 developing MLI morphologies. Each data point represents the developmental morphometry of one MLI reconstruction and color-coded by pseudotime stage. **c** Validation of PHATE ordering by projecting expert-directed maturity for each cell. **d** Projection of total axonal lengths onto PHATE manifold. **e** Projection of inferred MLI identity based on TMX-induced lineage tracing: BC, early-born identity (P0 TMX; $n = 423$ cells; pink) or SC, late-born identity (P4–P7 TMX; $n = 309$ cells; teal) from $N = 32$ mice. **f** Nearest 5 neighbors were obtained for each cell in the PHATE coordinate space. Proportion of neighbors with the same identity were plotted for BC (left) and SC (right) m-type identities. Line plot shows mean with shaded lines for SEM. Dashed gray line represents the null distribution within each pseudotime bin. For pseudotime 0–0.2: $p < 0.0001$ (BCs) and $p < 0.05$ (SCs); 0.2–0.4: $p < 0.01$ (BCs) and $p < 0.001$ (SCs); 0.4–0.6: $p < 0.05$ (BCs) and $p < 0.0001$ (SCs); 0.6–0.8: $p < 0.001$ (BCs) and $p < 0.0001$ (SCs) based on one-way t-test. **g** Confocal images show progression of axon morphogenesis of P0-injected RFP-labeled MLIs (greyscale) at 5, 10, 16 days post injection (DPI). Cerebellar layers are marked by DAPI (blue, left). Axon arbor reconstructions are colorized based on the cell's PHATE pseudotime maturity (magenta cells, 0–0.29; green cells, 0.3–0.59; teal cells, 0.6+). **h** Pseudotime annotation of late-born MLIs, similar to **g**. **i** Line plots of individual morphometric measurements along pseudotime show differences in soma volume, axon length and horizontal axon span between BC- and SC-fated populations (mean ± 95% CI). Scale bars are 50 μm. EGL external granule layer, PWM prospective white matter.

dimensional morphometric data to infer the trajectories of MLI subtype development and to visualize transitions. To assess the developmental progressions of the dataset, we manually annotated each cell into one of four maturation stages according to a qualitative assessment of dendritic arborization and soma shape (Supplementary Fig. S4). This scheme served as an expert-directed ordering to validate the performance of pseudotime algorithms in aligning axonal reconstructions along a maturation trajectory.

**Trajectory inference by PHATE and Palantir reveals emergence of BC and SC m-types.** We asked whether trajectory inference algorithms could distinguish differences in axon morphogenesis between BC- and SC-fated MLIs. We first applied PHATE to analyze the entire dataset of 732 developmental MLI reconstructions. PHATE was developed for visualization of branching data structures, with preservation of both local and global similarities[67]. The PHATE-generated trajectory ordered the P0 and P4-injected MLI reconstructions along a linear pseudo-temporal timeline (hereafter pseudo-timeline; Fig. 8b). The PHATE ordering did not simply reflect similarities in anatomical locations (Supplementary Fig. S5). Rather, the ordering of cells followed a developmental progression, as confirmed by our expert-directed staging metric showing that the stage 1 immature cells were plotted at the beginning of pseudotime and stage 4 cells were located at the end (Fig. 8c). Cells along the PHATE trajectory showed increasing total axonal lengths, consistent with an advancing maturation state (Fig. 8d). Projection of early- and late-born MLIs revealed a segregation between the two labeled populations along the trajectory (Fig. 8e). To quantify this observation, we performed nearest neighbor analysis to test if developing MLIs are likely to reside near cells within the same birthdate cohort due to similarities in their morphological phenotypes (Fig. 8f). We reasoned that if the MLI precursors have yet to commit to BC- or SC-specific differentiation during early stages of axonogenesis, the identity of a cell's nearest neighbor should follow a random distribution. Alternatively, deviation from the null hypothesis infers an early bias of MLI m-type identities. By the earliest pseudotime stage, there is a significant co-sorting of P0-injected BC-fated cells with 77.3% (± 5.4%) of nearest neighbors belonging to the same early-born cohort, and P4-injected SCs with 82.0% (± 3.9%) of nearest neighbors that are SC-fated (Fig. 8f). Thus, pseudotime inference suggests that BC- and SC-fated subtypes can be distinguished by differences in their morphological signatures.

We next sought to map the pseudotime trajectory onto real time in situ. We projected pseudotime values onto morphological reconstructions of their corresponding cell within the confocal images. As expected, each image contained a mixture of labeled MLIs at different stages of maturation and axonal progression (Fig. 8g, h, similar to the schematic shown in 8a). For both P0- and P4-injected MLIs, cells analyzed at 5-6 DPI were ordered at the beginning of pseudotime, while cells at 16–21 DPI cells were assigned late pseudotime values (Fig. 8g, h; magenta and cyan axonal traces, respectively). At intermediate time points, cells were stratified according to the inside-out layering of the molecular layer such that early pseudotime cells were immature and located in the superficial molecular layer, and advanced pseudotime values tracked with cells in lower positions and with increasing axonal complexity. MLIs at early pseudotime stages were *Pvalb*-negative and located in the upper molecular layer (Fig. 8g, h and Supplementary Fig. S6), consistent with tangentially migrating MLI precursors[62,63,68]. They bore a single process that elaborates branches at later stages (compare magenta and green traces in Fig. 8g, h), suggesting that MLIs extend axons during migration before reaching their final positions. We compared individual morphometric quantifications across the pseudo-timeline and found that presumptive BCs and SCs exhibit progressive axonal arbor growth but bear differences in certain features (Fig. 8i). The late-born SC population progressed with increasing total axonal lengths but exhibited a distinct expansion then retraction in axonal span (Fig. 8i). Thus, the early/BC- and late/SC-fated MLI precursors express divergent morphological phenotypes, prior to settling at their final locations, and suggest that BC and SC identities diverge during early phases of migration.

A potential limitation of the pseudotime application could arise from axonal pruning, where decreases in several input parameters could weaken the discrimination of maturing cells and confound the pseudotemporal trajectory. To identify regressive trends and other indications of pruning of axonal arbors, we compared BC and SC measurements during development and at maturity. We examined trends across the expert-directed maturation bins that were ordered independently of axonal information, and found that developing BCs and SCs progress in total axonal length and branch complexity (Supplementary Fig. S7). These axonal features were smaller compared to maturity, suggesting that BC and SC axonal arborizations increase through later stages of maturation, as local branching continues[69,70]. A notable exception is that axonal spans for both populations at maturity remain at a plateau (Supplementary Fig. S7). The axonal features of cells ordered by pseudotime also showed similar progressive trends (Supplementary Fig. S7 and Fig. 8g-i). Although we could not detect obvious patterns of large-scale axonal pruning in terms of declining axonal features at maturity, these analyses are limited to coarse measures of arbor size. Thus, there is the possibility of

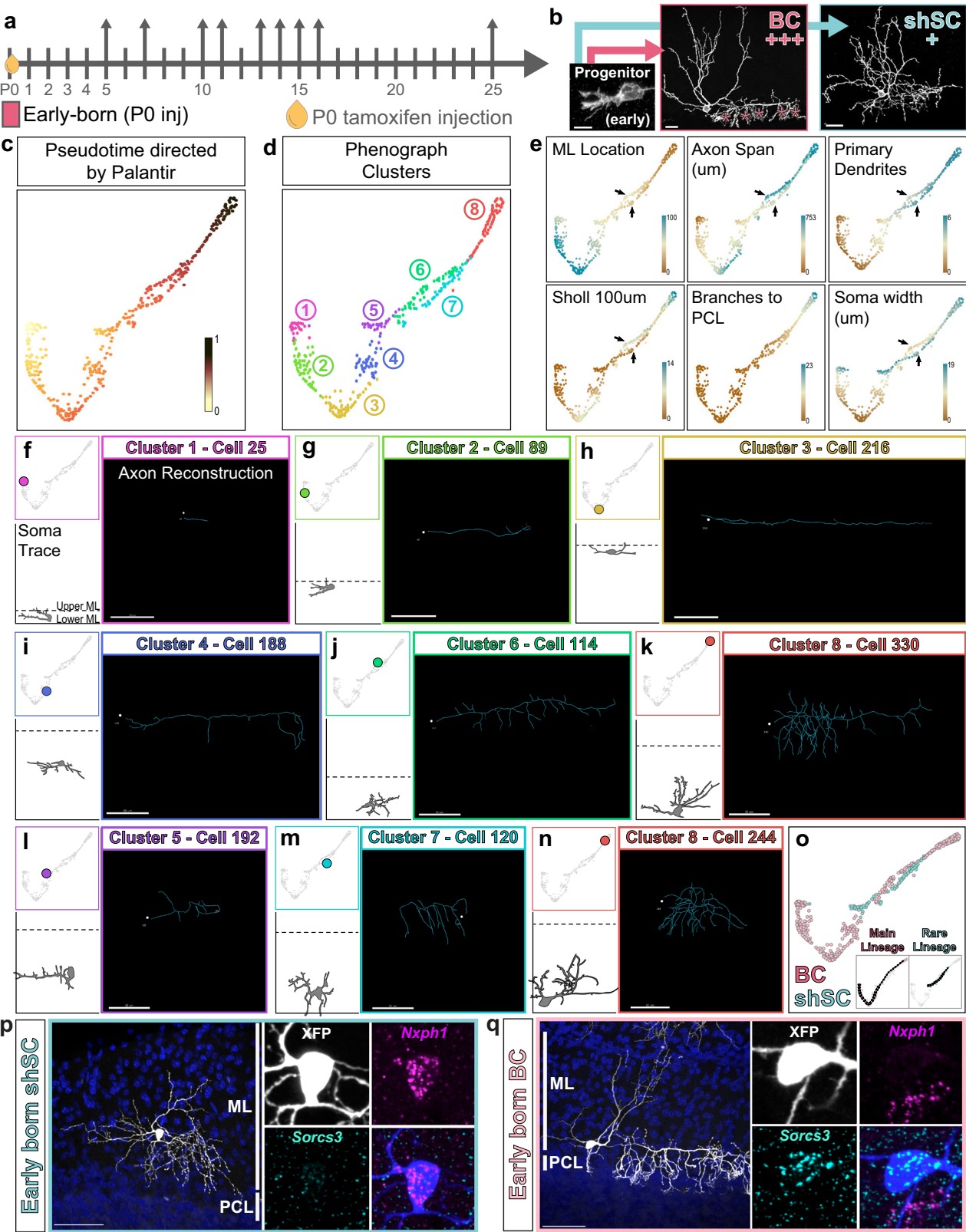

some degree of axon refinement or iterative branch addition-retraction at late stages that are not quantified in our dataset. Nevertheless, our findings demonstrate that BC and SC axon phenotypes segregate prior to arbor maturation.

Given the strength of pseudotime applications for identification of rare cell states or lineages[5,71,72], we next asked if morphological pseudotime can similarly identify rare short-range

SCs from the BCs, which are both labeled by TMX at P0-P1 (Figs. 9a, b and 7f). We first applied PHATE to analyze the early-born MLI-labeled cohort (423 cells) but this method did not yield separate trajectories. Since PHATE was developed for enhanced denoising to remove spurious edges[67], we reasoned that the rare early-born SCs may be inadvertently merged as noise. We next selected Palantir, a pseudotime algorithm with demonstrated

**Fig. 9 Pseudotime ordering indicates divergent early-born MLI populations. a** Scheme for the TMX induction (P0, oil droplet) and collection (arrows) of early-born MLIs. **b** Summary for the mature morphology and identities of P0 TMX labeled MLIs, based on Ascl1-CreER fate mapping in Fig. 7. Most P0-labeled MLIs mature into m-type BCs (+++) with smaller numbers of lower ML short axonal range SCs (shSCs, +). **c** Palantir-generated pseudotime ordering of early-born MLIs (n = 423 cells). **d** Phenograph-generated division of the axonal dataset into eight clusters that reflect different morphological stages. **e** Heatmap representation of single morphometric parameters corresponding to Palantir-ordered MLIs. Arrows highlight differences between the BC and early-born SC trajectories. BCs occupy higher positions within the ML during migration (ML location) with axons that span a greater distance (axon span and Sholl at 100 μm). **f–n** Examples of single MLI morphologies plotted within each cluster 1–8. Top left, position of cell in pseudotime trajectory. Bottom left, Camera Lucida illustration of cell soma and dendrites, positioned to scale by ML location, to validate progressive maturation of cells. Right, Axonal reconstruction with soma outlined by white dot. Representative examples of the BC morphogenesis trajectory are shown in **f–k**. Representative examples of the short-range SC morphogenesis trajectory are shown in **l–n**. **o** Inferred pseudotime trajectories for early-born MLIs. BCs form the main lineage, highlighted in pink, while short-range early-born SCs form a rare lineage, teal. **p** morFISH co-staining of P0-induced MLIs in Ascl1-CreER; Ai14-TdTomato mice confirm labeling of short-range *Nxph1*+ SCs in the lower ML (magenta), and **q** *Sorcs3*+ basket cells (cyan). n = 3 cells (**p**) and 16 cells (**q**) from four animals. Scale bars are 50 μm.

utility and resolution for identifying rare cellular lineages and differentiation trajectories from transcriptional datasets[65]. Palantir rendered a trajectory that followed a linear progression, as confirmed through the default (Fig. 9c) and expert-directed maturation staging (Supplementary Fig. S8). To evaluate the accuracy of the manifold, we clustered the trajectory into eight developmental states using PhenoGraph (Fig. 9d)[73]. Inspection of the clusters along the trajectory revealed a robust ordering of BC axon morphogenesis (Fig. 9e–k and Supplementary Fig. S9). The earliest stage is represented by cluster 1, where cells display the simplest morphologies consisting of a single, short axonal extension (Fig. 9d, f). By contrast, the end of the axonogenesis is marked by cluster 8 that contains cells with complex axonal arbors and PC-targeting collaterals (Fig. 9k). Cells in the intervening clusters display intermediate morphologies that support the idea that MLI axonogenesis proceeds during migration. Immature BCs in clusters 2 and 3 elaborate a tangentially oriented trailing process along the superficial ML (Fig. 9g, h). As BCs mature in clusters 4, 6, and 8, they occupy lower laminar positions and arborize axons (Fig. 9i–k and Supplementary Fig. S9). Thus, pseudotime reconstructed the progression of BC axon morphogenesis from the large set of snapshots of developing MLIs.

The pseudotime trajectory also indicated a separate lineage of cells in clusters 5 and 7, which exhibited differences such as shorter axon processes and arbor span and reduced branch complexity compared to developing BCs in clusters 3, 6 (Fig. 9d, l–n). These features are consistent with the short axon SCs, and confirm the presence of divergent phenotypes within early-born MLIs (Fig. 9o). To further test whether early-born MLIs give rise to distinct BC and SC subtypes, we performed morFISH for *Sorcs3* and *Nxph1* on P0-induced MLIs. Indeed, early-born populations give rise to *Sorcs3*+ BCs and *Nxph1*+ short-range, non-soma-targeting SCs (Fig. 9p, q). Therefore, BCs and lower SCs residing in the lower molecular layer are born in the same temporal cohort but exhibit distinct axonal phenotypes. Together, these findings suggest an overlapping, rather than sequential, emergence of BC/SC identities.

In summary, our results provide a developmental map of cerebellar MLI diversification through the lens of morphology, and identify new molecular and spatial correlates of BC and SC subtypes. This study also demonstrates the power of pseudotime for modeling morphogenesis over weeks of development and for resolving morphological trends and rare neuronal subtypes.

## Discussion

Over 130 years ago, Santiago Ramón y Cajal took advantage of the intriguingly heterogeneous and accessible organization of the cerebellar MLIs to propose and substantiate the neuron doctrine[31,74–76]. In homage to the original master, we re-examined MLI diversity using genetic and computational methods of modern single-cell biology. We analyzed mature and developing MLIs based on morphology, lineage, and spatial and molecular correlates. MLIs divide into two discrete groups when either morphology and gene expression signatures are considered, but the cell type divisions do not align with each other. Morphologically, MLIs comprise discrete basket and stellate cell subtypes, with continuous variation observed among SCs. By applying clustering and pseudotemporal ordering techniques to morphometric data, we demonstrate that the axonal arbor is the defining structure for BC-SC classification and emergence of type-specific phenotypes. Although MLI m-types are discordant from t-types[45], the BCs and longer axon SCs can be identified by a set of continuously varying markers belonging to the *Sorcs3*+ t-type, while SCs with shorter axons are enriched for *Nxph1*. The molecular profiles are not expressed until late stages of MLI maturation, indicating that these t-type identities are limited to mature MLIs. By contrast, morphology is a critical modality for tracking MLI diversification. Through novel pseudotime applications that robustly reconstructed and quantified MLI morphogenesis, we detected an early emergence of BC and SC subtypes based on divergent axonal phenotypes. Early-born MLIs give rise to *Sorcs3*+ BCs and non-basket forming *Nxph1*+ SCs that settle in similar positions in the lower molecular layer. In contrast to the idea that BC and SC m-types are generated sequentially and differentiate at final laminar positions[38,77], our findings support a model of BC/SC subpopulations that arise from early-born MLIs and express divergent axonal development during their migration. This study also provides a proof-of-concept for integrating pseudotime morphology as a modality for studying neuronal diversification.

**A revised taxonomy of MLIs.** The classical basket and stellate cell division is based on canonical features displayed by deep BCs and superficial SCs[30,31,78]. However, most MLIs show intermediate BC-SC morphologies that have led many, including Cajal, to propose that MLIs form one continuous population[18,33–36]. Our results reconcile both sides of the debate by establishing a discrete BC-SC division and continuous variation present within the SC group. Our morphological dataset sampled the entire MLI population across molecular layer positions, as additionally confirmed by *Nxph1* and *Sorcs3* co-labeling which together labeled all MLIs. Iterative removal of morphological parameters identified the axonal arbor as the defining feature for BC-SC classification. Although basket formations are a canonical BC feature, this parameter alone does not predict BC versus SC identity, as shown by the examples of SCs with PC-soma targeting collaterals and the leave-one-out clustering analyses. Pinceaux terminals on the other hand, could be exclusive to BCs but these structures should be confirmed with molecular markers[79], and were thus excluded from the clustering analyses. Laminar location is also dispensable

for sub-clustering MLIs, indicating that BC/SC morphological identities do not simply reflect a dependence on laminar position within the molecular layer.

The SC class exhibits graded morphologies that were previously attributed to the entire MLI population[18,35,36]. A subset of parameters describing SC dendrites and axonal span co-varied to a modest degree with molecular layer position, consistent with previous observations of continuous variation in dendritic and axonal structures[33,34,36]. Although we cannot exclude the possibility that SCs divide further into discrete subtypes with significantly increased sampling, our statistical analyses support a continuous model for SCs. Reiterative subsampling and clustering of SCs was error-prone, likely due to intermediate phenotypes and graded individual morphometric measures. Moreover, morphometric analysis by PAGA, which is commonly applied to transcriptomic datasets to measure how clusters occupy overlapping gene expression space[12,45,50], showed connectivity between short and long axonal SC nodes. Axonal variation among SCs produces a remarkable degree of local diversity with important implications for connectivity patterns and local network properties, such as electrical coupling and feedforward connectivity[35,80]. Thus, the SC continuum adds a morphological example of continuous variability. Continuous heterogeneity has emerged as an important feature of neuronal diversity, but mainly from gene expression studies[2,12,13,28,81–84].

**Morphologically and transcriptionally defined MLIs are discordant**. It is puzzling that classically and morphologically defined MLIs do not align with the recently described MLI1/*Sorcs3*+ and MLI2/*Nxph1*+ t-types from single-cell transcriptomics of mature cerebella[45]. A limited labeling of cells from either group showed no obvious correspondence to canonical BC or SC features[45]. We confirmed the discordance between m-types and t-types by morFISH but discerned the following relationships. First, BCs are exclusively represented in the MLI1/*Sorcs3*+ group and express high levels of *Grm8*, and thus marked by *Sorcs3*+/*Grm8*$^{HIGH}$/*Cacna1e*$^{LOW}$ expression. Second, long axonal SCs are also contained within the *Sorcs3*+ group and express high levels of *Cacna1e* with graded *Grm8* expression. Third, *Nxph1* marks shorter axonal range SCs that we found in the lower to upper third-quarter of the molecular layer. *Nxph1* also marks a subset of SCs in the very top molecular layer which we could not morphologically distinguish from nearby *Sorcs3*+ SCs. These results raise the interesting possibility that the early-born BCs and SCs in the lower layer are morphologically and molecularly divergent (BC/MLI1-*Sorcs3*+ and SC/MLI2-*Nxph1*+), while later-born SC/*Sorcs3*+ and SC/*Nxph1*+ form heterogeneous populations. Finally, there is a clear laminar organization within these t-types, as shown by smFISH of transcripts identified from this dataset and quantitative smFISH of *Grm8* and *Cacna1e*. In the previous report[45], most electrical properties subdivided along MLI1 or MLI2, but one electrical feature varied with laminar position. A laminar organization likely shapes the morphological and functional connectivity of MLIs, as recruitment motifs, inhibitory outputs, and other synaptic properties differ with molecular layer position[35,80,85,86].

The discordance between MLI m-types and t-types reflects the emerging challenge of reconciling neuronal subtypes across different modalities. For instance in hippocampal CA1, there is a striking lack of correlation between morphological, transcriptomic, and electrophysiological profiles of GABAergic *Pvalb*-expressing interneurons[14]. Although *Pvalb* interneurons parse into distinct morphological subtypes, single-cell transcriptional profiling revealed one continuous entity with some individual markers correlating with morphology. The *Pvalb* morphological

subtypes also share a homogeneous biophysical signature[14]. Similarly in the cortex, anatomically distinct cortico-cortical projection neurons were defined transcriptomically by a single, continuous cluster[11]. Increasing the scale of single-cell profiling might provide comprehensive descriptions of neuronal subtypes that align across modalities. However, large-scale cross-modal profiling of cells in the motor and visual cortex in mice show good correspondence between t-, m-, and electrical-types for major class divisions, but the separations are less defined for subclasses and there is continuous variation along some modalities[13,15,87,88]. Likewise, large-scale morphological and transcriptomic profiling of projection neurons reveals alignment at the level of major types, but correlations are limited when comparing subtypes[89]. A final important consideration is that morphological and connectivity features are established during development. Morphologically defined subtypes, such as the BC/SCs, might be uncoupled from other cell type divisions because the developmental programs that shape them are not represented by the transcriptional signatures at maturity.

**Pseudotime modeling of morphogenesis reveals BC and SC phenotypic divergence**. By extending unbiased, quantitative morphological analyses to development, we defined the developmental stages and phenotypes relevant to subtype divergence. MLIs are the last category of interneurons generated from the GABAergic lineage and derive from a progenitor pool residing in the prospective white matter (PWM)[40,57–60]. Birthdating and transplantation studies have led to the prevailing idea that the BC and SC fate choice is not intrinsically determined nor restricted at birth[37–39,90]. Rather, it has been suggested that BC/SC fates are instructed by extrinsic cues related to sequential birth order and laminar position[37–39,90], but tracking their differentiation has been elusive due to the lack of methods to distinguish them.

Reconstruction of axon morphogenesis of birthdate-labeled MLIs by pseudotime algorithms revealed BC/SC divisions prior to arrival at final locations and t-type marker expression. The 'pseudo-dynamic' analysis ordered the morphometric dataset spanning two weeks of development along a progression that matched visual validations, and detected developmental differences that are subtle to the eye. To our knowledge, this is the first demonstration of pseudotime algorithms applied to morphological information obtained from neurons, and fills an important gap for quantifying neuronal morphogenesis on the scale of days to weeks. The early segregation of BC and SC phenotypes rendered by PHATE occurs during migration, as indicated by the pseudotime staging and related images showing MLIs located in the upper layer with elongated soma and simple dendritic processes that are characteristics of tangential migration[63,91]. We also noted long simple axons, which resemble processes that extend from migrating cortical interneurons and form type-specific axonal morphologies[92,93]. The clustering is not attributed to temporal cohort effects (i.e. batch effects related to injection or collection times), nor due to canonical axon structures such as basket formations as these appeared at later stages and were not prominently represented in the developmental dataset. The segregation of BC and SC phenotypes during migration suggests that BC and later-born SC differentiation might be influenced by cues that differ with temporally changing microenvironments and/or with laminar location, as the molecular layer expands. In this model, the phenotypic specification of BC/SC during migration does not exclude the possibility that MLIs retain plasticity until they arrive at their final locations and express mature t-type markers. Identifying cues that determine MLI subtype specification versus commitment will be important for understanding how interneuron

heterogeneity deriving from a common progenitor pool is established.

The second notable finding is that BC and SC axonal arborization is largely progressive. MLI axonal lengths and complexity increased over the manual and pseudotime staging and were greatest at maturity, consistent with the increased BC axon branching around the PC soma observed later in development[70]. We did not detect obvious large-scale axonal refinement that would have been marked by expansions then retractions in a number of axonal features, as recently shown for cortical chandelier interneuron arbors[94,95]. We did detect retraction of axonal spans of SCs, indicating that pseudotime ordering can account for regressive events in single or subsets of parameters, further highlighting the utility of this approach for reconstructing morphogenesis. As the methods are limited to coarse changes in axonal arborization, we cannot discount the possibility that MLIs undergo finer regressive events. BC/SC phenotypic segregation was quantified early in axonogenesis, prior to the late phase of local branching and potential remodeling. Nonetheless, axonal refinement is an important consideration, as multiple regressive parameters could limit the performance of pseudotime algorithms that assume globally linear trajectories[96]. Further investigations are needed to assess the performance for cell types that undergo non-linear growth. Increasing the number of input parameters, adding progressive innervation features such as varicosities or terminal structures, and developing algorithms optimized for morphometric datasets are future directions for strengthening and broadening pseudotime applications for studies of neuronal diversification.

Finally, in using these approaches, we also identified the phenotypic division of early-born SCs that give rise to the *Nxph1+* non-basket SCs. The differentiation of early-born SCs and BCs suggests a model in which m-type specifications are not simply dictated by birthdate order or final locations. However, the timing of SC/BC fate decisions remains unclear. Migrating SCs and BCs can be distinguished morphologically, but the SC-*Nxph1* and BC-*Sorcs3* markers are not detected until MLIs settle and express *Pvalb*. Additional snRNA-Seq profiling of developing cerebella did not produce clues, as the MLI1/MLI2 trajectories diverge late following a shared *Pax2+* path[45]. One possibility is that early-born MLIs are initially equivalent, but a subset is at a disadvantage and fails to form baskets or retracts their soma-targeting processes, producing the non-basket lower SCs. While our data do not exclude this possibility, we observed that PC-soma targeting occurs at late stages of maturation, consistent with previous reports[69]. Basket formation is dependent on secreted Sema3A signaling and Neuropilin1/Neurofascin186 interactions[70], and thus it would be interesting to test whether the Nrp1 signaling is restricted to the BC-fated cells. Since the PWM is hypothesized to be instructive for MLI genesis and differentiation[38,77], another possibility is that divergent early-born BC and SC fates are instructed by cues in the PWM or along their migration. For instance, BCs and lower SCs are born at similar times, but lower SCs might adopt a different fate leading to SC-*Nxph1+* identity by taking longer to traverse the PWM and molecular layer. Further lineage tracing and cross-modal profiling studies are needed for defining MLI specification and relevant signals at these earliest stages, and to inform tests on the mechanism and timing of BC/SC fate decisions.

In conclusion, our study leveraged computational methods to characterize MLI development with single-cell resolution and temporal coverage, while annotating individual cells using anatomical, lineage, and gene expression information. Such approaches will inform future studies for factors that shape neuronal patterning and for linking local variations in connectivity patterns to function. Importantly, restoring anatomy and spatial-temporal contexts to single-cell profiling will provide the integrated descriptions needed for understanding how diverse neuronal subtypes arise and assemble into circuits.

## Methods

**Mouse strains.** Mouse lines used in this study have previously been described. GABAergic neuron targeting *Gad2-ires-Cre* (JAX accession #: 010802) was obtained from Jackson Laboratories[47]. *Ascl1-CreERT2* (JAX #: 012882) was obtained from Jackson Laboratories[97]. Cre reporter lines Ai14 *Rosa-Cag:LSL-TdTomato* (*Rosa^{Ai14:STOP-TdTomato}*, JAX #: 007908) and *Rosa-Actb:membTdTomato-membGFP* (*Rosa^{mTmG}*, JAX #: 007576) were obtained from Jackson Laboratories[98,99]. Mice were maintained on a C57/B6J or mixed C57/B6J and FVB background. All experiments were carried out in accordance with the Canadian Council on Animal Care guidelines for use of animal in research and laboratory animal care under protocols approved by the Centre for Phenogenomics Animal Care Committee (Toronto, Canada) and the Laboratory Animal Services Animal Care Committee at the Hospital for Sick Children (Toronto, Canada).

**AAV viral labeling of neurons.** Recombinant Brainbow AAV9-hEF1a-LoxP-TagBFP-LoxP-eYFP-LoxP-WPRE-hGH-InvBYF and AAV9-hEF1a-LoxP-mCherry-LoxP-mTFP-LoxP-WPRE-hGH-InvCheTF viruses[46] were a gift from Dawen Cai & Joshua Sanes and obtained from the University of Pennsylvania Vector Core (Addgene viral preps # 45185-AAV9; # 45186-AAV9). In total, ~$1 \times 10^{12}$ viral genome particles per mL of each vector was prepared in sterile phosphate-buffered saline (PBS, pH = 7.4). To introduce virus into the cerebellum, P0-14 mouse pups (P0 for PCs, P5 -14 for MLIs) were anesthetized with either ice (P5 or younger) or isoflurane using a rodent anesthesia machine (P6 and older, 4% $O_2$). A 25 gauge needle was used to make a small puncture into the caudal-medial position of the right cortical lobe, and 1.5 μL of rAAV virus was injected into the lateral ventricles with a Hamilton syringe and 33 gauge blunt-ended needle. Injection procedures were repeated for the left cortical lobe for bilateral labeling of neurons. Animals were sacrificed and cerebellums dissected at P75 for mature classifications of MLI morphologies.

**Tamoxifen induced labeling of neurons.** Tamoxifen (Sigma, T5648) was dissolved in corn oil (Sigma, C8267) to a concentration of 10 mg/mL (high dose injections; Fig. 7c, middle and right panels), 0.5 mg/mL (low dose injections for sparse labeling using the *Rosa^{mTmG}* Cre reporter), or (0.05 mg/mL (low dose injections for sparse labeling using *Rosa^{Ai14:TdTomato}* Cre reporter). P0 - P7 pups were put on ice for 1 min for anesthesia and to reduce oil leakage following injections. A 0.5cc insulin syringe (BD) was used to introduce 10–20 μL of tamoxifen solution through intraperitoneal (IP) injections. The needle tip was held inside the pup for 20 s to prevent excessive leakage. IP injections of pups were reproducible for labeling basket cells and early-born stellate cells at P0, P1 and stellate cells at P4, P7, as judged by laminar locations and phenotypes[100]. Pups were sacrificed at least 3 days post-injection.

**Histology.** Mice were either anesthetized by hypothermia (P5 or younger) or under isoflurane (4% in O2 for induction; 1.5–2% in O2 for maintenance), and trans-cardially perfused with physiological saline solution (0.9%; Baxter) followed by 4% paraformaldehyde (PFA) in PBS. Animals were perfused by the gravity perfusion method. Brains were post-fixed in 4% PFA overnight at 4 °C or for 3.5 h at room temperature (RT).

100 μm sagittal sections of the vermis cerebellum were sectioned using a vibratome (Leica). Sections were incubated for 3.5 h in blocking buffer (0.5% Triton-X, 4% normal donkey serum in PBS), and incubated for 72 h at 4 °C with primary antibodies in a 24-well plate on a rocker. Following 3 × 15 min PBS-T (0.5% Triton-X) washes, sections were incubated for 3.5 h at RT with Alexa-conjugated secondary antibodies. Sections were mounted onto glass slides using Fluoromount G (Southern Biotech). Nuclei were labeled using DAPI or NeuroTrace Nissl 435/455 (Invitrogen). Primary antibodies used for this study were as follows: chicken anti-GFP (1:2000, Aves Laboratories, GFP-1010); rabbit anti-mCherry (1:500, Kerafast, EMU106); rat anti-TFP (1:500, Kerafast, EMU104), guinea pig anti-TagBFP (1:500, Kerafast, EMU105); Rabbit anti-RFP (1:1000, Rockland, 600-401-379); goat anti-Parvalbumin (1:1000, Swant, PVG213), and rabbit anti-Calbindin (1:1000, Sigma, C9848). All secondary antibodies were used at a dilution of 1:1000 and were as follows: Donkey anti-goat AF647 (Life Tech, A21447); Donkey anti-goat AF488 (Life Tech, A11055); Donkey anti-rabbit AF488 (Jackson Immuno, 711-545-152); Donkey anti-rabbit AF568 (Life Tech, A10042); Donkey anti-rabbit AF647 (Jackson Immuno, 711-605-152); Donkey anti-chicken AF488 (Jackson Immuno, 703-545-155); Donkey anti-guinea pig AF647 (Jackson Immuno, 706-605-148); Donkey anti-rat AF647 (Jackson Immuno, 712-605-153).

**Confocal imaging for neuronal morphology.** Images of single MLIs were taken on a Leica SP8 scanning confocal microscope, using a ×40 oil objective (NA = 1.30). Z-stacks were collected with a 0.5 μm (mature dataset) or 1 μm (developmental dataset) step size throughout the depth of the cells including the entire dendritic and axonal arbor. Only neurons where arbors were not cut off by the vibratome

were imaged and analyzed. To optimize the consistency of the morphological comparisons, we selected neurons in the cerebellar vermis to capture cells in the optimal sagittal orientation and with complete dendritic and axonal arborizations within the 100 μm slice. The Z-compensation feature was used to avoid signal saturation throughout the depth of the tissue. Images were acquired with slightly different XY pixel sizes to accommodate varying arbor sizes, and to be able to capture single MLIs within one field of view. XY pixel sizes used were maintained at ~135 nm.

**MLI counts.** To account for the percent laminar distribution of singly labeled MLIs following Brainbow injections, we divided the ML into four strata, and counted the number of labeled MLIs within each strata. The XScope app (iconfactory) was used to create a 1 × 4 grid, which was resized to match the area of the ML. The number of MLI were then counted within each strata, using the cell counter plugin in FIJI[101]. Each counting frame additionally consisted of one inclusion and one exclusion edge, and cells were counted if found entirely within the counting frame or overlapping with the inclusion edge but not the exclusion edge.

**Manual curation of MLI morphologies.** Canonical BCs and SCs were identified according to standards in literature[36,48,53,102]. Canonical BCs were identified by: (1) soma location within the bottom 1/3 of the ML, (2) a long horizontal axonal shaft that gives rise to basket terminals, and (3) a dendritic arbor that reaches the superficial ML. Canonical SCs were identified by: (1) soma location within the upper 1/2 of the ML, (2) absence of basket terminals, and (3) radially oriented dendritic and axonal arbors. The manually curated set of BCs likely contained fewer cells than both hierarchical clustering and UMAP analyses due to our strict laminar location cut-off.

**Morphological reconstructions.** In all, 3D reconstructions for single-neuron morphologies were completed in Imaris (Bitplane; v. 8.2–9.5). Dendritic and axonal arbors were semi-automatically reconstructed using the filament tracer (autodepth mode). Surface render of the cell soma was performed using the surfaces module. We note that due to the membrane-targeted nature of the Brainbow fluorophores, the soma volume data for mature MLIs more accurately describes the volume of the cell membrane, and not the entire soma.

**Data compilation of single-neuron anatomical features.** For each neuron, we: (1) registered the spatial coordinates to note the cell location and folia of origin in the vermis cerebellum; (2) performed surface rendering for the cell soma, (3) performed morphological reconstructions, and (4) extracted 27 quantitative features to describe the dendrites, axon, soma, and location (Supplementary Table 1). 19 out of 27 features were selected for final clustering analyses to avoid bias from duplicated features (i.e. a single weighted basket measurement was compiled, to avoid the introduction of bias from 4 features that measure axonal baskets; see Supplementary Table 1). The experimenter was blinded to the injection time point while compiling and scoring morphological features.

Basic morphological features were exported from Imaris using the statistics tab. These include: axon/dendrite length, number of Sholl intersections at 10, 50, 100, 150, and 200 μm, mean/max branch level, axon straightness, and volume of somatic render. The remaining features were manually compiled:

*Mature dataset.*

(1) The number of filopodia was calculated as the number of terminal branches under 1.5 μm in length.
(2) Filopodia density was calculated by normalizing total filopodia numbers to the dendrite length.
(3) Z-depth, height of ML covered by dendrites/axons, and axonal span along the ML were measured using the measurement tool in Imaris.
(4) For analyses of upwards- and downwards-oriented axon collaterals (numbers, length, and percentage), collaterals directed towards the superficial or deep ML were highlighted in Imaris, and the corresponding data were compiled from the statistics tab. For collaterals that were accurately angled, only branches >30° away from the main axon were included in the analysis.
(5) The presence of axon-carrying dendrites was noted for cells where the axon initial segment originates from the basal dendrites, not the soma.
(6) Folial location was noted in folia 1, 3, 5, 6, 7, 8, 9, and 10. This data was not included in the final clustering analyses, but used for confirmation that MLI identities are not biased by the cell's folia of origin.
(7) Relative molecular layer (ML) position was measured using the measurement tool within slice mode. ML heights were calculated as the distance from the top of the PCL to the top of the ML. MLI soma heights were calculated as the distance from the top of the PCL to the center of the MLI soma. MLI laminar locations were calculated as MLI soma height/ML height.
(8) For each cell, the number of PC soma-targeting axon terminals were categorized and quantified according to a weighted scale: (i) full baskets that fully enwrap the PC soma, and form pinceau structures along the axon

initial segment were given a weight of 1; (ii) 'basket-like' terminals that enwrap PCs and reach the base of the soma or axon initial segment but without pinceau formations were given a weight of 0.75; (iii) soma-targeting terminals that partially envelop the PC soma were given a weight of 0.5.
(9) The number of primary dendrites were counted as the number of branches which arises from the soma.

*Developmental dataset.*

(1) Overlapping parameters were compiled following similar protocols as the mature dataset. These include axonal height, depth and span information, measurements of upwards- and downwards-oriented axon collaterals, laminar locations, and the number of primary dendrites.
(2) The number of axonal branches which reached into the PCL were counted, to account for developmental changes as most basket-forming cells have not formed full basket formations at the time-points analyzed.
(3) Soma diameter for each cell was measured along three dimensions using the measurement tool within slice mode (X = direction along the PCL, Y = direction perpendicular to the PCL, and Z = direction along parallel fibers). The final somatic volume was approximated using the ellipsoid volume equation: $V = 4/3\pi \times X\text{-diameter} \times Y\text{-diameter} \times Z\text{-diameter}$.

We note that although a full 3D volumetric render of the soma similar to what was done for the mature dataset would have been more accurate, the process was time and computationally intensive. Approximating the soma volume allowed for the large-scale nature of our study.

**Clustering analyses.** Prior to morphometric clustering, each dataset was zero-mean standardized as previously described[103]. t-SNE was performed with the Rtsne package (version 0.15) for the R statistical environment (R Foundation for Statistical Computing 2018; version 3.5.1) with the following parameters: seed = 510, theta = 0, max_iter = 10,000, and perplexity = 7. The elbow method and silhouette analysis were performed using the NBClust package (version 3.0) in R. UMAP analysis was performed in Python (Python Software Foundation n.d.; version 3.7.3; jupyter notebook: v. 6.0.3) using the umap (v.0.3.9) and sklearn packages (v.0.22.2), with the following parameters: min_neighbors = 7, min_dist = 0.1, metric = 'euclidean'. Hierarchical clustering was performed using the SciPy.cluster.hierarchy package in Python, using Ward's method[104]. PAGA was performed in Python using the following parameters: svd_solver = 'arpack', n_neighbors = 4, n_pcs = 10, clustering resolution = 0.5, PAGA threshold = 0.1.

**Cross validation using iterative reclustering.** For iterative hierarchical clustering using resampled MLIs, the random.sample() function was used in Python to randomly select 5, 10, 15, 20, 25, 30, 35, 40, 45, 50, 55, 60, 65, or 70 cells from the dataset of 79 MLIs. Each sample size was resampled 20 times. To identify the accuracy of the resulting classification, the resulting hierarchical clustering graphs were compared to the organization of the original graph of 79 MLIs. The BC and SC clades were identified within the resulting graph by identifying the branch where most cells were previously identified as BCs or SCs, respectively. The number of resampled cells that were correctly classified into the BC or SC categories was divided by the total number of resampled cells to obtain the accuracy of the classification. Similarly, the short- and long-axonal range SC subclades were identified and compared to the original hierarchical clustering graph.

**Hierarchical clustering and PAGA ranking.** To rank cells by hierarchical clustering (HC), the cell within the BC clade at the leftmost side of the HC tree was ranked as 1, the second cell was ranked as 2, and so on. To rank cells within the PAGA graph, previous protocols were followed[12,51]. Briefly, the diffusion pseudotime time (DPT) value was obtained for each cell and ranked from the smallest to the largest value.

**snRNA-Seq data extraction, normalization, and clustering analyses.** The raw snRNA-Seq data of 780,553 cerebellar nuclei were downloaded as .h5ad files from the Broad institute Single Cell Portal[51]. To extract MLIs, the total dataset was processed and clustered to reveal all cerebellar cell-types. The subsetting function was then used to extract the two MLI clusters which were enriched for *Gad2+/Pvalb+/Prkcd+/Sorcs3+* and *Gad2+/Pvalb+/Prkcd+/Nxph1+*, respectively. Specifically, nuclei with fewer than 200 UMIs, identified as putative doublets, or with high mitochondrial gene expression were discarded. Data preprocessing including UMI normalization, highly variable gene selection, and scaling were performed in Scanpy (version 1.7.2) as described in Wolf et al.[105]. Principal component analysis (PCA) was performed with 40 components followed by Leiden clustering with a resolution of 0.1 to identify major clusters of cerebellar cell-types (resulting in 25 clusters, but containing multiple clusters of granule cells). The raw, unprocessed cell x gene matrix for the two MLI clusters were then extracted and saved as a separate .h5ad file for subsequent analyses. The .h5ad files were loaded into Scanpy for clustering and analysis. Data preprocessing including UMI normalization, highly variable gene selection, and scaling were performed in Scanpy as before. PCA was performed with 15 components followed by Leiden clustering with a

resolution of 0.3. The top differentially expressed transcripts were found using the rank_genes_groups() function in Scanpy by selecting transcripts that were identified by all of the 't-test', 'wilcoxon', and 'logreg' methods.

**Single-molecule fluorescence in situ hybridization**. HCR amplification-based smFISH was performed following previous protocols[54,106,107]. HCR 3.0 probes were purchased from Molecular Instruments for *Sorcs3, Nxph1, Grm8, Cdh22,* and *Ptprt.* The remaining probes were designed in house with the following parameters: (1) 25nt (HCR 2.0) or 2x 25nt (HCR 3.0) target region, (2) no more than 14nt cross-binding to any off-targets in the mouse transcriptome for each 25nt target region, (3) Tm between 45 and 65 °C, and (4) no more than 4 consecutive bases which are identical. All subsequent steps were performed under RNASe-free considerations and using buffers prepared with DEPC-treated water.

12mm #1.5 coverslips (Fisher Scientific) were cleaned in 1N HCl overnight at RT, followed by 5x washes in milliQ water. 1% bind-silane solution (GE) was prepared in pH 3.5 10% (v/v) acidic ethanol solution, according to manufacturer's instructions. The cleaned coverslips were immersed in the 1% bind-silane solution for 1 h at RT, followed by 3x washes with ethanol. Silanized coverslips were heat-cured on a hot plate at >120 °C for over 30 min Coverslips were then cooled to room temperature and treated with 0.2 mg/mL PDL (sigma) for 8–16 h at RT. PDL- and silane-treated coverslips were then rinsed 3x in milliQ water, air-dried, and stored dry at 4 °C for up to 2 weeks.

Animals for smFISH were perfused and collected following similar protocols as the histology section. Postfixed cerebella were cryopreserved in 30% sucrose until they sank (~1–2 days), embedded and frozen in OCT cryopreservation agent (Tissue-Tek), and sagittal sections were collected at 16 μm onto silane- and PDL-treated coverslips. Tissue sections were allowed to dry on the coverslips for ~2 h at RT, and stored at −80 °C in homemade 3D printed coverslip storage chambers until smFISH was performed.

Prior to commencing smFISH, coverslips containing tissue sections were permeabilized in 70% ethanol overnight at 4 °C. Coverslips were then allowed to dry completely before clearing with 4% SDS at room temperature for 30 min, 3x washes in PBS, and 3x rinses in 70% ethanol. Coverslips were allowed to dry completely before hybridization. Primary probe hybridization was performed at 2 nM probe concentration in HCR hybridization buffer (Molecular Instruments) at 37 °C overnight. 3x washes were performed in wash buffer (Molecular Instruments) at 37 °C, followed by 3x washes in 2X SSCT (0.1% Triton-X) at RT.

4% acrylamide solution was prepared for sample clearing containing 4% (v/v) 19:1 acrylamide/bis-acrylamide, 60nM Tris·HCl pH 8, and 0.3 M NaCl. The acrylamide solution was degassed for 10 min on ice using nitrogen gas. Tissue sections were acclimated to the acrylamide solution for 1–15 min 10% (w/v) ammonium persulfate (APS) was prepared fresh and kept on ice. A 4% hydrogel solution was prepared by mixing the 4% acrylamide solution with 0.05% APS and 0.15% TEMED. Gelling chambers were prepared in a tupperware container containing open 24 well plates containing water to create a humidified chamber. Slides covered by parafilm were placed on top of the 24 well plates. A 15 μL drop of the hydrogel solution was placed on top of the parafilmed slide, followed by the washed coverslip, tissue side down. The gelling chamber is degassed with nitrogen gas for 10 min, placed in the 4 °C for 30 min before transferring to a 37 °C incubator for 2 h for gel polymerization.

Clearing buffer was prepared by mixing 1:100 proteinase K, 50 mM Tris·HCl pH 8, 1 mM EDTA, 0.5% Triton-X, 500 nM NaCl, and 1% SDS. Following gellation, coverslips were gently lifted from the gelling chamber and washed in 2X SSCT. SSCT was then replaced by the clearing buffer, and samples were allowed to clear overnight in a humidified chamber at 37 °C. After clearing, samples were washed 3x in 2X SSCT prior to HCR amplification.

HCR hairpins were prepared by heating to 95 °C for 90 s, and allowed to cool to room temperature slowly over 30 min to form metastable hairpins. Amplification solutions were prepared by mixing 2 μL of H1 and 2 μL of H2 hairpin in 100 μL of amplification buffer for each sample (Molecular Instruments). Washed samples were pre-amplified in 100 μL of amplification buffer for at least 10 min, and amplified in 100 μL of hairpin solution overnight at RT in the dark. Following amplification, samples were washed 3 x in 2X SSC, stained with 1:3000 DAPI (ThermoFisher) for 10 min at RT, and mounted in Fluoromount G.

**Confocal imaging of smFISH**. smFISH images were acquired on a Leica SP8 scanning confocal microscope with integrated Lightening deconvolution (Leica), a white light laser capable of super-continuous excitation at any user-defined wavelength between 470–670 nm, and using a 40X oil objective (NA = 1.30). Z-stacks were collected with a 0.33 μm step size throughout the depth of the tissue, and following nyquist requirements for lateral sampling. Five-color smFISH images were acquired using DAPI, Alexa 488 (excitation: 496 nm; emission: 502–546 nm), Alexa 546 (excitation: 550 nm; emission: 560–580 nm), Alexa 594 (excitation: 600 nm; emission: 606–663 nm), and Alexa 647 (excitation: 653 nm; emission: 658–775 nm).

**Single-cell quantifications for smFISH**. smFISH images were segmented and quantified using Imaris (Bitplane). A pseudo-channel was created for segmentation purposes by smoothing DAPI using a 3 × 3 gaussian blur kernel. Cell segmentation

was performed using the pseudo-channel through surface creation, with a high enough smooth factor to enlarge the soma surface to cover cell-derived smFISH signal. smFISH punctas were detected using the spot detector for each channel individually. Soma localized smFISH punctas were identified using the 'spots close to surface' XTension. The total surface object was divided into individual cells using the 'split surfaces' XTension. The number of smFISH puncta were quantified for each cell using the 'spots close to surface' XTension, and recorded for each channel along with the soma location obtained from the statistics tab.

**smFISH on morphology co-labeled tissue (morFISH)**. smFISH was performed on morphology co-labeled tissue as described in the smFISH section, with a few changes in protocol. Cerebella were sectioned at 50 μm to capture full MLI morphologies while maintaining a reasonable thickness to enable successful smFISH staining. All sectioning and staining protocols were performed while protecting the samples from light to retain the native fluorescence of the sample. No tissue clearing was performed for morFISH except a 30 min incubation in 8% SDS, to retain the protein labeling of morphology-labeled fluorescent proteins. For morFISH analyses, images of sparsely labeled MLIs were acquired by confocal microscopy. Cells were only selected for analyses if enough of the dendritic and axonal arbors were captured to confidently assign them into a morphological category. Single MLIs were categorized into: (1) basket cells, (2) definitively short-range stellate cells, or (3) definitively long-range stellate cells. Following morphological classification, the number of smFISH puncta for each marker was quantified and recorded along with the soma location of the cell.

**Clustering analyses for smFISH**. Clustering for P17 and P55 smFISH data were performed using the Scanpy package (version 1.7.2) in Python: (1) the raw smFISH puncta counts were filtered to remove cells that expressed more than 5 punctas of *Nxph1* and *Sorcs3* as they represented 'doublets' by visual inspection; (2) the raw smFISH puncta counts were imported into Scanpy as an AnnData object; (3) the data was normalized using the normalize_total() function; (4) the data was scaled to clip values exceeding standard deviation of 10; (5) PCA was run with the 'arpack' svd_solver; (6) the neighborhood graph of cells was computed using n_neighbors = 30 and n_pcs = 3; (7) the graph was embedded in two dimensions using UMAP; and (8) clustering was performed using the Leiden method[108].

**Camera Lucida illustrations**. Camera Lucida traces of dendritic morphologies were created using the Affinity Designer (Serif) app on an iPad (Apple). Maximum z-projections of the dendritic arbors were obtained in FIJI. The resulting image was inserted in Affinity Designer. Traces of the dendritic and somatic outlines were completed using an Apple pencil.

**Pseudotime trajectory inference using Palantir**. We adapted the Palantir pseudotime algorithm (v.0.2.2) to align our single-cell snapshots in the developmental space[65]: (1) The developmental dataset was zero mean standardized outside of Palantir. This step was performed individually for the total MLI dataset, and the early-born MLI subset; (2) the normalized datasets were imported into Palantir as a normalized dataframe; (3) principal component analysis was performed to determine the principal components; (4) Diffusion maps of the data were obtained to determine the low dimensional phenotypic manifold of the data, with n_components = 5; (5) t-SNE representation of the data was created in the embedded space; (6) MAGIC imputation[109] was used to map trends for each morphological parameter onto the t-SNE map; (7) starter cells were manually assigned following expert curation of the corresponding morphology traces; and (8) clustering was performed using the Phenograph package and the Louvain method[73], with $k = 20$.

**Trajectory inference using PHATE**. We adapted the PHATE algorithm (v.1.0.2) for aligning our single-cell snapshots in the developmental space[67]: (1) The developmental dataset was zero mean standardized in the same way as our Palantir analyses; (2) the normalized datasets were imported into PHATE as a normalized dataframe; (3) the dataframe was used to generate a PHATE estimator object with the following parameters: knn = 3; (4) diffusion pseudotime was performed on the PHATE operator using Scanpy[105]; (5) expert-directed maturity stages were projected onto the PHATE manifold as a color label. Expert directed staging of MLI maturation was determined based on dendritic progression and categorized in a 4-stage maturation scheme (Supplementary Fig. 4).

**Nearest neighbor analysis**. The nearest neighbor analysis were performed following PHATE-based pseudotime trajectory inference by: (1) binning the pseudo-timeline into 0.2 unit increments; (2) identifying the 5 nearest neighbors for each MLI reconstruction within the PHATE manifold; (3) calculating the proportion of nearest neighbors with identical terminal identity for each cell. This process was performed separately for BC-fated (P0 injected) and SC-fated (P4-7 injected) cells. Error bars represent 95% bootstrapped confidence intervals. The null hypothesis was calculated for each bin separately using the equation: (Number of BC/SC in bin) / (Total number of cells in bin). This was done separately for BC- and SC-fated cells.

**Experimental design and statistical analysis**. Morphological reconstructions for the 79 mature MLIs were compiled from 9 P75 mice (6 males, 3 females) across cerebellar folias. Developmental axonal reconstructions for the 732 MLIs were compiled from 32 mouse pups, which included the following injection to collection timelines: (1) P0 to P5; (2) P0 to P7; (3) P0 to P8; (4) P0 to P10; (5) P0 to P13; (6) P0 to P16; (7) P0 to P25; (8) P1 to P13; (9) P1 to P16; (10) P1 to P25; (11) P2 to P13; (12) P2 to P16; (13) P2 to P27; (14) P3 to P13; (15) P3 to P16; (16) P4 to P10; (17) P4 to P13; (18) P4 to P15; (19) P5 to P13; (20) P5 to P14; (21) P5 to P16; and (22) P7 to P14. For smFISH, two animals were analyzed for each of the P17 and P55 time points. In total, 126 P17 and 232 P55 cells were imported into Scanpy. The early-born population for pseudotime analyses contained all P0 injected samples, as well as a small number of mature P1 labeled cells which were confirmed as BCs due to the elaboration of soma-targeting axons. The late-born population contained all P4–P7 injected samples. Statistical analyses were performed using the GraphPad Prism software or the SciPy package in Python. Means of two groups were compared using the two-tailed student's t test with the Mann-Whitney nonparametric test. Graphs were created using the Seaborn (v.1.4.1) or Matplotlib (v.3.2.1) packages in Python, or using GraphPad Prism (v.7.0c).

**Reporting summary**. Further information on research design is available in the Nature Research Reporting Summary linked to this article.

## Data availability

Digital reconstructions generated in this study have been submitted to NeuroMorpho.Org under the Wang_Lefebvre archive [neuromorpho.org/NeuroMorpho_ArchiveLinkout.jsp?ARCHIVE=Wang_Lefebvre&DATE=2022-06-08a]. Morphometric data processed from the reconstructions and used for clustering (mature) and pseudotime trajectory inference (development) are provided in the Source Data file. The snRNA-Seq dataset[45] analyzed in this study are available in the NCBI GEO repository under accession code GSE165371. Source data are provided with this paper.

## Code availability

Code generated in this manuscript is available on Github located at https://github.com/wang-wendyy/Morphological_Pseudotime.

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

## Acknowledgements

This work was supported by an Ontario Graduate Scholarship (to W.X.W); a Canada Research Chair (Tier 2), a Sloan Fellowship in Neuroscience (FG-2015-65234), an NSERC Discovery grant (RGPIN-2016-06128), and CIHR Project grant (PJT-148961; to J.L.L). We thank Dr. Shreejoy Tripathy and members of the Lefebvre lab for helpful comments on this manuscript, and Dr. Manu Setty for discussion on morphological pseudotime. We thank Paul Paroutis and the SickKids Imaging Facility for assistance in optimizing five color imaging. We thank Scott Gigante for computational assistance for binning of PHATE pseudo-timelines.

## Author contributions

W.X.W. and J.L.L. conceptualized the project. W.X.W. designed and performed experiments and computational analyses. W.X.W. and J.L.L. wrote the manuscript.

## Competing interests

The authors declare no competing interests.

## Additional information



