## [Peer Review File · Nature Communications]

REVIEWER COMMENTS

Reviewer #1 (Remarks to the Author):

Wang and Lefebvre present a detailed quantitative analysis of the morphologic features of molecular layer interneurons (MLI) in both the developing and mature rodent cerebellum. Their data support the presence of two distinct morphological cell classes (basket cells and stellate cells), with a substantial degree of heterogeneity observed within the stellate cell class. Although the dataset is unidimensional (morphology only), what this study lacks in breadth it makes up for in depth of analysis; in particular I found the application of pseudotime analysis to morphologic features quite innovative. The findings help to resolve a long-standing controversy in the field and will be of direct interest to a specialty audience of neurobiologists studying cerebellar development, cell types and circuitry. In addition, the methodological approach may be of broader interest to the community of developmental neurobiologists and those interested in cell type classification in the nervous system. I believe the study could potentially merit publication in Nature Communications if the following major concerns can be addressed regarding their analysis and interpretation of the data.

Major concerns:

1. There are several major logical and statistical errors in the nearest neighbor analysis shown in Figure 7G, which supports the major novel conclusion of the paper in my opinion (that molecular layer interneurons identity is established relatively early during development). From looking at the data, I think the effect they are trying to claim is probably real, but it is not currently proven the way the data are presented and analyzed. Some suggestions are as follows:

a. The authors state in the text: “We reasoned that if the terminal fates of MLI precursors remain undetermined at the beginning of axonogenesis, the identity of a cell’s nearest neighbor should follow a largely random distribution. Correspondingly, roughly 50% of early pseudotime cells should possess a nearest neighbor of the same prospective terminal fate.” However, the null distribution will only have a mean of 50% if the probability of each cell type is exactly 50% in each bin, which is certainly not the case since the overall proportion is 423/732 (~58%) BCs and 309/732 (~42%) SCs. Computing the mean of the null distribution overall would then be approximately $(423/732 * 423 + 309/732 * 309) / 732 \sim 51.2\%$. This should be computed separately for each bin, rather than plotting a flat line across 50% as the “null hypothesis” which makes zeros sense.

b. To make any statistical claims based on this data, there has to be some measure of variance. Confidence intervals can be computed on binomial data a number of ways, including Clopper-Pearson confidence intervals, bootstrapping (i.e. randomly shuffle the labels 1000 times and recompute your test statistic), etc. In any case, there should be some measure of variance and a computed p-value to support the authors’ conclusions.

c. The statement in lines 248-249 that Figure 7G is evidence of “75-90% accuracy” is completely misleading. The absolute value of percent correct in this situation is meaningless unless compared to a true null distribution.

2. It doesn't make sense to me why the authors have used one analysis for a subset of the developmental data (Palantir) and a different analysis for the rest (PHATE). It seems that if Palantir did not work well for the whole data set, they should just not use it at all (get rid of panels 6B-O and 7A-C). If they want to keep both analyses, it should be more clearly explained why both are needed and what the differences are in methodology that might explain why one works better than the other in certain situations.

a. The conclusions given about Figure 7A-C are “difficult to interpret” and “the overall performance with this dataset was limited”. It seems that these panels could be either deleted or moved to a supplemental figure if there are really no main conclusions to be drawn, regardless of whether they keep the rest of the Palantir analysis shown in Figure 6.

Minor concerns:

1. Some of the wording in the abstract and introduction led me to believe that gene expression might also be evaluated in this study. I was subsequently disappointed when I realized, several pages in, that this was not the case. Some suggestions to reduce this effect for future readers:

a. Make it more clear in the abstract and introduction that you are talking about morphologic cell types specifically, throughout. When you use the phrases “cell types”, “cell classes”, “diversification”, “pseudo-temporal profiling” and “pseudotime trajectory mapping” without specifying the modality, many people will automatically assume you are talking about transcriptomic cell types.

b. In line 187-189 the authors state “The differentiation of MLIs is largely uncharacterized and there are no established molecular markers to distinguish BCs and SCs at maturity nor during development (Sotelo 2015; Schilling and Oberdick 2009).” This was a key part of the rationale that helped me understand why you chose to study morphologic diversity rather than do scRNA-seq, and should be mentioned much earlier in the introduction rather than buried halfway through the results in my opinion.

c. Cross-modal analysis of transcriptome and morphology would significantly strengthen this study, and potentially be able to identify molecular features/markers that distinguish these two presumptive cell types (see Kim et al., 2020 for an example of a similar scenario in which cell types could not be distinguished by transcriptome alone, but gene expression differences can be found when correlated with morphology). This question could be addressed using Patch-seq, for example (Cadwell et al., 2016, 2017; Fuzik et al., 2016; Scala et al., 2019). I realize it is likely beyond the scope of the current manuscript, but would be something to consider as a potential future direction and possibly include in the discussion section.

2. In lines 137-139 the authors state: “We reasoned that further divisions may be ambiguous due to the limitations of these tests or due to heterogeneity among the SC population.” However, this is a relatively small dataset of only 79 of complete reconstructions in mature animals, so I would assume that the main limitation is the amount of data available. Was there any sort of power analysis done to estimate how many reconstructions would be needed to fully characterize any morphologic “subclades”?

3. Relating to Figure 3 – it would be better to use an unbiased approach such as regression to identify the morphologic features that define the different clades/subclades rather than manual “inspection”.
4. I find this phrase in the abstract “the utility of quantitative single-cell methods to morphology for defining the diversification of neuronal subtypes” confusing and recommend rewording.
5. Possible typo in line line 203: “Ascl1-CreER; Ai4flox-STOP-TdTomato” - should this be Ai14 rather than Ai4?
6. In some places, I think the findings are somewhat overstated, for example in calling them “two lineages of early-born MLIs” (line 285). They are, if I understand correctly, derived from the same progenitor pool in the ventricular zone but this paper is suggesting that somewhere between the intermediate progenitor zone in the PWM and their arrival in the molecular layer, they become fated to one or the other cell type. It is an important assertion but to my mind does not make them two distinct lineages.
7. The axes in PHATE figures 7D-F and 9B-D are illegible because they are too small. Some other axes labels are a bit small as well, but these are by far the worst.
8. How can the authors know that the labeled cells represent a random sample of all MLI and are not enriched for labeling a subset of progenitors? This should be discussed somewhere.
 - a. In the Methods: “Subcutaneous injections can be substituted for labeling of stellate cells (P4 – P7 injections), but I. P. injections are necessary for capturing basket cells in our postnatal injection scheme due to its faster acting nature for tamoxifen introduction and activation.” - raises issue of bias in labeling.
9. Were experimenters blind to treatment condition during scoring of morphologic features? this should be stated in Methods.
10. What is the rationale for studying the vermis rather than the cerebellar hemispheres?
11. The raw reconstructions should be uploaded to neuromorpho.org upon publication if possible.
12. The number of male and female animals, and number at each injection-to-collection time point should be specified in the Methods section if possible.

Reviewer #2 (Remarks to the Author):

In their manuscript, Wang and Lefebvre investigated one long-lasting debate about the cell fate identity of the two molecular layer interneurons, as to whether they belong to the same class of interneuron or not. They used a morphological approach by reconstructing hundreds of interneurons during development in combination with several algorithms to highlight their distinctive

phenotypical/morphological traits. They discovered that MLIs are far more heterogeneous than originally thought, and the overall picture of 2 distinct cell-types prevails. Besides, they tracked the early development of axonal morphologies and identified an early-born SC subpopulation. The question addresses here is an important one, not only for cerebellar aficionados but also for neuroscientists interested in the field of cell diversification, which is critical for normal brain function. Although the study provides an up-to-day comprehensive quantification of MLIs morphology during development, the overall conclusion drawn from the analysis does not enhance our understanding of MLIs cell-type identity nor provides an answer to their belonging to different precursor pools. It only adds arguments in favor of the distinct precursor pools. My major concerns are as follows:

- Figures 1 to 4: These figures, although beautiful, are essentially the validation of the two types of MLIs, the Basket and Stellate cells. They can be simplified to facilitate the reading of the manuscript. Fig 1-2 and Fig 3-4 can be merged.

- The case of the two cells (Figure 4E and F) needs further explanation. Although the overall morphology seems similar, they targeted different domains of the PCs. Are they BCs or SCs? What will need to be considered for their classification?

- In the pseudotime ordering, the authors assumed that immature cells located on the apical ML are migrating (Figure 8A, B). It will be important to use some markers such as PAX-2 and Parv to be sure that these cells are still immature and potentially migrating.

- The authors need to analyze the settling patterns of MLIs (A-P, M-L, and lobules) in the *Ascl1* experiments. Do BCs and SCs distribute similarly? Mostly clusters or isolated cells? The idea being that different distributions of BCs and SCs might indicate their distinctive integration patterns. This will be interesting in the case of early-born SC.

- One important feature of SCs axons is the presence of varicose collateral axons (Chan -Palay and S. Paley 1972). Do the authors confirm these features and do they observe differences in the SCs population?

- During development, axonal rearrangement can be massive and might not follow a standard (linear) progression. Do the authors take this information into account in the pseudotime ordering? Will it impact the classification?

Reviewer #3 (Remarks to the Author):

The paper describes a new statistical approach to neuronal morphometry and applies this tool to the development of the GABAergic neurons of the cerebellum molecular layer (ML). The authors borrow the mathematical models that are normally applied to RNAseq data to perform multivariate analysis in multi-dimensional space. They use the pseudotime ordering afforded by these methods to reconstruct pseudolineages. Using these methods, the authors infer that early born stellate (SC) and basket cells (BC) of the molecular layer can be discriminated on the basis of patterns of axonogenesis from an early stage of development. They therefore suggest that neurons may be specified or committed (- “instructed” is the term used) to a given fate before they reach their final positions (rather than fate being determined by position). This runs contrary to a model that ML interneurons essentially comprise a single population of cells with two axonal types. Cells that are born early and reside close to Purkinje cells extend axons to envelope the cell bodies (basket cells). Cells in upper layers tend not to develop baskets (stellate cells).

(NB: The lower molecular layer is APICAL and the upper layer BASAL. The authors have this reversed throughout and it might be easier to refer to superficial/deep or pial/abpial).

The quality of labelling and experimental data in the paper is exemplary and the approach is novel. The cell labelling data is beautiful and the biological problem is an interesting one – trying to decipher the lineages of a relatively unexplored interneuron population in the cerebellum. I have no doubt that this methodological approach would be of interest to a large number of scientists in offering the promise of an automated lineage analysis from mixed populations of morphologies.

The paper presents two conclusions – one biological and one methodological. While the methods are novel and interesting each conclusion raises some concerns.

For the biological model, I think the authors have only generated a question that must now be answered (and has been previously addressed by grafting) by an experimental approach. The most compelling evidence that there is something to investigate further is the inference that basket and stellate cells born in the same temporal cohort might be morphologically distinct throughout their entire development. The data suggests that the statistical method has identified otherwise obscure differences in very young cells that distinguish them by morphology alone (not position). This needs to be far better illustrated and laminar position needs to be excluded as a factor from the analysis (or better explained).

For the methodological approach, the statistics offer a powerful tool for allocating very similar looking cells to correct lineages based on multivariate analysis morphology alone. While I have reservations that the authors can draw any biological by combining cells from different temporal cohorts (too many assumptions) it is nevertheless an intriguing and very interesting approach. The modelling raises the possibility of distinguishing the ultimate fate of otherwise indistinguishable developing ML and SC cells by calculating their “nearest neighbours” in morphological space. The proof-of-principle – that a cell can be assigned “blind” to a particular cohort – has only been performed with two cells. To be convincing, this needs to be tested with a large population of cells (100’s?) – at least enough to generate statistical confidence limits for the method.

Overall the approach seems to rest on implicit assumptions that could be debatable. They at least need to be justified.

- growth is progressive with no regressive events (branches cannot be lost developmentally).
- cell shape is determined purely by intrinsic factors
- the microenvironment in which cells develop is constant over time. This is particularly important given that transplantation experiments suggest the opposite.

Overall, my feeling is that the biological conclusions are overstated and would be better described as inferences. The extrinsic factors of laminar position, microenvironment etc. are never clearly dealt with. There needs to be a far more considered discussion of the range of explanations for what might be happening in development (remodelling, regressive events) and whether the statistical evidence excludes these alternatives. This is particularly important given that the cerebellum has recently been shown to be an extraordinarily plastic developmental structure (Wojcinski et al 2017. Nat Neurosci 20, 1361-1370).

Seen as an innovative methods paper, the biological questions are less important. The authors may have demonstrated a methods for teasing apart almost indistinguishable populations. However, there needs to be a stronger proof of principle approach (2 cells are not enough).

Detailed comments.

I have divided the detailed comments by Figures. An overall comment is that the distribution of main text and supplementary data is confusing, but that more confusing is the separation of injections, cell morphologies with their respective analysis. The paper would have been much easier to read if it had been broken down into a series of clear steps rather than combining injections data in Figure 5.

****FIGURES 1-4:**

These show that the tendency for MLI cells to form baskets is limited largely, but not exclusively to the lower ML.

Within the continuously variable population, I worry that parameters are not dimensionless and that a number of measures are clearly dependent of position (not independent). For example, an apical stellate cell can have no branches directed up and vice versa for a basal Basket cell. Measures of %ML covered by axon and dendrite, length and % of branches directed up or down are just a proxy for laminar position. Axon span is inversely correlated with basket number while axon length remains constant (Fig.S3). This, for example, could suggest that there is relatively consistent length of axon but the overall span is reduced when this finite length is wrapped around Purkinje cell bodies.

It appears that BCs and SCs are morphologically identical apart from the formation of baskets. The only truly discriminative variable for cell type is axon morphology.

For example, Figure 2 beautifully shows that, broadly speaking, only cells in the lower ML have baskets but there are cells with baskets in the middle molecular layer and stellate cells in the lower molecular layer without baskets. The decision (Figure 3) to subjectively classify some cells with baskets as SC3 stellate cells (rather than displaced basket cells) is unclear and slightly confusing.

For the entire population cells that are closer to Purkinje cell layer have bigger soma, a shorter axon span and less densely packed dendritic trees (Figure S3). This trend remains the same when cells with baskets are removed. When the presence of baskets is allowed as a parameter, BC cells segregate as a cluster (Figure S3, S4). When axonal parameters are removed including the discriminatory weighted basket number the cluster disappears (Figure S2B, Figure 4I).

Therefore, while it is fair to say that baskets define basket cells (Line 179), for all other parameters BCs fall within the range of values defined by the SC population and are therefore indistinguishable.

This is a very strong and inherently interesting result. For this reason, a more detailed explanation of the next sections (which argue the opposite) is particularly important.

****FIGURE 5:**

Tamoxifen labelling at 3 time points beautifully confirms that laminar position is determined by birthdate

Lower ML cells (and hence the majority of BCs) born first. Some cells in this early born population either fail to form baskets or lose baskets during development.

I think it is important that the possibility that there may be a loss of axons should be taken into account as is an alternative. This is particularly relevant given well documented evidence of climbing fibre competition for synaptic territory on the (same) Purkinje cell bodies.

The following statement could give the impression that authors think that this is evidence of separate fate allocations – a statement that I don't think is supported by the evidence

(Line 224-5): "The marking of early-born MLIs that elaborate distinct phenotypes in the lower ML suggests the divergence of BC and SC fates during early postnatal development".

I feel that this figure should be broken into two parts and included with Figure 6 and 7 as follows.

****FIGURE 6:**

This should be combined with Figure 5 P0 injections and Figure S5

In the following sections, I feel that the paper could be considerably improved by grouping cell labelling strategy and analysis together and bringing Supplementary Figure 5 into the main body of the paper

Figure 6 concerns only the injections of type that give rise to views in Fig.5C and D.

- The P0 injections in Figure 5C and D should be included with Figure 6 to make it clear that the analysis is of the P0 injected population.
- The labelling strategy, cell morphologies during development and final population structure should all be within the same figure.
- I think it would be very useful to have the stages (1-4) for this population brought in from Fig.s5

- It would be really informative to see a photomicrograph of the complex mix of morphologies indicated schematically in Figure 6A (does it look like Fig.8C).
- Finally, the authors should show a representative sample of cells from cluster 5 versus 3 and 6 versus 7 mapped onto laminar position. We have to be visually reassured that laminar position is independent of morphological differences. We need to be able to have clear pictures of the morphological differences between early born stellate and basket cells, which are described in the text (Lines 274-286).

**FIGURE 7:

This should be combined with a new, distinct text section and be combined with Fig.5 (P4,7 injections) and Figure S5

I'm not convinced that Figure 7A, B, C. add much to the paper as a whole and whether these should (if needed to be shown) be placed into Supplementary data.

The later stage injections (Fig.5) combined with Figure S5 (to give a view of proposed developmental stages 1-4 for this population) could then be combined with PHATE data which the authors feel is a more productive approach (Figure 7D-G).

My chief concerns about this data is that it mixes cohorts from two very different temporal environments. There are two potential confounding factors:

1. Extrinsic factors: How has the extracellular environment changed between P0, P4 and P7? Cells will grow very differently on different substrates and the subtle differences in shape that cluster cells together may simply be a product of a temporally changing microenvironment
2. Crowding: Are later born cells excluded from the lower ML restricting them to the upper ML. If laminar position (and co-dependent parameters) are taken out of the statistics would the clustering (and nearest neighbour relations) remain?

While the nearest neighbour conjecture is plausible "if the terminal fates of MLI precursors remain undetermined at the beginning of axonogenesis, the identity of a cell's nearest neighbour should follow a largely random distribution" (line 303-304), this does not allow any conclusion about any biological mechanism or commitment.

The only experimental test of commitment to a certain fate is heterochronic transplantation.

****FIGURE 8:**

This confirms that late born cells occupy more superficial laminae and do not make baskets. Beautiful as it is, this figure is not necessary for the paper.

I would not agree that this figure support pseudotime clusterings. What it supports is that axons of cells that become basket cells undergo significant remodelling to make these baskets. The data plots in Fig.8G are not clearly explained, nor are any details given of the derivation of statistical significance in the differences in the data (no error bars etc).

This figure is essentially redundant

****FIGURE 9:**

Proof of concept requires a much large number of cells from a range of stages (1-4)

Regardless of any underlying biology, if the manifold was able to correctly allocate a cell to the correct lineage the analysis would indeed prove powerful. In other words, it should be able to statistically identify a cell of a given birth date by morphology alone at any given developmental stage (1-4)

This needs to be done for hundreds of (not just two) cells to derive some statistical power for the approach.

****Discussion**

The overall approach is interesting but the interpretation and conclusions are overstated. What the study has determined is an inference that there might be two lineages at P0 (Fig.6) and suggested (but not proved) a powerful predictive approach for analysing morphology (Fig.9) by large scale sampling.

I feel that it cannot show that fates are “instructed” early (line 477). The only test for this is heterochronic transplantation (Leto et al). The main problem (to me) is that, by combining data from multiple temporal cohorts into a single data set, any differences in extrinsic variables are removed.

For the biological question, factors such as regressive events, a changing microenvironment and, most importantly, that this data is only generating inferences that must be tested are not addressed.

For the powerful methodological conclusions, there is simply not enough data to offer a convincing proof-of-principle. So, for example, this statement is simply not a valid conclusion – but a possible hypothesis (Line 364): “In doing so, we determined that MLI subtype identities emerge during migration prior to reaching sites of final integration”.

In the section (Line 389) “A revised taxonomy of MLIs”, I am not sure what the authors are proposing is a new taxonomy. Their data clearly supports a continuous variation in dendritic form with a sharp discontinuity in axonal structures (lower MLI cells almost exclusively form baskets). This seems to be – albeit beautifully demonstrated by the experiments - essentially the current model.

In the section (Line 427) “MLI axonogenesis begins during migration” I think all the useful insight comes from cell labelling and not from the statistical model. That young cells extend neurites (that may or may not be young axons) is evident from the pictures (Figure S5) and not from the statistics (Line 443).

In the section (line 452) “early emergence of MLI identities”, I find it difficult to get the arguments about biological mechanism and timing of fate decisions. The implications that BC identity is predetermined in white matter, and how numbers might be matched to available Purkinje cell space needs to be better explored. Overall, the cerebellum appears to be a network of cells where cell number matching is tightly controlled by interactions and can be recapitulated following injury (see Joyner lab recent results). However, I do like the argument that lower ML SC might simply be slower to reach their potential targets and fail to establish or retain baskets. This is not the same, I feel, as being “instructed”.

AUTHORS' RESPONSE

We thank the reviewers for their thoughtful reviews and suggestions on our manuscript **Morphological pseudotime ordering and fate mapping reveal diversification of cerebellar inhibitory interneurons (NCOMMS-20-15800)**. We considered all of the comments and have extensively revised our manuscript. We include new data and new figures. In response to a reviewer comment, we present a new cross-modal analysis of molecular layer interneurons (MLIs) in which we compare our morphological divisions to transcriptomic signatures from a recent report of snRNA-Seq dataset obtained from the whole adult cerebellum (Kozareva et al BioRxiv, 2020, recently published now in Nature 2021). This analysis was critical because the transcriptional signatures do not corroborate with the Stellate cell/Basket cell division, and confused the long-standing descriptions of MLIs. In new data (**Figures 4-6**), we map the expression of top differentially-expressed transcripts by dual smFISH-morphology to correlate the morphological and transcriptional subtypes, and to define MLI diversification over development. We elaborate on these findings below, and include a point-by-point response to the reviewers' comments. All changes in the manuscript are indicated in blue. We think the revisions and new data greatly improved the significance, clarity and rigor of our manuscript.

REVIEWER COMMENTS

Reviewer #1 (Remarks to the Author):

Wang and Lefebvre present a detailed quantitative analysis of the morphologic features of molecular layer interneurons (MLI) in both the developing and mature rodent cerebellum. Their data support the presence of two distinct morphological cell classes (basket cells and stellate cells), with a substantial degree of heterogeneity observed within the stellate cell class. Although the dataset is unidimensional (morphology only), what this study lacks in breadth it makes up for in depth of analysis; in particular I found the application of pseudotime analysis to morphologic features quite innovative. The findings help to resolve a long-standing controversy in the field and will be of direct interest to a specialty audience of neurobiologists studying cerebellar development, cell types and circuitry. In addition, the methodological approach may be of broader interest to the community of developmental neurobiologists and those interested in cell type classification in the nervous system. I believe the study could potentially merit publication in Nature Communications if the following major concerns can be addressed regarding their analysis and interpretation of the data.

Major concerns:

1. There are several major logical and statistical errors in the nearest neighbor analysis shown in Figure 7G, which supports the major novel conclusion of the paper in my opinion (that molecular layer interneurons identity is established relatively early during development). From looking at the data, I think the effect they are trying to claim is probably real, but it is not currently proven the way the data are presented and analyzed. Some suggestions are as follows:

a. The authors state in the text: "We reasoned that if the terminal fates of MLI precursors remain undetermined at the beginning of axonogenesis, the identity of a cell's nearest neighbor should follow a largely random distribution. Correspondingly, roughly 50% of early pseudotime cells should possess a nearest neighbor of the same prospective terminal fate." However, the null

distribution will only have a mean of 50% if the probability of each cell type is exactly 50% in each bin, which is certainly not the case since the overall proportion is 423/732 (~58%) BCs and 309/732 (~42%) SCs. Computing the mean of the null distribution overall would then be approximately $(423/732 * 423 + 309/732 * 309) / 732 \sim 51.2\%$. This should be computed separately for each bin, rather than plotting a flat line across 50% as the “null hypothesis” which makes zero sense.

b. To make any statistical claims based on this data, there has to be some measure of variance. Confidence intervals can be computed on binomial data a number of ways, including Clopper-Pearson confidence intervals, bootstrapping (i.e. randomly shuffle the labels 1000 times and recompute your test statistic), etc. In any case, there should be some measure of variance and a computed p-value to support the authors’ conclusions.

Response R1-1:

a. For the nearest neighbor analysis, we re-calculated the null distribution for each pseudotime bin as suggested. We performed this analysis separately for BCs and SCs, as sampling was not equivalent for each pseudotime bin. This is shown in the revised **Figure 8f**.

b. We have generated confidence intervals (95% CI) for each pseudotime bin using bootstrapping with replacement, which are now included in **Figure 8f**. We have additionally included the p values for each bin within both the BC and SC cohorts. As the reviewer suggested, these analyses confirm our findings that BC- and SC-specific phenotypes segregate early in pseudotime.

c. The statement in lines 248-249 that Figure 7G is evidence of “75-90% accuracy” is completely misleading. The absolute value of percent correct in this situation is meaningless unless compared to a true null distribution.

R1-1c: The reviewer referred to lines 348-349, which related to former Figure 9 that presented a test to predict MLI identity based on morphology and the nearest neighbor analysis. In response to reviewer 3’s critique, we deleted Figure 9 as we acknowledge that the classifier requires testing of 100s of additional cells to demonstrate a statistically robust outcome.

2. It doesn’t make sense to me why the authors have used one analysis for a subset of the developmental data (Palantir) and a different analysis for the rest (PHATE). It seems that if Palantir did not work well for the whole data set, they should just not use it at all (get rid of panels 6B-O and 7A-C. If they want to keep both analyses, it should be more clearly explained why both are needed and what the differences are in methodology that might explain why one works better than the other in certain situations.

a. The conclusions given about Figure 7A-C are “difficult to interpret” and “the overall performance with this dataset was limited”. It seems that these panels could be either deleted or moved to a supplemental figure if there are really no main conclusions to be drawn, regardless of whether they keep the rest of the Palantir analysis shown in Figure 6.

R1-2: We agree with the reviewer. We reorganized this section, and deleted former Figure 7A-C, which showed the Palantir projection for the whole dataset. We maintained the PHATE analysis for the complete MLI dataset (Figure 8), and Palantir for the early-born BC/SC dataset (Figure 9), as

these data together rendered a robust division of BC/SC phenotypes. They also provide proof-of-concepts for applying pseudotime algorithms to morphological data for modelling trajectories spanning several weeks of development. For both analyses, we validated the progression for cells in each cluster through visual inspection, and quantification of individual features along the pseudo-timeline (Figure 8i). PHATE and Palantir were both developed for modelling transcriptomic data. Given the large number of pseudotime algorithms currently available, each with varying strengths and weaknesses, we believe that our validation of multiple pseudotime algorithms towards morphological data is a strength of our study.

We reorganized the rationale for using two separate approaches in our manuscript. We selected PHATE as it is a graph diffusion-based embedding approach developed for preserving both local and global similarities in data structure: “PHATE was developed for visualization of branching data structures, with preservation of both local and global similarities (Moon et al., 2019)” (line 339). It is designed to handle noisy, non-linear relationships between data points and preserve transitions. It has been applied to non-transcriptomic datasets, including images of human facial expression (Moon et al. 2019), leading us to believe that it was well suited for application to neuronal morphology.

Palantir, on the other hand, is a pseudotime algorithm with demonstrated utility and higher-resolution for identifying rare cellular lineages and differentiation trajectories from transcriptional datasets (Setty et al. 2019): “We next selected Palantir, a pseudotime algorithm with demonstrated utility and resolution for identifying rare cellular lineages and differentiation trajectories from transcriptional datasets (Setty et al., 2019)”. (Line 387).

Accordingly, Palantir proved effective for deciphering two separable trajectories of early-born MLIs in our study, including a rare population of early-born stellate cells. We additionally applied PHATE towards this purpose, but was unsuccessful, likely due to the denoising capabilities to avoid spurious paths but which removed rare populations as noise. On the other hand, Palantir was useful in the detection of rare cellular lineages in our dataset but tended to oversegment for more heterogeneous populations, especially if the data are not equally distributed. Differences in the performance of pseudotime algorithms for various datasets is consistent with what has been for transcriptomics datasets (Saelens et al., 2019), but further suggest that future studies will benefit from algorithms optimized for morphological data.

Minor concerns:

1. Some of the wording in the abstract and introduction led me to believe that gene expression might also be evaluated in this study. I was subsequently disappointed when I realized, several pages in, that this was not the case. Some suggestions to reduce this effect for future readers:

a. Make it more clear in the abstract and introduction that you are talking about morphologic cell types specifically, throughout. When you use the phrases “cell types”, “cell classes”, “diversification”, “pseudo-temporal profiling” and “pseudotime trajectory mapping” without specifying the modality, many people will automatically assume you are talking about transcriptomic cell types.

R1-3: We thank the reviewer for the point, since these terms are now widely used for cell-type classification in transcriptomics. The distinction is important for the revised manuscript, as we updated it with a cross-modal comparisons of MLI subtypes. We distinguish morphological types and transcriptomic types, adopting the terms m-type and t-type, respectively. We modified the abstract and the introduction to orient the readers and clarify the scope of our study. Where

relevant, we clarify that we adapt methods widely used for single cell transcriptomic analysis to morphometric information.

b. In line 187-189 the authors state “The differentiation of MLIs is largely uncharacterized and there are no established molecular markers to distinguish BCs and SCs at maturity nor during development (Sotelo 2015; Schilling and Oberdick 2009).” This was a key part of the rationale that helped me understand why you chose to study morphologic diversity rather than do scRNA-seq, and should be mentioned much earlier in the introduction rather than buried halfway through the results in my opinion.

R1-4: In the introduction, we state the rationale for focusing on morphological information, as it provides the basis for the classification of MLIs into BCs and SCs, as well as the historical observations of continuous variation among MLIs.

We also point out the lack of molecular markers to distinguish them at maturity and during their differentiation. “A barrier to resolving MLI subtypes has been the lack of molecular markers to distinguish BCs and SCs and to track MLI differentiation (Glassmann et al., 2009; Schilling and Oberdick, 2009; Sotelo, 2015).

c. Cross-modal analysis of transcriptome and morphology would significantly strengthen this study, and potentially be able to identify molecular features/markers that distinguish these two presumptive cell types (see Kim et al., 2020 for an example of a similar scenario in which cell types could not be distinguished by transcriptome alone, but gene expression differences can be found when correlated with morphology). This question could be addressed using Patch-seq, for example (Cadwell et al., 2016, 2017; Fuzik et al., 2016; Scala et al., 2019). I realize it is likely beyond the scope of the current manuscript, but would be something to consider as a potential future direction and possibly include in the discussion section.

R1-5: We thank the reviewer for the suggestions. We agree that an understanding of neuronal cell types is strengthened when they are characterized by using multiple modalities. In the revised manuscript, we incorporate cross-modal analysis by comparing subtype divisions based on morphological and transcriptome information, using a recently reported snRNA-Seq dataset of the adult mouse cerebellum (Kozareva et al., 2020, *Biorxiv*). Kozareva et al 2020 subdivided the MLIs into two transcriptomic types marked by *Sorcs3* and *Nxph1* expression, respectively. The authors noted that these types do not obviously correlate with canonical BC/SC morphologies, but this analysis was limited to a qualitative survey of a small number of cells, some of which we noted do not reside within the molecular layer. To determine how the MLI transcriptomic types relate to our morphological types, we further analyzed the snRNA-Seq dataset and identified other differentially expressed genes (DEGs). We mapped the RNA transcripts onto the morphological groups by smFISH, and established the following:

#1. The *Sorcs3*⁺ and *Nxph1*⁺ molecularly-defined MLI types do not correlate with the BC or SC morphological division, confirming findings from Kozareva et al., 2020 (new **Figure 4, 5**).

#2. Within the *Sorcs3*⁺ transcriptomic population, we identified DEGs that show continuous variation between clusters. In particular, *Grm8* and *Cacna1* are expressed in an opposing gradient (new **Figures 4c-d, 6a-g**). We show by smFISH that the continuous organization of some

transcripts within the *Sorcs3*⁺ population spatially corresponds to a laminar organization within the molecular layer.

#3. The BC/SC morphological types can be defined by marker combinations (new **Figure 5**):

BCs: *Sorcs3*⁺/*Grm8*^{+HIGH}/*Cacna1e*^{+LOW}. BCs are *Nxph1*-negative.

SCs: are marked by either of these non-overlapping signatures:

- i) *Sorcs3*⁺/*Grm8*^{+LOW}/*Cacna1e*^{+HIGH} longer axonal-range m-type SCs; or
- ii) *Nxph1*⁺. These cells mostly consist of non-basket forming SCs with short axonal spans we observe in the lower to upper two-thirds of the molecular layer. However, *Nxph* expression is not limited to short-range morphologies as we observed *Nxph1*⁺ SCs with longer-axons in the very upper ML.

#4. In smFISH of developing MLIs (P7-P17; **Figure 6**), we show that expression of these markers emerge later in development, after MLIs settle in their location and express Parvalbumin (Maricich, S.M., and Herrup, K. (1999)). *Sorcs3* and *Nxph1* RNA are not detected in immature MLIs marked by *Pax2*. In quantitative smFISH analysis at P17, we show that the marker signatures initially have broader, overlapping expression patterns, but segregate by P55 (**Figure 6, supplementary Fig. 3**).

The new molecular-morphological analysis advances this study in the following ways: First, our study illustrates how cell-type identities can differ when evaluated through different modalities, i.e. morphology vs transcriptional signatures. A small subset of gene expression differences can be found when correlated with morphology/connectivity, similar to Kim et al., 2020; Que et al. 2021. Second, by extending the cross-modal comparison to development, as done in previous studies, our study illustrates how morphological signatures established during development may not be represented by transcriptional signatures at maturity. Future studies are required to identify relevant genetic programs that underlie the morphological diversification of MLIs. We expand on these points in the discussion.

2. In lines 137-139 the authors state: “We reasoned that further divisions may be ambiguous due to the limitations of these tests or due to heterogeneity among the SC population.” However, this is a relatively small dataset of only 79 of complete reconstructions in mature animals, so I would assume that the main limitation is the amount of data available. Was there any sort of power analysis done to estimate how many reconstructions would be needed to fully characterize any morphologic “subclades”?

R1-6: The reviewer raises an important but complex point of sample size for clustering. We did not perform power analysis as there are no established methods for defining statistical power for clustering analyses to our knowledge, particularly when the structure of the data is unknown (Dalmaijer et al., ArXiv, 2020). 79 reconstructions is a large dataset for related cell types arising from similar anatomical locations (ie. Que et al. 2021 performed morphological analysis of 66 Pvalb⁺ interneurons from the hippocampus). While we cannot fully exclude the possibility that subdivisions can be obtained for SCs by significantly expanding the dataset, which was not feasible at this time, we do not see how this would enhance the findings of the paper. Instead we further analyzed the continuous features of SCs by performing the following:

#1. We performed iterative clustering analysis to compare the statistical outcomes of MLI classification by altering sample sizes. We sampled smaller numbers of cells randomly selected from our dataset, and assessed the success of subdividing the population and classifying each cell to

their respective BC or SC division. A group of 20 reconstructions is sufficient to reproduce the discrete BC and SC division (**Figure 2j**; classification accuracy = $96 \pm 1.26\%$ for 20 cells). By contrast, the division of SCs into long- and short-range cells was error prone and only reached $89 \pm 2.3\%$, when using 70 cells (**Figure 2j**).

#2. To test whether SCs exhibit continuous variation, we applied partition-based graph abstraction (PAGA) analysis of the morphometric data. PAGA was developed for mapping of discrete and continuous cell transitions in single cell data, and is used routinely for single cell transcriptomic analyses (Kozareva et al., 2020; Stanley et al., 2020; Wolf et al., 2019). The PAGA plot for mature MLI morphologies confirmed discrete clustering between BCs and SCs, and identified connectivity within the SCs (**Figure 2c**). We inspected the cells sorted in each of the SC nodes, and determined that they corresponded to SCs with shorter axons and longer axons, further suggesting continuity between the SCs. Plotting individual axonal parameters of SCs against their hierarchical rank also revealed a continuum of measurements such as axonal span (**Figure 2k**). If there was a discrete separation of SCs into subtypes, then we would expect the plot to show discontinuous distributions.

We removed the term ‘morphological subclades’.

3. Relating to Figure 3 – it would be better to use an **unbiased approach** such as regression to identify the morphologic features that define the different clades/subclades rather than manual “inspection”.

R1-7: We thank the reviewer for the excellent suggestion. As stated in **R1-6**, we removed the term subclade and analyzed the SC morphological data using multiple approaches that support a continuity of the morphological features within the SC subtype.

4. I find this phrase in the abstract “the utility of quantitative single-cell methods to morphology for defining the diversification of neuronal subtypes” confusing and recommend rewording.

R1-8: We modified this sentence.

5. Possible typo in line 1203: “Ascl1-CreER; Ai4flox-STOP-TdTomato” - should this be Ai14 rather than Ai4?

R1-9: We corrected the typo.

6. In some places, I think the findings are somewhat overstated, for example in calling them “two lineages of early-born MLIs” (line 285). They are, if I understand correctly, derived from the same progenitor pool in the ventricular zone but this paper is suggesting that somewhere between the intermediate progenitor zone in the PWM and their arrival in the molecular layer, they become fated to one or the other cell type. It is an important assertion but to my mind does not make them two distinct lineages.

R1-10: We revised the statements regarding our findings (see Reviewer 3 for examples). We clarified the terminology by stating that the MLIs derive from a common lineage but diverge into distinct subtypes.

7. The axes in PHATE figures 7D-F and 9B-D are illegible because they are too small. Some other axes labels are a bit small as well, but these are by far the worst.

R1-11: We increased the font sizes for the figure labels.

8. How can the authors know that the labeled cells represent a random sample of all MLI and are not enriched for labeling a subset of progenitors? This should be discussed somewhere.

a. In the Methods: “Subcutaneous injections can be substituted for labeling of stellate cells (P4 – P7 injections), but I.P. injections are necessary for capturing basket cells in our postnatal injection scheme due to its faster acting nature for tamoxifen introduction and activation.” - raises issue of bias in labeling.

R1-12: To ensure that our labeling methods sampled the full MLI population, we used molecular layer position as a readout to show that the adult dataset includes morphologies across the molecular layer depth (by the AAV/Gad2-Cre, **Figure 1f**). Our new smFISH data of the MLI subtype molecular markers *Nxph1* and *Sorcs3* confirm that we are analyzing both transcriptional MLI subpopulations (**Figure 5**). As shown in new **Figure 4e, f**, *Nxph1* and *Sorcs3* label mutually exclusive MLIs subtypes, but together label the entire MLI population (marked by pan-MLI/Purkinje cell marker Parvalbumin). This experiment was important because the *Nxph1* + subtype has a lower density and is mostly excluded in the deepest layer of the molecular layer.

In the discussion, we note: “Our morphological dataset sampled across the MLI population based on molecular layer position, and further confirmed by co-labeling with *Nxph1* or *Sorcs3* which together account for all MLIs. (Line 449)

For the developmental analysis (in reference to the comment re. Methods), the tamoxifen-inducible *Ascl1-CreER* line is biased in that it labels MLI progenitor subsets in a birthdate-specific manner as that is the feature of the transgene (Sudorov et al. 2011). As shown in **Figure 7**, we confirmed that tamoxifen induction at different postnatal timepoints labels MLI subsets that occupy different strata, including early-born BCs in the deepest molecular layer (P0 injected) to the superficial SCs (P7 injected).

In the Methods, we clarify that we delivered tamoxifen to pups by I.P. injections and reproducibly obtained BC/early born SCs or late-born SCs. We removed: “Subcutaneous injections can be substituted for labeling of stellate cells (P4 – P7 injections)...”; we had initially referred to the SC method because it had been used by other groups, but this turned out to be confusing. We replaced it with: “IP injections of P0-P1 pups proved to be reproducible for labeling basket cells and early born stellate cells at P0, P1 and stellate cells at P4, P7, as judged by laminar locations and phenotypes.

9. Were experimenters blind to treatment conditions during scoring of morphologic features? this should be stated in Methods.

R1-13: We now state in the Methods: “The experimenter was blinded to the injection time point while compiling and scoring morphological features”.

10. What is the rationale for studying the vermis rather than the cerebellar hemispheres?

R1-14: We focused the analysis on the vermis cerebellum for three reasons. First, this allowed us to maintain developmental consistency for CreER-labeling across animals, as the MLIs labeled in the lateral cerebellar hemispheres displayed different developmental stages. Second, MLIs display planar morphologies that align the dendritic arbors of Purkinje cells, along the sagittal plane. The orientation of the vermis cerebellum along the sagittal plane allows capture of the entire MLI dendritic and axonal morphologies within a single 100µm slice. We lose this orientation in lateral sections due to the curved shape of lateral hemispheres. Third, the height of the molecular layer varies greatly through regions of the cerebellar hemispheres, which complicates normalization of morphological parameters to ML position.

We state in the results: “We selected neurons in the cerebellar vermis for capturing these planar cells within the sagittal orientation and for consistent molecular layer thickness.” In the methods: “To optimize the consistency of the morphological comparisons, we selected neurons in the cerebellar vermis to capture cells in the optimal sagittal orientation and with complete dendritic and axonal arborizations within the 100µm slice.”

11. The raw reconstructions should be uploaded to neuromorpho.org upon publication if possible.

R1-15: We have been in contact with Neuromorpho and will deposit our raw reconstructions as .ims files with Neuromorpho.org upon publication.

12. The number of male and female animals, and number at each injection-to-collection time point should be specified in the Methods section if possible.

R1-16: We thank the reviewer for this point. The sex information for the mature MLI dataset is included in the “Materials and Methods” section, under “Experimental design and statistical analysis”. In total, 79 cells were analyzed from 9 animals, consisting of 6 males and 3 females. The methods section with the number of each injection-to-collection time point is also included. However, we did not record sex information for our developmental analyses (32 animals in total), due to difficulties in sexing pups. We did not genotype for sex.

Reviewer #2 (Remarks to the Author):

In their manuscript, Wang and Lefebvre investigated one long-lasting debate about the cell fate identity of the two molecular layer interneurons, as to whether they belong to the same class of interneuron or not. They used a morphological approach by reconstructing hundreds of interneurons during development in combination with several algorithms to highlight their distinctive phenotypical/morphological traits. They discovered that MLIs are far more heterogeneous than originally thought, and the overall picture of 2 distinct cell-types prevails. Besides, they tracked the early development of axonal morphologies and identified an early-born SC subpopulation. The question addresses here is an important one, not only for cerebellar aficionados but also for neuroscientists interested in the field of cell diversification, which is critical for normal brain function. Although the study provides an up-to-day comprehensive quantification of MLIs morphology during development, the overall conclusion drawn from the analysis does not enhance our understanding of MLIs cell-type identity nor provides an answer to their belonging to different precursor pools. It only adds arguments in favor of the distinct precursor pools.

My major concerns are as follows:

1. Figures 1 to 4: These figures, although beautiful, are essentially the validation of the two type MLIs, the Basket and Stellate cells. They can be simplified to facilitate the reading of the manuscript. Fig 1-2 and Fig3-4 can be merged.

We thank the reviewer for these insightful suggestions.

Response R2-1: We merged Figures 1 and 2 together (**Figure 1**). We reorganized former Figures 3 and 4 together and added new data (**Figure 2 and a smaller Figure 3**).

2. The case of the two cells (Figure 4E and F) needs further explanation. Although the overall morphology seems similar, they targeted different domains of the PCs. Are they BCs or SCs? What will need to be considered for their classification?

R2-2: Indeed the two cells from the former Figure 4E and F (now **Figure 2g, h**) are morphologically similar, but one extends a few axon collaterals targeting PC somata. Both cells are categorized as stellate cells by multiple unsupervised clustering analyses of morphological features.

Cells in the BC group display PC-targeting axon collaterals and basket formations, which range from full and partial baskets; most appeared to form pinceaux, but we did not confirm these structures by molecular criteria (ie. KV1.1, PSD-95). We found that many SCs display graded morphologies with PC soma-targeting collaterals and partial baskets (**Figures 1i and 2h**). A major conclusion of this study is that BC vs SC classification cannot be defined by the presence of basket or axon collaterals alone, and thus requires multiple axonal parameters (see also **R3-3**).

3. In the pseudotime ordering, the authors assumed that immature cells located on the apical ML are migrating (Figure 8A, B). It will be important to use some markers such as PAX-2 and Parv to be sure that these cells are still immature and potentially migrating.

R2-3: We confirmed that morphologically immature MLIs located in the superficial ML, similar to those shown in **Figure 9g, h** (previously: Figure 8A, B), are Parvalbumin negative (immunostaining shown in **Supplementary Figure 6**). These cells have elongated soma, and tangentially oriented processes, in contrast to the Pvalb⁺ cells in the lower ML with rounded soma and increasingly complex arbors. Unfortunately, we were unable to obtain good immunolabeling of Pax2, despite testing two commercial antibodies. Instead, we show in **Figure 6** smFISH stainings for *Pvalb* and *Pax2* that show the spatial segregation of immature *Pax2*⁺ cells to the upper ML and maturing *Pvalb*⁺ cells to the lower ML at P10.

4. The authors need to analyze the settling patterns of MLIs (A-P/fovia, M-L, and lobules) in the *Ascl1* experiments. Do BCs and SCs distribute similarly? Mostly clusters or isolated cells? The idea being that different distributions of BCs and SCs might indicate their distinctive integration patterns. This will be interesting in the case of early-born SC

R2-4: We did not observe differences in distributions for BC/early-born or SC/late-born cells associated with specific lobules, folia or anterior-posterior regions. To visualize this, we added a

PHATE plot in which each cell is coded for its folia location; it shows a mixed distribution of cells (**Supplementary Figure S5a**, and pasted below).

With regard to the medial/lateral axis, we focused our analyses on the vermis cerebellum for consistency of cell labelling, molecular layer thickness, MLI orientation and developmental staging (See R1-14). Although we did not note obvious differences, we cannot exclude the possibility that BCs and SCs follow distinct migration patterns within the lateral cerebellum. We report in the text that MLI analysis is focused on the cerebellar vermis. In the results, “We selected neurons in the cerebellar vermis for capturing these planar cells within the sagittal orientation and for consistent molecular layer thickness”. **In the methods:** To optimize the consistency of the morphological comparisons, we selected neurons in the cerebellar vermis to capture cells in the optimal sagittal orientation and with complete dendritic and axonal arborizations within the 100 μ m slice”.

5. One important feature of SCs axons is the presence of varicose collateral axons (Chan-Palay and S. Palay 1972 PMID 5042759). Do the authors confirm these features and do they observe differences in the SCs population?

R2-5: We thank the reviewer for the interesting remark. We noted the presence of axonal varicose collaterals in SCs labeled by membrane-targeted XFPs, and observed them in SCs across the continuum (see Figure 2). We did not include axonal varicosities as one of the morphometric parameters and thus we do not have a quantitative measurement of varicose axon collaterals by subtypes. Qualitatively, we noted that basket cells also present axonal varicosities in their upwards orientated axon collaterals, suggesting that this feature is not specific to stellate cells. We include an example image of axonal varicosities on BCs below.

6. During development, axonal rearrangement can be massive and might not follow a standard (linear) progression. Do the authors take this information into account in the pseudotime ordering? Will it impact the classification?

R2-6: We thank the reviewer for raising this important question, as our experimental approach using pseudotime and morphometric quantifications are well suited for this type of longitudinal analysis over several days. A similar one was raised by R3 (**R3-4**).

The quantifications and visual validations of the axonal reconstructions (**Figures 8 and 9**) support a progressive growth and arborization pattern for the SC- and BC-fated groups. We did not detect features suggesting that MLIs undergo regressive axonal pruning or remodeling, which might have been fortuitous for using this dataset as a test site for this application.

We incorporated several validation measures to discern the performance of pseudotime. By visual inspections, we confirmed an overall progressive axonal growth by comparing cells with advancing pseudotime rank and their dendritic and axonal morphologies. As shown in **Figure 8g-h** and **9**, MLIs with early pseudotime ranks extend long primary axons (i.e. purple, green traces) and then exhibit axonal branching at later stages (blue traces). Quantifications of individual parameters over pseudotime (**Figure 8**) or spanning the latest pseudotime stage to maturity (**Supplementary Figure S7**) also show increases in total axonal length, span and branching, supporting a progressive growth along the the pseudo-timeline. We noted that branch collaterals to Purkinje cell soma form with more advanced stages (Figure 8g, h), consistent with previous characterizations (Cioni et al Current Biology 2013; Telley et al. Neuron 2016). One interesting exception is the axonal span for the SCs (along the horizontal axis), which retract in span over development, but not in branching complexity or total axon length (**Figure 8i**, confirmed through **Supplementary Figure S7**). The ability to model regressive events in single morphological parameters over pseudotime highlights the utility of this approach for tracking morphogenesis.

We cannot comment on whether the pseudotime algorithms will properly order axonal rearrangements. We raise this point in the discussion: “The second notable finding is that BC and

SC axonal arborization is progressive. Axonal lengths and complexity increased over pseudotime and were greatest at maturity, consistent with reports that axonal branching towards the PC soma and baskets form later in development (Telley et al., 2016). From the quantifications and visualizations of MLI labeling, we did not detect axonal pruning, but did detect retraction in the axonal span of SCs. Thus, pseudotime ordering can account for regressive events in single or subsets of parameters, further highlighting the utility of this approach for reconstructing morphogenesis. As pseudotime algorithms assume globally linear trajectories however, further investigations are needed to assess the performance for cell types that undergo non-linear growth such as axonal remodeling.

Reviewer #3:

The paper describes a new statistical approach to neuronal morphometry and applies this tool to the development of the GABAergic neurons of the cerebellum molecular layer (ML). The authors borrow the mathematical models that are normally applied to RNAseq data to perform multivariate analysis in multi-dimensional space. They use the pseudotime ordering afforded by these methods to reconstruct pseudolinesages. Using these methods, the authors infer that early born stellate (SC) and basket cells (BC) of the molecular layer can be discriminated on the basis of patterns of axonogenesis from an early stage of development. They therefore suggest that neurons may be specified or committed (- “instructed” is the term used) to a given fate before they reach their final positions (rather than fate being determined by position). This runs contrary to a model that ML interneurons essentially comprise a single population of cells with two axonal types. Cells that are born early and reside close to Purkinje cells extend axons to envelope the cell bodies (basket cells). Cells in upper layers tend not to develop baskets (stellate cells).

1. (NB: The lower molecular layer is APICAL and the upper layer BASAL. The authors have this reversed throughout and it might be easier to refer to superficial/deep or pial/abpial).

Response R3-1: We thank the reviewer for raising this point. We adopted the terms ‘superficial’ and ‘deep’ in the text to refer to the upper and lower (closer to the Purkinje cell layer) molecular layer , respectively.

The quality of labelling and experimental data in the paper is exemplary and the approach is novel. The cell labelling data is beautiful and the biological problem is an interesting one – trying to decipher the lineages of a relatively unexplored interneuron population in the cerebellum. I have no doubt that this methodological approach would be of interest to a large number of scientists in offering the promise of an automated lineage analysis from mixed populations of morphologies.

The paper presents two conclusions – one biological and one methodological. While the methods are novel and interesting each conclusion raises some concerns.

For the biological model, I think the authors have only generated a question that must now be answered (and has been previously addressed by grafting) by an experimental approach. The most compelling evidence that there is something to investigate further is the inference that basket and stellate cells born in the same temporal cohort might be morphologically distinct throughout their entire development. The data suggests that the statistical method has identified otherwise obscure differences in very young cells that distinguish them by morphology alone (not position). This

needs to be far better illustrated and laminar position needs to be excluded as a factor from the analysis (or better explained).

For the methodological approach, the statistics offer a powerful tool for allocating very similar looking cells to correct lineages based on multivariate analysis morphology alone. While I have reservations that the authors can draw any biological by combining cells from different temporal cohorts (too many assumptions) it is nevertheless an intriguing and very interesting approach. The modelling raises the possibility of distinguishing the ultimate fate of otherwise indistinguishable developing ML and SC cells by calculating their “nearest neighbours” in morphological space. The proof-of-principle – that a cell can be assigned “blind” to a particular cohort – has only been performed with two cells. To be convincing, this needs to be tested with a large population of cells (100’s?) – at least enough to generate statistical confidence limits for the method.

Overall the approach seems to rest on implicit assumptions that could be debatable. They at least need to be justified.

- growth is progressive with no regressive events (branches cannot be lost developmentally).
- cell shape is determined purely by intrinsic factors
- the microenvironment in which cells develop is constant over time. This is particularly important given that transplantation experiments suggest the opposite.

Overall, my feeling is that the biological conclusions are overstated and would be better described as inferences. The extrinsic factors of laminar position, microenvironment etc. are never clearly dealt with. There needs to be a far more considered discussion of the range of explanations for what might be happening in development (remodelling, regressive events) and whether the statistical evidence excludes these alternatives. This is particularly important given that the cerebellum has recently been shown to be an extraordinarily plastic developmental structure (Wojcinski et al 2017. Nat Neurosci 20, 1361-1370).

Seen as an innovative methods paper, the biological questions are less important. The authors may have demonstrated a methods for teasing apart almost indistinguishable populations. However, there needs to be a stronger proof of principle approach (2 cells are not enough).

Detailed comments.

I have divided the detailed comments by Figures. An overall comment is that the distribution of main text and supplementary data is confusing, but that more confusing is the separation of injections, cell morphologies with their respective analysis. The paper would have been much easier to read if it had been broken down into a series of clear steps rather than combining injections data in Figure 5.

****FIGURES 1-4:**

These show that the tendency for MLI cells to form baskets is limited largely, but not exclusively to the lower ML.

2. Within the continuously variable population, I worry that parameters are not dimensionless and that a number of measures are clearly dependent of position (not independent). For example, an apical stellate cell can have no branches directed up and vice versa for a basal Basket cell. Measures of %ML covered by axon and dendrite, length and % of branches directed up or down are just a proxy for laminar position. Axon span is inversely correlated with basket number while axon length remains constant (Fig.S3). This, for example, could suggest that there is relatively consistent length of axon but the overall span is reduced when this finite length is wrapped around Purkinje cell bodies.

R3-2: We thank the reviewer for all of these insightful comments. A strength of the study is that the continuous SC group was obtained by analyzing a multidimensional, morphometric dataset and confirmed by multiple unsupervised clustering tests (ie. UMAP, re-iterative clustering, PAGA; **Figure 2**). To address potential relationships with laminar position, we performed regression analyses of all morphometric features versus soma position within the molecular layer for the entire MLI dataset and the SC subset (this data was shown in previous manuscript and is currently **Supplementary Figure 2**). We find that the majority of parameters are not dependent on laminar position, particularly the axonal features which are necessary and sufficient for BC/SC division (shown in **Figure 3**):

-Of the 27 features, 8 showed a trend with laminar position with regression values, $R^2 = 0.118 - 0.66$. Only 4 of those variables continue to show a correlation for the 61 SCs ($R^2 = 0.16 - 0.62$). These 8 parameters are: Dendritic relative height coverage [%height of molecular layer ML], $R^2 = 0.62$; Dendrites angled up, $R^2 = 0.27$; Dendritic Sholl at 10 um, $R^2 = 0.12$; Dendritic Sholl at 100 um, $R^2 = 0.356$; Axon basket formations, weighted, $R^2 = 0.30$; Axons relative ML height coverage, $R^2 = 0.12$; Axon span, horizontal, $R^2 = 0.13$; Soma volume, $R^2 = 0.16$. When taken together, the regression analysis suggests that a subset of dendritic and axonal parameters co-varies, albeit modestly, with laminar position, further illustrating the spatial character of SC heterogeneity.

-Since dendrites grow to the superficial edge of the molecular layer, it is probable that these 4 dendritic features are dependent on ML position, and the graded dendritic measurements arise from growth constraints of the ML boundary (and has been suggested by Rakic, 1972). However, dendritic parameters are not sufficient for clustering BCs and SCs (**Figure 3**).

-Of the 6 axonal features, basket formation shows the strongest correlation, $R^2 = 0.30$. However, the presence of basket formations alone is not sufficient to subdivide BCs from SCs (**Figure 3**), and several SCs show partial baskets (**Figure 2**).

We comment in the results: “Finally, the axonal parameters co-vary only weakly, if at all, with soma position in the molecular layer (e.g. axon basket formations, $r^2 = 0.30$; axon span $r^2 = 0.13$; Supplementary Fig. 2), further indicating that BC and SC identities are divided on the basis of axonal morphology rather than laminar position.

Thus the morphometric measures are not simply a proxy for laminar position. However, the graded and spatial character to the continuous SCs, suggesting that SC differentiation is influenced by laminar position and potential extrinsic cues.

We elaborate on these points in the Discussion:

“Laminar location is also dispensable for sub-clustering the MLIs, indicating that BC/SC morphological identities do not simply reflect a dependence on laminar position within the molecular layer.

“A subset of parameters describing SC dendrites and axonal span co-varied to a modest degree with molecular layer position, consistent with previous observations of continuous variation in dendritic and axonal structures (Rakic 1972; Paula-Barbosa et al. 1983; Sultan and Bower 1998).

3. It appears that BCs and SCs are morphologically identical apart from the formation of baskets. The only truly discriminative variable for cell type is axon morphology.

For example, Figure 2 beautifully shows that, broadly speaking, only cells in the lower ML have baskets but there are cells with baskets in the middle molecular layer and stellate cells in the lower molecular layer without baskets. The decision (Figure 3) to subjectively classify some cells with baskets as SC3 stellate cells (rather than displaced basket cells) is unclear and slightly confusing.

R3-3: We thank the reviewer for this point. We removed descriptions of ‘subclades SC1-SC4’, as they were used to describe examples of graded morphologies of SCs, but we see that it was confusing. As noted in responses **R1-6** and **R2-2**, the clustering analyses grouped MLIs with partial basket formations and PC soma targeting collaterals into the SC group. In new data shown in **Figure 3e**, elimination of the basket information does not disrupt the BC vs. SC classification. We conclude that BC and SC classification is achieved by a set of multiple axonal parameters, and is not defined by the presence of basket formations alone.

For the entire population, cells that are closer to Purkinje cell layer have bigger soma, a shorter axon span and less densely packed dendritic trees (Figure S3). This trend remains the same when cells with baskets are removed. When the presence of baskets is allowed as a parameter, BC cells segregate as a cluster (Figure S3, S4). When axonal parameters are removed including the discriminatory weighted basket number the cluster disappears (Figure S2B, Figure 4I).

Therefore, while it is fair to say that baskets define basket cells (Line 179), for all other parameters BCs fall within the range of values defined by the SC population and are therefore indistinguishable.

This is a very strong and inherently interesting result. For this reason, a more detailed explanation of the next sections (which argue the opposite) is particularly important.

**FIGURE 5:

4. Tamoxifen labelling at 3 time points beautifully confirms that laminar position is determined by birthdate.

Lower ML cells (and hence the majority of BCs) born first. Some cells in this early born population either fail to form baskets or lose baskets during development.

I think it is important that the possibility that there may be a loss of axons should be taken into account as an alternative. This is particularly relevant given well documented evidence of climbing fibre competition for synaptic territory on the (same) Purkinje cell bodies.

R3-4: The reviewer raises the interesting possibility that SC axon development might undergo regressive events such as basket pruning. In this scenario, loss of the baskets would suggest that developing SCs (i.e. early born lower SCs) are similar to BCs but have lost the canonical BC phenotype. As indicated by reviewer 2 (**R2-6**), regressive changes in morphology could be a caveat in the pseudotime analyses.

From the axonal quantifications of developing MLIs along the pseudotime and the manual validations, we do not detect reductions in total axonal lengths or loss of basket structures for the SC- and BC-fated groups. The data support progressive axonal arborization and growth:

- We visually confirmed the pseudotime order with the progression of the traces in confocal images (i.e. soma and axonal morphology, laminar location; **Figures 8g-h, 9**). Both BC and SC-fated cells show increasing axonal branching at the latest stages examined (~P25). BC-fated cells extend increasing numbers of PC-targeting collaterals and elaborate basket formations at the latest stages examined (i.e. BC in **Figure 8g** at ~P17; **Figure 9k**). These observations are consistent with Telley et al. Neuron 2016, who reported that baskets form late in development following the directed outgrowth of axonal collaterals towards the PC soma.

-Quantifications of individual parameters over the pseudotime or compared to mature stages show increases in total axonal lengths, branch complexity (ie. Scholl), and horizontal axon span (**Figure 8d, 8i; Figure 9e; Supplementary Figure S7**). Projecting these morphometric values across PHATE and Palantir plots illustrate the progression. A notable exception is the horizontal axon spans for presumptive SCs, which retract or at the latest stage (**Figure 8i**).

-We cannot exclude the possibility that there are competitive interactions between early-born MLIs to stabilize axons and elaborate baskets onto PC soma, leading to basket forming vs. non-basket forming cells. Excluding this possibility would require a separate study with manipulations to address axonal competition, and it is beyond the scope of this study. Our quantifications and Our identification that the early-born SCs (non-basket forming) are marked by *Nxph1*⁺ further supports that these cells have distinct molecular identities, in addition to morphological identities. However, additional snRNA-Seq profiling of developing cerebella did not produce obvious clues, as the MLI1/MLI2 trajectories diverge late following a shared *Pax2*⁺ path (Kozareva et al., 2021). Future experiments are needed to track the differentiation of these cells with greater resolution.

5. The following statement could give the impression that authors think that this is evidence of separate fate allocations – a statement that I don't think is supported by the evidence

(Line 224-5): "The marking of early-born MLIs that elaborate distinct phenotypes in the lower ML suggests the divergence of BC and SC fates during early postnatal development".

R3-5: We removed this statement, and other comments that suggest that phenotypic divergence indicates fate allocation to BC and SC fates. Instead, we emphasize that our data show distinct phenotypes and support early divergence of BC and SC morphological phenotypes.

6. I feel that this figure [5] should be broken into two parts and included with Figure 6 and 7 as follows.

****FIGURE 6:**

This should be combined with Figure 5 P0 injections and Figure S5

In the following sections, I feel that the paper could be considerably improved by grouping cell labelling strategy and analysis together and bringing Supplementary Figure 5 into the main body of the paper

Figure 6 concerns only the injections of type that give rise to views in Fig.5C and D.

- The P0 injections in Figure 5C and D should be included with Figure 6 to make it clear that the analysis is of the P0 injected population.
- The labelling strategy, cell morphologies during development and final population structure should all be within the same figure.
- I think it would be very useful to have the stages (1-4) for this population brought in from Fig.s5

R3-6: We thank the reviewer for these suggestions. We re-organized the developmental sections as suggested. In the **revised Figure 8**, we included the injection scheme for early and late born cells as suggested, and aligned the PHATE pseudotime analysis with total cell morphologies. We applied a similar logic to **Figure 9**, to focus on the two early- born populations. To orient the reader, we maintained **Figure 7** to introduce the *Ascl1-CreER* labeling strategy and validation. We retained the staging scheme as a supplementary figure (now **Supplementary Figure S4**) as we could not include all of these images into one coherent figure. We think these changes significantly improve the clarity of the manuscript.

7. It would be really informative to see a photomicrograph of the complex mix of morphologies indicated schematically in Figure 6A (does it look like Fig.8C).

R3-7: **Figure 8** now includes the cartoon (**Figure 8a**) and the confocal images (**8g, h**) showing the mixtures of developmental morphologies typically acquired at a single time point.

8. Finally, the authors should show a representative sample of cells from cluster 5 versus 3 and 6 versus 7 mapped onto laminar position. We have to be visually reassured that laminar position is independent of morphological differences. We need to be able to have clear pictures of the morphological differences between early born stellate and basket cells, which are described in the text (Lines 274-286).

R3-8: In a new **supplementary Figure S9**, we added the raw confocal images for the cells from all clusters shown in **Figure 9**; the upper and lower molecular layer boundaries are annotated to show laminar positions. Since the confocal images are complex with overlapping cells, we maintained the axonal reconstructions, and the soma and dendritic traces in Figure 9. The soma and dendrite traces and their placement (bottom left of **Figure 9 f - n**) are rendered from the raw images.

****FIGURE 7:**

This should be combined with a new, distinct text section and be combined with Fig.5 (P4,7 injections) and Figure S5

The later stage injections (Fig.5) combined with Figure S5 (to give a view of proposed developmental stages 1-4 for this population) could then be combined with PHATE data which the authors feel is a more productive approach (Figure 7D-G).

9. I'm not convinced that Figure 7A, B, C. add much to the paper as a whole and whether these should (if needed to be shown) be placed into Supplementary data.

R3-9: As suggested, we removed the former Figure 7A-C (Palantir analysis to the whole MLI population). In response to reviewer 1 (**R1-2**), we clarify on why the PHATE and Palantir algorithms were used for different aspects of the analyses.

10. My chief concerns about this data is that it mixes cohorts from two very different temporal environments. There are two potential confounding factors:

1. Extrinsic factors: How has the extracellular environment changed between P0, P4 and P7? Cells will grow very differently on different substrates and the subtle differences in shape that cluster cells together may simply be a product of a temporally changing microenvironment

R3-10: We thank the reviewer for this comment, as it touches on important concepts and strengths of our study that we failed to articulate in the Discussion. PHATE and Palantir rendered a pseudotime ordering that followed MLI maturation, as validated by our 'maturation' states (**Figure 8C; new Supplementary Fig S8**). If the analysis was biased by temporal cohorts of collection, then we would expect pseudotime to poorly match the maturation stage, and instead track with cell or animal age. We found the opposite.

We agree that this study does not extract the influence of extrinsic factors, nor control for differences in microenvironments encountered by the P0-labelled versus the P4-P7-labelled MLI cohorts. During postnatal development, the molecular layer expands with the growth of Purkinje cell dendrites and parallel fibers. We expect that different temporal MLI cohorts will be influenced by different extrinsic cues. With the exception of a few studies (ie. *Sema3A-Neuropilin* signaling in basket formation, (Cioni et al *Current Biology* 2013; Telley et al. *Neuron* 2016), the extrinsic factors that pattern MLIs are unknown. These studies do not identify potential differentiation or fate specification factors that influence BC/SC divergence.

We elaborate on these points in the Discussion: TThe clustering is not attributed to temporal cohort effects (i.e. batch effects related to injection or collection times), nor due to canonical axon structures such as basket formations as these appeared at later stages and were not prominently represented in the developmental dataset. The segregation of BC and SC phenotypes during stages of migration suggest that BC and later-born SC differentiation might be influenced by cues that differ with temporally changing microenvironments and/or with laminar location, as the molecular layer expands.

11. #2. Crowding: Are later born cells excluded from the lower ML restricting them to the upper ML.

If laminar position (and co-dependent parameters) are taken out of the statistics would the clustering (and nearest neighbour relations) remain?

R3-11: Yes, once later born cells migrate to the superficial molecular layer they become restricted due to the neuropil of earlier-born MLI axons and Parallel fibers in the lower molecular layer.

To test for the effect of laminar position or axonal span, we rendered the PHATE trajectory following removal of: i) soma location within the molecular layer; ii) soma location, no axon height relative to molecular layer; and iii) Axon span (**new Supplementary Figure S5b-d**). The co-clustering of early born vs late born cells remains, showing that it does not simply arise due to laminar position but rather from multidimensional features. This new data further supports our conclusion that the early-born and late-born MLI populations express divergent morphological phenotypes during development.

12. While the nearest neighbour conjecture is plausible “if the terminal fates of MLI precursors remain undetermined at the beginning of axonogenesis, the identity of a cell’s nearest neighbour should follow a largely random distribution” (line 303-304), this does not allow any conclusion about any biological mechanism or commitment.

The only experimental test of commitment to a certain fate is heterochronic transplantation.

R3-12: We agree with the reviewer that the statistical analyses are not sufficient to draw conclusions on fate commitment or biological mechanisms. The nearest neighbor shows a co-clustering of early-born vs late-born cells based on similarities in morphological phenotypes, and supports our assertion of subtype divergence. We tempered the use of the term ‘fate’, replaced with descriptive terms such as phenotype, and revised the discussion. We considered experiments to address fate commitment of MLI subtypes, but they require identification of molecular drivers that emerge prior to expression of *Sorcs/Grm8* and *Nxph1*, which may or may not exist and are beyond the scope of the current study.

We modified that section in the result: To quantify this observation, we performed nearest neighbor analysis to test if developing MLIs are likely to reside near cells within the same birthdate cohort due to similarities in their morphological phenotypes (Fig. 8f). We reasoned that if the MLI precursors have yet to commit to BC- or SC-specific differentiation during early stages of axonogenesis, the identity of a cell’s nearest neighbor should follow a random distribution. Alternatively, deviation from the null hypothesis infers an early bias of MLI m-type identities. By the earliest pseudotime stage, there is a significant co-sorting of P0-injected BC-fated cells with 77.3% ($\pm 5.4\%$) of nearest neighbors belonging to same early-born cohort, and P4-injected SCs with 82.0% ($\pm 3.9\%$) of nearest neighbors that are SC-fated (Fig. 8f). Thus, pseudotime inference suggests that BC- and SC-fated subtypes can be distinguished by differences in their morphological signatures.

****FIGURE 8:**

13. This confirms that late born cells occupy more superficial laminae and do not make baskets. Beautiful as it is, this figure is not necessary for the paper.

I would not agree that this figure support pseudotime clusterings. What it supports is that axons of cells that become basket cells undergo significant remodelling to make these baskets. The data plots

in Fig. 8G are not clearly explained, nor are any details given of the derivation of statistical significance in the differences in the data (no error bars etc).

This figure is essentially redundant

R3-13: In the updated Figure 8, we combined the confocal images with the PHATE analyses, to clearly validate the performance of the pseudotime ordering, and to illustrate the diversity of developmental states at any one time point. Since each MLI has an ID and its pseudotime rank, we annotated each reconstruction according to its pseudotime rank to show the axonal progression (Figure 8g, h). We agree with the reviewer that these images are useful for visualizing changes in axonal arborization, which further support several findings from the pseudotime ordering. The line plots in Figure 8i show quantifications of individual parameters (soma volume, total axon length, axon span) with 95% confidence intervals for the presumptive BC and SC cohorts. These plots further show progressive axonal development and subtype-specific differences. We clarified the results and figure legend.

****FIGURE 9:**

14. Proof of concept requires a much large number of cells from a range of stages (1-4)

Regardless of any underlying biology, if the manifold was able to correctly allocate a cell to the correct lineage the analysis would indeed prove powerful. In other words, it should be able to statistically identify a cell of a given birth date by morphology alone at any given developmental stage (1-4).

This needs to be done for hundreds of (not just two) cells to derive some statistical power for the approach.

R3-14: Although we were encouraged by the success of the proof-of-concept to predict MLI class within the PHATE readout, we understand the reviewer's point regarding the need for larger sampling to demonstrate statistical power. However it was not feasible to do so for reasons related to reduced research capacity due to COVID, and so we removed this figure from the manuscript.

15. The overall approach is interesting but the interpretation and conclusions are overstated. What the study has determined is an inference that there might be two lineages at P0 (Fig.6) and suggested (but not proved) a powerful predictive approach for analysing morphology (Fig.9) by large scale sampling.

I feel that it cannot show that fates are "instructed" early (line 477). The only test for this is heterochronic transplantation (Leto et al). The main problem (to me) is that, by combining data from multiple temporal cohorts into a single data set, any differences in extrinsic variables are removed.

R3-15: We rewrote the Discussion to elaborate on points raised by the reviewers, on alternative interpretations and caveats, and to temper our conclusions.

Line 477: "Although transplantation studies suggest that commitment to MLI terminal identities remains plastic (Leto et al. 2009), our data indicate that MLI fates are instructed earlier. Moreover, we demonstrated that MLI subtype identities are not dictated by ML laminations."

R3-16: We removed this statement, and other statements relating our findings to fate commitment, etc. We continue to use terms such as identities in specific contexts, when it relates to phenotypes.

For the biological question, factors such as regressive events, a changing microenvironment and, most importantly, that this data is only generating inferences that must be tested are not addressed.

R3-17: We agree that this data generates inferences and new hypotheses that can be tested in future studies. In the Discussion, we raise the possibility that changing microenvironments could influence BC and SC differentiation, leading to their distinct phenotypes: “The clustering is not attributed to temporal cohort effects (i.e. batch effects related to injection or collection times), nor due to canonical axon structures such as basket formations as these appeared at later stages and were not prominently represented in the developmental dataset. The segregation of BC and SC phenotypes during stages of migration suggest that BC and later-born SC differentiation might be influenced by cues that differ with temporally changing microenvironments and/or with laminar location, as the molecular layer expands.

We address the issue of regressive events: “The second notable finding is that BC and SC axonal arborization is progressive. Axonal lengths and complexity increased over the pseudotime and were greatest at maturity, consistent with reports that axonal branching towards the PC soma and baskets form later in development (Telley et al., 2016). From the quantifications and visualizations of MLI labeling, we did not detect axonal pruning but did detect retraction in the axonal span of SCs. Thus, pseudotime ordering can account for regressive events in single or subsets of parameters, further highlighting the utility of this approach for reconstructing morphogenesis. As pseudotime algorithms assume globally linear trajectories however, further investigations are needed to assess the performance for cell types that undergo non-linear growth such as axonal remodeling. Pseudotemporal approaches will also benefit from algorithms optimized for morphometric datasets.

For the powerful methodological conclusions, there is simply not enough data to offer a convincing proof-of-principle. So, for example, this statement is simply not a valid conclusion – but a possible hypothesis (Line 364): “In doing so, we determined that MLI subtype identities emerge during migration prior to reaching sites of final integration”.

R3-18: We removed this statement. In the discussion, we replaced with statements such as:

“Finally, in using these approaches, we also identified the phenotypic divergence of early-born SCs that give rise to the *Nxph1*+ non-basket SCs. These findings further support the idea that m-type identities are established prior to arriving at final locations.

In the section (Line 389) “A revised taxonomy of MLIs”, I am not sure what the authors are proposing is a new taxonomy. Their data clearly supports a continuous variation in dendritic form with a sharp discontinuity in axonal structures (lower MLI cells almost exclusively form baskets). This seems to be – albeit beautifully demonstrated by the experiments - essentially the current model.

R3-17: We maintained the subheading, as it summarizes our findings. As stated in the Introduction and again in the discussion, there were two models prior to our study. One model suggested that

MLIs consisted of discrete stellate cells and basket cells with discontinuous morphological features (championed by Palay, ...). The second is a one cell type model with continuous variability across the molecular layer (supported by work from Bower, etc). We provide a new Model that reconciles both a discrete organization of BCs and SCs, and continuous heterogeneity within the SCs. This model was never formally described, but is quite clear now with our detailed analyses.

In the revised manuscript we also address the lack of correspondence between the morphologically BC/SCs and transcriptionally-defined MLI1/2 subtypes (from Kozareva et al 2021). This new molecular data was initially perplexing and thus required comparison to our morphological study. With our new smFISH analysis, we linked highly expressed transcripts from the t-types to the m-types: BCs = *Sorcs3+*; *Grm8^{HIGH}*; and SC continuum comprising *Nxph1+* and *Sorcs3+*; *Grm8^{LOW}*; *Cacna1^{HIGH}* cells. We also propose that the lack of correspondence between morphologically-defined MLIs with the transcriptomic divisions is due to the developmental genetic programming of morphogenesis: In the Discussion, “A final important consideration is that morphological and connectivity features are established during development. Morphologically-defined subtypes, such as the BC/SCs, might be uncoupled from other cell type divisions because the transient intrinsic and extrinsic programs that shape them are not represented by the transcriptional signatures at maturity.

In the section (Line 427) ”MLI axonogenesis begins during migration” I think all the useful insight come from cell labelling and not from the statistical model. That young cells extend neurites (that may or may not be young axons) is evident from the pictures (Figure S5) and not from the statistics (Line 443).

R3-18: In the revised manuscript, we removed the “”MLI axonogenesis begins during migration” section, and reduced the discussion.

We respectfully disagree that all useful information is restricted to cell labeling. Our observations first came from pseudotime visualization and morphometric quantifications across this progression. The quantitative approaches helped us to make sense of the substantial heterogeneity we obtained during development, and allowed us to discern trends, which we could subsequently validate and refine based on the images. For instance, the pseudotime modeling illustrated changes in axonal lengths and span at early pseudotime stages (Figure 8), which we subsequently identified in confocal images as migratory cells located in the superficial layer.

In the section (line 452) “early emergence of MLI identities”, I find it difficult to get the arguments about biological mechanism and timing of fate decisions. The implications that BC identity is predetermined in white matter, and how numbers might be matched to available Purkinje cell space needs to be better explored. Overall, the cerebellum appears to be a network of cells where cell number matching is tightly controlled by interactions and can be recapitulated following injury (see Joyner lab recent results). However, I do like the argument that lower ML SC might simply be slower to reach their potential targets and fail to establish or retain baskets. This is not the same, I feel, as being “instructed”.

R3-19: We thank the reviewer for raising these new questions and relating to published and preprint literature. We reworded the discussion to state that our data supports an early emergence of MLI identities based on their morphological phenotypes. We removed terms such as ‘instruct’ ie. In the Discussion, “Through novel pseudotime applications that robustly reconstructed and quantified MLI morphogenesis, we detected an early emergence of BC- and SC-fated neurons by divergent axonal phenotypes.

One possibility that the reviewer raises (and in **R3-4**) is competition for basket cell innervation of Purkinje cells, and other mechanisms for cell number matching. Assessing target-dependent axon competition and whether SCs would then form more baskets is a new line of study requiring manipulations. As reported by the Joyner group, the cerebellum has a remarkable plasticity and scaling in response to injury. Although their work focused on granule cell replacement, they recently showed that the later stage gliogenic *Ascl1*⁺ progenitors can undergo adaptive reprogramming and replenish granule cells but could replenish other GABAergic cells. These are all interesting questions that we would like to address in future experiments.

REVIEWER COMMENTS

Reviewer #1 (Remarks to the Author):

The authors have addressed all of my initial concerns. In particular, I applaud their impressive efforts towards a cross-modal analysis of molecular layer interneurons that integrates morphology with single-cell transcriptomics via smFISH. I believe the additions and changes have substantially strengthened the paper, and that it will be an authoritative reference on the subject of molecular layer interneuron cell types.

Reviewer #2 (Remarks to the Author):

The authors have made improvements in the revised manuscript. I no longer have any major problems. However, I would like to suggest the following:

1- In Figure 2h, the location of the Asterix is not consistent with the position of the Purkinje cell (PC) soma. I do not believe that this soma can be located so deep in the molecular layer. In representative figure 2h, the Asterix should be removed and the tip of the branch that finally reaches the PC soma can be highlighted with an arrow.

I am still not convinced that the collaterals of SCs form baskets. I agree that some collaterals reach the soma but without forming a basket as in Figure 2h. I suggest deleting the bracket "(-basket) and (+basket)" and replacing it with (-soma) (+soma).

In the text, line 159: "A subset of long-range SCs extended onto collaterals targeting the PC soma but were otherwise indistinguishable from non-basket forming SCs." I suggest removing the non-basket forming SCs. Authors can use the non-soma targeting SCs, instead.

The author added in their list of 28 morphological parameters "half basket", Could it just be collateral that reaches the soma. The notion of half-basket is not clear to me. I believe that the notion of soma targeting is more appropriate than half-basket.

Reviewer #3 (Remarks to the Author):

The authors have extensively revised their manuscript and include a new set of transcriptomic data that examines the correlation between molecular and morphological identity within the MLI population. The manuscript figures are generally clear and the new data introduces an interesting dimension to the story. As with the original submission the quality of data and its presentation is very high

The revised manuscript clarifies that basket cells (BC) and stellate cells (SC) cluster primarily on the basis of axon morphology. New data shows that transcriptomic phenotype correlates broadly with laminar position of MLIs. The discussion is more balanced in its interpretation of the data, potential caveats and the limits of the analysis.

As in my previous review, I think the manuscript offers an interesting new perspective on methodology (with the important caveat, below) that is of general interest. The study presents a comprehensive description of MLI maturation in terms of morphology and now, molecular phenotype. I am still less convinced that the approach is able to answer the more specific biological question that it sets out to address: whether the prevailing model of MLI specification should be revised.

These two questions still remain for me:

- Has this methodological challenge of possible regressive events been sufficiently accounted for?
- How strong are the conclusions about when phenotype specification occurs?

Regressive events and methodology

Axon refinement in development seems to be a widespread if not universal phenomenon and should be explained and referenced as a potential factor. I think that the assumption that there are no regressive axon events is still hard to justify and needs to be flagged even earlier in the manuscript as a caveat. What would the impact be on the analysis? It is possible that axon pruning would confuse both manual and pseudotime staging? Although the authors state that "We did not observe obvious patterns of axonal pruning that could confound the pseudo-timed measurements. [Line 375-3766], it is hard to see how pruning could be observed using these methods in a mixed population. What would be the hallmark of pruning?

Does the data contradict the prevailing model?

As the authors responded in the rebuttal and now point out in the manuscript: “The segregation of BC and SC phenotypes during stages of migration suggest that BC and later-born SC differentiation might be influenced by cues that differ with temporally changing microenvironments and/or with laminar location, as the molecular layer expands” [Line 545-547].

Differences in cell shape on different substrates do not necessarily equate to specification or fate allocation. We are still left with the possibility that a common MLI precursor (the prevailing biological model), which nevertheless looks different as it migrates within early vs late environments, only becomes specified when it stops migrating. The new FISH data seems to support the prevailing model by indicating that fate is fairly plastic until late stages.

Small points

Line 166. – “no individual features exhibited bimodality” Figure S1 is a box-whisker plot and I’m not sure that I quite agree with the authors assertion. For example, filopodial density looks to be distinctly bimodal in the BC population

Figure 6h – The figure is difficult to interpret with colours of puncta that are so closely matched

Figure 7b and 9a – yellow droplets are very faint. I also think the red versus blue blocks might be a bit more clearly depicted – maybe as a semi-transparent overlays on the timeline

Figure 8a needs to be described in the text (the description in the figure legend is fine). It is important to explain that there are a variety of different stages of cell maturation at any given point of analysis

Line 356 (Figure 8e and f) “this pseudotime inference suggest that BC- and SC- fated subtypes can be distinguished by difference in their morphological signatures”. Could this not simply be that cells born at different times (that ultimately settle in different laminae) encounter different environments and have different morphologies (see above).

Line 369-372 and Fig 8gh. This argument is rather circular. First “a single process ...[is] consistent with MLI precursors that undergo tangential migration” [Line 369] becomes “thus MLIs extend axons during tangential migration” [Line 371}

Line 397 – “clusters confirm our finding that MLI axonogenesis occurs during migration” I don’t think it is really possible for the clustering to substitute for observation here. It is not confirmation but evidence that might support an argument that...

Figure 9 p and q confirms that stellate cells can be born along side basket cells. I think that the argument in the text [Line 409-414] here belongs in the discussion and not the results. Is the observation not equally supportive of an origin for short axon stellate cells as basket cells that fail to make baskets or indeed have lost baskets through remodelling?

AUTHORS' RESPONSE TO REFEREES:

Morphological pseudotime ordering and fate mapping reveal diversification of cerebellar inhibitory interneurons (NCOMMS-20-15800B).

Referee #1 (Remarks to the Author):

The authors have addressed all of my initial concerns. In particular, I applaud their impressive efforts towards a cross-modal analysis of molecular layer interneurons that integrates morphology with single-cell transcriptomics via smFISH. I believe the additions and changes have substantially strengthened the paper, and that it will be an authoritative reference on the subject of molecular layer interneuron cell types.

We thank the reviewer for the positive assessment of the revised manuscript.

Referee #2 (Remarks to the Author):

The authors have made improvements in the revised manuscript. I no longer have any major problems. However, I would like to suggest the following:

1- In Figure 2h, the location of the Asterix is not consistent with the position of the Purkinje cell (PC) soma. I do not believe that this soma can be located so deep in the molecular layer. In representative figure 2h, the Asterix should be removed and the tip of the branch that finally reaches the PC soma can be highlighted with an arrow.

R2-1: We thank the reviewer for this point. We reviewed the raw fluorescent image for the trace shown in Fig. 2h and corrected the location of the PC soma, now annotated with an arrow (image is shown below, with PC soma co-stained using NeuroTrace (Nissl)). The MLI axon collateral does indeed reach deep in the molecular layer, to the base of the PC soma where it elaborates a complex terminal. The MLIs in Figure 11, 1j also target the base of PC soma.

2- I am still not convinced that the collaterals of SCs form baskets. I agree that some collaterals reach the soma but without forming a basket as in Figure 2h. I suggest deleting the bracket "(-basket) and (+basket)" and replacing it with (-soma) and (+soma).

In the text, line 159: "A subset of long-range SCs extended onto collaterals targeting the PC soma but were otherwise indistinguishable from non-basket forming SCs." I suggest removing the non-basket forming SCs. Authors can use the non-soma targeting SCs, instead.

R2-2: We see the reviewer's point that the term 'basket' has a classic meaning used to describe the BCs with canonical, full basket structures. We modified the language, as suggested.

As shown in Figure 2h and Figure 1j, l, we observed many SCs with axon collaterals that form terminals at the base of the PC soma but do not fully envelop the PC soma, as canonical 'baskets' do (with pinceaux). Some BC collaterals also partially cover the PC soma (distal terminals in Figure 1g).

To distinguish these substantial PC soma-targeting structures from those collaterals that simply extend a branch tip to the PC soma, we quantified them using a 'weighted basket scale' described in the Methods, and in the response below.

To address the reviewer's point, we replaced terms such as 'partial basket' and 'basket-forming SCs' with 'PC soma-targeting' or 'SCs with collaterals that target the base of PC soma' in the manuscript:

For Figure 2g, h, we replaced 'basket' labels with 'soma+' and 'soma-', and clarify in the figure legend: "MLI ...extends descending axon collaterals PC soma-targeting axon collaterals (pink arrow at PC soma; axon terminal reaches the PC soma base)."

We modified text in lines 169-170:

"A subset of long-range SCs extended axon collaterals resembling basket formations that partially enveloped PC somas and reached the soma base or axon initial segment (Fig. 2h; see also Fig. 1j, l), but were otherwise indistinguishable from non-soma-targeting SCs (Fig. 2g)."

and lines 194-196

"Additionally, BC-SC clustering remained upon removal of the feature describing axon terminals that fully or partially enwrap PC somas, (Fig. 3e; Supplementary Fig. S1; see Weighted Basket scale in Methods)."

3 -The author added in their list of 28 morphological parameters "half basket", Could it just be collateral that reaches the soma. The notion of half-basket is not clear to me. I believe that the notion of soma targeting is more appropriate than half-basket.

R2-3: We initially adopted the term 'half-baskets' to describe the substantial axon terminals that partially enveloped PC soma, and that are morphologically distinct from those collaterals that simply reach the top of the PC soma. The 'half-baskets' parameter was included in the developmental list of 28 morphological parameters (Supplementary Table 2) but not in the mature parameter list (Supplementary Table 1). We used the 'Half-baskets' feature for the developmental analysis to quantify complex terminals that partially envelop PC soma, and to

distinguish them from the “Number of branches contacting PC soma”. This was an additional parameter to direct and confirm the maturity of pseudotime, as only a small group of maturing cells (21 out of 732 cells) displayed these structures.

We clarify in the Methods the ‘weighted basket scale’ we used to quantify the range of PC soma targeting structures (see line 782):

“For each cell, the number of PC soma-targeting axon terminals were categorized and quantified according to a weighted scale: i) full baskets that fully enwrap the PC soma, and form pinceau structures along the axon initial segment were given a weight of 1; ii) ‘basket-like’ terminals that enwrap PCs and reach the base of the soma or axon initial segment but without pinceau formations were given a weight of 0.75; iii) soma-targeting terminals that partially envelop the PC soma were given a weight of 0.5”.

We also added an image into Supplementary Figure 1 to illustrate our weighted quantification scheme (Supplementary Fig. 1b). A similar version is pasted below.

Weight	Example
1	0.75	0.5	
Referee #3 (Remarks to the Author):

The authors have extensively revised their manuscript and include a new set of transcriptomic data that examines the correlation between molecular and morphological identity within the MLI population. The manuscript figures are generally clear and the new data introduces an interesting dimension to the story. As with the original submission the quality of data and its presentation is very high.

The revised manuscript clarifies that basket cells (BC) and stellate cells (SC) cluster primarily on the basis of axon morphology. New data shows that transcriptomic phenotype correlates broadly with laminar position of MLIs. The discussion is more balanced in its interpretation of the data, potential caveats and the limits of the analysis.

As in my previous review, I think the manuscript offers an interesting new perspective on

methodology (with the important caveat, below) that is of general interest. The study presents a comprehensive description of MLI maturation in terms of morphology and now, molecular phenotype. I am still less convinced that the approach is able to answer the more specific biological question that it sets out to address: whether the prevailing model of MLI specification should be revised.

These two questions still remain for me:

- Has this methodological challenge of possible regressive events been sufficiently accounted for?
- How strong are the conclusions about when phenotype specification occurs?

We thank the reviewer for the positive comments on the revised manuscript, and for pointing out that the “manuscript offers an interesting new perspective on methodology ... that is of general interest”.

We provide point-by-point responses, though lengthy, to ensure that we adequately address the outstanding concerns. We also elaborate in the manuscript on the caveats and future implications of the methodology and biological questions.

1 - Regressive events and methodology

Axon refinement in development seems to be a widespread if not universal phenomenon and should be explained and referenced as a potential factor. I think that the assumption that there are no regressive axon events is still hard to justify and needs to be flagged even earlier in the manuscript as a caveat. What would the impact be on the analysis? It is possible that axon pruning would confuse both manual and pseudotime staging? Although the authors state that “We did not observe obvious patterns of axonal pruning that could confound the pseudo-timed measurements. [Line 375-376], it is hard to see how pruning could be observed using these methods in a mixed population. What would be the hallmark of pruning?

R3-1:

Axon refinement in development seems to be a widespread if not universal phenomenon and should be explained and referenced as a potential factor.

We agree that pruning should be considered a potential factor, and we now expand on this point in the Results and Discussion. We do not know the extent to which pruning is widespread, as studies of axonal refinement in the mouse CNS is limited to a small group of cell types. This knowledge is limited due to the complexity and limited approaches to quantify axonal structures. From the few quantified examples, the degree varies by cell type, and in many cases, pruning occurs at later stages of axon development and affects a subset of parameters. For instance, the classic retinal ganglion cell axon pruning during eye-specific segregation is accompanied by progressive growth and local branching of the terminal arbor, leading to overall increases in total axonal length and branching (Sretavan & Shatz, 1986; PMID 3944621). As we elaborate below and in the Discussion, the potential impact on the pseudotime algorithms will depend on the proportion of features that undergo regressive events. But with these considerations in mind and validations to address them, we see that our quantitative methods will be widely useful for analyzing axonal development.

I think that the assumption that there are no regressive axon events is still hard to justify and needs to be flagged even earlier in the manuscript as a caveat.

We used the morphometric quantifications to detect hallmarks of large-scale axonal pruning: i.e. expansions and then retractions in total axon lengths, branch bifurcations, complexity, etc. Similar comparisons during development and at maturity have been done for other cells that undergo developmental axon pruning [Chandelier cells: Tai et al., 2019; PMID 30846310; Steinecke et al., 2017; PMID 28584877; RGC axons: Sretavan & Shatz, 1986; PMID 3944621; parasympathetic ganglia: Shen et al., 2017; 28157072]. We do not see indications of large-scale axon pruning of MLIs by these measures. We cannot rule out the possibility of some degree of axon refinement or remodeling that is not quantified by our methods. However, it is important to note that our findings in Figure 8 show an early divergence of BC and SC axonal phenotypes, and this precedes the later stages of increased axonal branching and potential refinement. Therefore, our main conclusion is not confounded by this caveat.

Nevertheless, we acknowledge that pruning should be considered as a potential limiting factor in applying pseudotime algorithms to developing neuron populations.

We elaborate on these points:

A new paragraph in the Results, line 408-424, in the Results (pertaining to our MLI analyses):

“A potential limitation of the pseudotime application could arise from axonal pruning, where decreases in several input parameters could weaken the discrimination of maturing cells and confound the pseudotemporal trajectory. To identify regressive trends and other indications of pruning of axonal arbors, we compared BC and SC measurements during development and at maturity. We examined trends across the expert-directed maturation bins that were ordered independently of axonal information, and found that developing BCs and SCs progress in total axonal length and branch complexity (Supplementary Fig.S7). However, these axonal features were smaller compared to maturity, suggesting that BC and SC axonal arborizations increase through later stages of maturation, as local branching continues (Cioni et al., 2013; Telley et al., 2016). A notable exception is that axonal spans for both populations plateau during development. The axonal features of cells ordered by pseudotime also showed similar progressive trends (Supplementary Fig.S7; Fig. 8g-i). Although we could not detect obvious patterns of large-scale axonal pruning in terms of declining axonal features at maturity, these analyses are limited to coarse measures of arbor size. Thus, there is the possibility of some degree of axon refinement or iterative branch addition-retraction at late stages that are not quantified in our dataset. Nevertheless, our analyses suggest that the segregation of BC and SC axonal phenotypes occurs during earlier stages of axonal development.”

In new plots shown in Supplementary Figure S7, we show the progressive increases of axonal features (complexity, total axon length, bifurcations) across the manual staging, and compared to maturity.

We added in the Discussion, lines 613-631:

“The second notable finding is that BC and SC axonal arborization is largely progressive. MLI axonal lengths and complexity increased over the manual and pseudotime staging and were greatest at maturity, consistent with the increased BC axon branching around PC soma observed later in development (Telley et al., 2016). We did not detect obvious large-scale axonal refinement that would have been marked by expansions then retractions in multiple axonal features, as recently shown for cortical chandelier interneuron arbors (Tai et al., 2019; Steinecke

et al., 2017). We did detect retraction of axonal spans of SCs, indicating that pseudotime ordering can account for regressive events in single or subsets of parameters, further highlighting the utility of this approach for reconstructing morphogenesis. As our methods are limited to coarse changes in axonal arborization, we cannot discount the possibility that MLIs undergo finer regressive events. In our study, the BC/SC phenotypic segregation was quantified early in axonogenesis, prior to the late phase of local branching and potential remodeling. Nonetheless, axonal refinement is an important consideration, as multiple regressive parameters could limit the performance of pseudotime algorithms that assume globally linear trajectories (Tritschler et al., 2019). Further investigations are needed to assess the performance for cell types that undergo non-linear growth. Increasing the number of input parameters, adding progressive innervation features such as varicosities or terminal structures, and developing algorithms optimized for morphometric datasets are future directions for strengthening and broadening pseudotime applications for studies of neuronal diversification.

What would the impact be on the analysis?

To relate the potential impact of pruning on pseudotime analysis, we first broadly distinguish two types of pruning: #1 arbor-scale pruning that changes the arbor size/shape, and leads to significant decreases in a few or several parameters; #2 fine-scale refinement, such as branch elimination, and detected by smaller changes in a few parameters.

#1: In an example of large-scale refinement, we show in Fig. 8i that the pseudotime detected a regressive change in stellate cell (SC) axon arborization in axonal span, although most other parameters followed progressive trends. This is an example in which the pseudo-timeline can recapitulate the developmental progression of the cells if there are a small number of parameters that regress but for which the majority of measurements progress in a unidirectional manner. As the reviewer points out, limitations of this approach could arise when a majority of input features regress. It is conceivable that for such datasets, pseudotime algorithms might not properly sort cells if they cannot statistically distinguish cells in the axonal pruning phase vs those in the growth phase with similar morphometric profiles. It has been pointed out as a general limitation of pseudotime algorithms by other groups (i.e. for looped data structures; new reference Tritschler et al., 2019 was added to the Discussion).

For #2, pseudotime will not track fine-scale refinements such as iterative branch addition/retraction or local branch elimination. Finer-scale quantifications or live imaging are required to characterize this scale of axon refinement.

It is difficult to fully predict the impact of regressive events on the analysis without formally testing pseudotime algorithms with an entirely new dataset of cells with a known phase of pruning. This is beyond the scope of this study but in our future plans.

What would be the hallmarks of pruning?

We expected to quantify pruning as morphometric reductions at maturity compared to earlier stages, ie. reductions in total axonal length, axon span, branch bifurcations, number of branches contacting PC soma, and/or axon complexity/Scholl, etc. The strength of our experimental design is the addition of the ‘manual staging’ of cell traces into four maturation bins, to compare changes in axonal features across these stages and at maturity.

Visually, after generating and analyzing 100s of reconstructions, the first author did not observe obvious reductions in overall arbor complexity, size or shape in mature MLIs compared to younger ones. We agree that it can be difficult to detect differences, especially in mixed populations. Again, our findings are strengthened by quantitative and visual validations.

It is possible that axon pruning would confuse both manual and pseudotime staging? it is hard to see how pruning could be observed using these methods in a mixed population.

It is possible that pseudotime staging could be confused due to axon pruning if multiple parameters undergo regressive changes. The expert-directed manual staging was not confused by potential axon pruning, but rather provided a separate method to track the developmental progression of traces within the mixed populations, because dendritic arborizations were progressive across these 4 stages. The manual staging provided a ‘trajectory’ that is not informed by axonal information, and thus not confounded by the possibility of pruning. Another feature we used is the relative laminar position (as cells in the lower ML within a labeled cohort are more mature and have increased local branching, as depicted in Figure 8g, h).

Thus the manual staging turned out to be a useful way to compare the axonal parameters across bins to determine if there are significant reductions that would indicate large-scale pruning. In new plots in Supplementary Fig.7, we show that total axon lengths, complexity, and branching of fate-labeled BCs and SCs increase along the developmental dataset. These axonal features are also greater in the mature datasets, indicating that axonal arborization and branching continues during later stages of maturation (which was not captured as deeply in our developmental dataset). Comparisons of the axonal features using the pseudotime stages show similar results (Supplementary Fig. 7, Figure 8).

We cannot rule out that some degree of branch refinement or retraction/addition occurs during the later stages, and that modest changes in the input features would weaken or confuse the pseudotime output. However, this caveat does not change our findings in Figure 8, because the phenotypic divergence detected by the pseudotime staging occurs at earlier stages of development, when MLIs have migratory and simpler axonal morphologies.

We hope that the reviewer sees how, with these approaches, we reconstructed axonal development of MLI subpopulations with information on lineage, position, and arbor parameters through maturation. We are eager to extend the concept of morphological pseudotime to track the development of other cell types, and advance automated methods of neuronal reconstructions and quantifications.

2- Does the data contradict the prevailing model?

As the authors responded in the rebuttal and now point out in the manuscript: “The segregation of BC and SC phenotypes during stages of migration suggest that BC and later-born SC differentiation might be influenced by cues that differ with temporally changing microenvironments and/or with laminar location, as the molecular layer expands” [Line 545-547].

Differences in cell shape on different substrates do not necessarily equate to specification or fate allocation. We are still left with the possibility that a common MLI precursor (the prevailing biological model), which nevertheless looks different as it migrates within early vs late environments, only becomes specified when it stops migrating. The new FISH data seems to

support to the prevailing model by indicating that fate is fairly plastic until late stages.

I am still less convinced that the approach is able to answer the more specific biological question that it sets out to address: whether the prevailing model of MLI specification should be revised.

R3-2: As outlined in the manuscript, the differentiation of MLIs into BC and SC subtypes is unknown (Lines 81-87; 325-327). Based on limited birthdating and transplantation studies, the prevailing model suggests that the interneuron lineage, including MLI precursors, differentiates in an inside-out maturation gradient that is temporally related to birthdate, but that their fates remain plastic until they reach target locations (Leto et al. 2006; 2009). The ‘fate’ readout was final laminar position, though a few examples of BC and SC morphologies indicate that transplanted cells adopt host-specific canonical phenotypes.

There are two important points here: #1: The limitation of the heterochronic transplantations is that it does not distinguish between the host environments encountered during migration vs those at their final location. Both environments could provide instructive cues. #2: Cell specification does not preclude developmental plasticity, as cells that are ‘specified’ can remain responsive to extrinsic cues and can be reversed or transformed in a different environment. Thus, BC and SC diversification could be specified at earlier stages, but final fates/phenotypes can remain plastic until their final placement.

The outstanding question we address here is: Do BC and SC identities (or phenotypes) diverge at their final location after migration, or earlier during their migration. Another advance is that we use a genetic fate-mapping tool to track MLI subtype development, and use empirically-derived morphological data for a phenotypic readout of subtype identities.

Our methodology and data answer the biological question by:

A) demonstrating that BC- and SC-fated (early/late born) cells express divergent axonal growth phenotypes during migration, before settling at the final locations. As pointed out by the reviewer, we suggest in the Discussion that “BC and later-born SC differentiation might be influenced by cues that differ with temporally changing microenvironments and/or with laminar location, as the molecular layer expands.”

B) identifying an early phenotypic division of early-born SCs from early-born BCs, despite sharing overlapping birthdate, migration environment, and laminar position. The early-born SCs give rise to molecularly and morphologically distinct *Nxph1*+ non-basket forming SCs.

whether the prevailing model of MLI specification should be revised.

These results revise the model in that BC and SC phenotype specification is detectable during migration. Our new model does not contradict or completely change the prevailing model. Our study does not directly test the plasticity for MLI precursors to adopt other subtype identities, nor revisits whether MLI subtypes are predetermined. These are questions we are eager to investigate in the future, *now* that we have uncovered new information on the spatial-temporal features of MLI diversification, the identification of an early-*Nxph1*+ SC, as well as new tools and markers.

We are still left with the possibility that a common MLI precursor (the prevailing biological model), which nevertheless looks different as it migrates within early vs late environments, only

becomes specified when it stops migrating. The new FISH data seems to support the prevailing model by indicating that fate is fairly plastic until late stages.

Classically, cell specification does not rule out plasticity, as ‘specified’ cells can respond and reverse to new environmental cues (as opposed to commitment, which is not reversible). Thus, our results that BC and SC phenotypic specification occurs during migration is compatible with the idea that they can retain plasticity until they stop migrating, and then express maturation markers (smFISH data). In this working model, BC and late SC undergo phenotypic specification during migration, but commit to terminal fate and a mature subtype phenotype when they reach their final location. Formally testing this model is beyond the scope of this current study. Our future plan is to profile developing MLIs to define the transcriptomic signatures associated with this differentiation trajectory, including specification, and correlate to morphological differentiation.

We added this statement to the Discussion (lines 607-611):

“In this model, the phenotypic specification of BC/SC during migration does not exclude the possibility that MLIs retain plasticity until they arrive at their final locations and express maturation t-type markers. Identifying cues that determine MLI subtype specification versus commitment will be important for understanding how interneuron heterogeneity deriving from a common progenitor pool is established.”

Differences in cell shape on different substrates do not necessarily equate to specification or fate allocation.

We disagree with this point. It is reasonable to equate phenotypic differences as a readout for BC/SC phenotypic specification, because the morphological differences provide empirical evidence for subtype divergence within the lineage. There are parallels to conventional transcriptional differences. As specification is not equivalent to commitment, MLIs in this state can be further shaped by different cues. Our approach detected distinct developmental trajectories, which are likely emerging due to interactions with their different spatial/temporal environments.

Note that we have refrained from using the terms ‘fate’ and ‘specification’ when describing our results and conclusions, and instead use ‘phenotypes’ and ‘identities’. We acknowledge that specification or fate allocation have particular definitions, and confirming these states requires manipulations as well as knowledge of molecular or regulatory signatures for these different stages.

How strong are the conclusions about when phenotype specification occurs?

Together, our data support our conclusion that MLI subtype identities emerge during migration, on the basis of quantified morphological phenotypes. The divergence of early and late born MLI identities may be influenced by interactions with their spatial-temporal environment, giving rise to distinct Sorcs3+ BC and SC subtypes. The discovery of the early-born Nxp1+ SCs raises new questions on the mechanisms leading to early-born BCs and SCs. Our findings provide new spatial/temporal features of BC/SC phenotypic specification which will inform future studies on the mechanisms regulating their specification and diversification. Answering these questions is important, as they can advance a broader understanding of how developmental plasticity and extrinsic cues contribute to the diversification and local tuning of interneuron populations.

Small points

a. Line 166. – “no individual features exhibited bimodality” Figure S1 is a box-whisker plot and I’m not sure that I quite agree with the authors assertion. For example, filopodial density looks to be distinctly bimodal in the BC population

The box whisker plots show median, first (25%) and their (75%) quartiles. We see that the filopodial density in the BC dataset is not normally distributed about the median. However, the point we make in the text refers specifically to the SC population, which has a larger sample size and thus permits us to analyze the descriptive statistics and make the point about data distribution.

The data points superimposed onto the box-whisker plots provide a means to visualize distribution for the SC population. We note that SC distributions for some parameters are skewed: ie. Axon length along the ML, Scholl intersections, Dendrite depth, Dendrite Length. We verified descriptive statistics, and these do not indicate bimodal distributions. To further verify the distribution, we plotted these parameters in violin plot format (attached below), and also included skew and kurtosis values in the source data. We do not observe evidence for bimodality in these parameters for the SCs.

In the Supplementary Figure legend for S1, we removed: All parameters follow a largely normal distribution, with no presence of bimodality observed.

b. Figure 6h – The figure is difficult to interpret with colors of puncta that are so closely matched

We thank the reviewer for the suggestion. It is challenging to present multiplexed imaging data with four colors. We tested other color combinations but did not find one that significantly improves the visibility of all channels, especially since we wanted to maintain a panel with colored puncta overlaid on the greyscale-DAPI labeling. For example, replacing green for yellow makes it difficult to see the yellow puncta on the nuclei. We hope the reviewer will agree that the single-channel images presented with DAPI outline, in addition to the merge versions, allow the reader to follow and interpret the cellular localization patterns.

c. Figure 7b and 9a – yellow droplets are very faint. I also think the red versus blue blocks might be a bit more clearly depicted – maybe as a semi-transparent overlays on the timeline

In Fig. 7b and 9a, we increased the opacity of the droplets. In Fig. 7b, we added the timeline as a semi-transparent overlay, with pink for early-born and teal for late-born injections. We hope the reviewer finds these modifications improve the visualization.

d. Figure 8a needs to be described in the text (the description in the figure legend is fine). It is important to explain that there are a variety of different stages of cell maturation at any given point of analysis

We added to Line 344:

“One challenge to quantifying developmental changes at a large scale is the variability in the progression or maturation of cells present in the tissue at any given time point. To address this confounding feature of development, we adapted a pseudo-temporal ordering approach to align snapshots of single-neuron morphologies over the course of maturation (Fig. 8a).

e. Line 356 (Figure 8e and f) “this pseudotime inference suggest that BC- and SC- fated subtypes can be distinguished by differences in their morphological signatures”. Could this not simply be that cells born at different times (that ultimately settle in different laminae) encounter different environments and have different morphologies (see above).

As we elaborate above in R3-2, we agree with the reviewer that the different morphologies of a lineage of cells born at different times could be shaped by the changing environment or differentiation programs. These observations and the morphological quantifications support a divergence in subtype identities, which is an objective of this study. We addressed this point in the Discussion, lines 596-611:

“The early segregation of BC and SC phenotypes rendered by PHATE occurs during migration, as indicated by the pseudotime staging and related images showing MLIs located in the upper layer with elongated soma and simple dendritic processes that are characteristics of tangential migration (Wefers et al., 2017, 2018). ... The segregation of BC and SC phenotypes during migration suggests that BC and later-born SC differentiation might be influenced by cues that differ with temporally changing microenvironments and/or with laminar location, as the molecular layer expands.”

As we elaborate in R3-2, we added this text to this paragraph:

“In this model, the phenotypic specification of BC/SC during migration does not exclude the possibility that MLIs retain plasticity until they arrive at their final locations and express maturation t-type markers. Identifying cues that determine MLI subtype specification versus commitment will be important for understanding how interneuron heterogeneity deriving from a common progenitor pool is established.

f. Line 369-372 and Fig 8gh. This argument is rather circular. First “a single process ...[is] consistent with MLI precursors that undergo tangential migration” [Line 369] becomes “thus MLIs extend axons during tangential migration” [Line 371]

We think that the reviewer notes that the sentences are repetitive. We simplified the statement, lines 396-398:

“MLIs at early pseudotime stages were *Pvalb*-negative and located in the upper molecular layer (Fig. 8g,h; Supplementary Fig. S6), consistent with tangentially migrating MLI precursors (Cameron et al., 2009; Groteklaes et al., 2020; Wefers et al., 2017). They bore a single process that elaborates at later stages, suggesting that MLIs extend axons during migration, before reaching their final positions.”

g. Line 397 – “clusters confirm our finding that MLI axonogenesis occurs during migration” I don’t think it is really possible for the clustering to substitute for observation here. It is not confirmation but evidence that might support an argument that...

We changed this sentence to: “Cells in the intervening clusters display intermediate morphologies that support the idea that MLI axonogenesis proceeds during migration. Immature BCs in clusters 2 and 3 elaborate a tangentially oriented trailing process along the superficial ML (Fig. 9g,h). As BCs mature in clusters 4, 6, and 8, they occupy lower laminar positions and arborize axons (Fig. 9i-k; Supplementary Fig. S9).”

h. Figure 9 p and q confirms that stellate cells can be born along side basket cells. I think that the argument in the text [Line 409-414] here belongs in the discussion and not the results. Is the observation not equally supportive of an origin for short axon stellate cells as basket cells that fail to make baskets or indeed have lost baskets through remodelling?

For the first point, we modified the statement within the results, lines 457-459:

“Therefore, BCs and lower SCs residing in the lower molecular layer are born in the same temporal cohort but exhibit distinct axonal phenotypes. Together, these findings suggest an overlapping, rather than sequential, emergence of BC/SC identities.

We moved the statement to the first Discussion paragraph, lines 486-489:

“In contrast to the idea that BC and SC m-types are generated sequentially and differentiate at final laminar positions (Leto and Rossi, 2012; Leto et al., 2009), our findings support a model of BC/SC subpopulations that arise from early-born MLIs and express divergent axonal development during their migration.

For the second point, “equally supportive of an origin for short axon stellate cells as basket cells that fail to make baskets or indeed have lost baskets through remodelling”, we raise this possibility in the Discussion, lines 641-654:

“One possibility is that early-born MLIs are initially equivalent, but a subset is at a disadvantage and fails to form baskets or retracts their soma-targeting processes, producing the non-basket lower SCs. While our data do not exclude this possibility, we observed that PC-soma targeting occurs at late stages of maturation, consistent with previous reports (Cioni et al., 2013). Basket formation is dependent on secreted Sema3A signaling and Neuropilin1/Neurofascin186 interactions (Cioni et al., 2013; Telley et al., 2016), and thus it would be interesting to test whether the Nrp1 signaling is restricted to the BC-fated subset of cells. Since the PWM is hypothesized to be instructive for MLI genesis and differentiation (Leto and Rossi, 2012; Leto et al., 2009), another possibility is that divergent early-born BC and SC fates are instructed by cues in the PWM or along their migration. For instance, BCs and lower SCs are born at similar times, but lower SCs might adopt a different fate leading to *SC-Nxph1*⁺ identity by taking longer to traverse the PWM and molecular layer. Further lineage tracing and cross-modal profiling studies are needed for defining MLI differentiation and relevant signals at these earliest stages, and to inform tests on the mechanism and timing of BC/SC fate decisions.”

We examined a very large dataset of MLI morphologies through development, and did not observe an obvious initial excess and subsequent loss of baskets. However, we acknowledge that addressing this question requires manipulations to reduce the number of BCs or competitive interactions between these cells. These experiments are beyond the scope of this current study. Based on the data in the current study, it is reasonable to emphasize our working model that the BC and short axon SC subpopulations diverge from early-born MLIs and settle in shared locations in the molecular layer. We also present alternate possibilities in the Discussion (ie. statements above). Given that MLIs form baskets at late stages, which also correspond to onset of PV expression and presumably t-type marker expression, we are eager to test our prediction that as BCs and non-basket forming SC elaborate at their target areas, they are already molecularly *Sorcs3*⁺ and *Nxph1*⁺ distinct, in addition to morphologically distinct. We plan to interrogate the origins of these two subpopulations in future studies. Taken together, we hope the reviewer is excited about how this present work sets a foundation for understanding cerebellar interneuron diversification, but for shedding light on the broader concept of local diversification of inhibitory interneurons through quantifications of morphological and in situ transcriptional signatures.

REVIEWERS' COMMENTS

Reviewer #3 (Remarks to the Author):

The authors have revised the text of the manuscript and responded to all my points with a series of revisions that give a more balanced biological consideration of the results generated by their methods. As in my original review, I think that the data is beautiful and the manuscript describes a methodological approach that will interest the broader neuroscience community - all my concerns are now addressed.